# A Little Help Goes a Long Way: Efficient LLM Training by Leveraging Small LMs

## Abstract

A primary challenge in developing large language models (LLMs) is their onerous pre-training cost. This paper explores a promising paradigm to improve LLM pre-training efficiency *and* quality by leveraging a *small* language model (SLM). In particular, this paradigm relies on an SLM to both (1) provide soft labels as additional supervision, and (2) select a small subset of valuable training examples. Put together, this enables an effective transfer of the SLM's predictive distribution to the LLM, while prioritizing specific regions of the training data distribution. Empirically, this leads to reduced LLM training time compared to standard training, while improving the overall quality. Theoretically, we develop a statistical framework to study the utility of SLMs in enabling efficient training of high-quality LLMs. Our framework characterizes how the SLM's seemingly low-quality supervision can enhance the training of a much more capable LLM. Furthermore, it also highlights the need for an *adaptive* utilization of such supervision, by striking a balance between the bias and variance introduced by the SLM-provided soft labels. We corroborate our theoretical framework by improving the pre-training of LLMs with 2.8B and 8.6B parameters by utilizing smaller LMs on the Pile dataset.

## 1 Introduction

Owing to the surge in their ever-growing capabilities, large language models (LLMs) (OpenAI, 2023; Gemini-Team et al., 2025; Anthropic, 2024; DeepSeek-AI et al., 2024) have become the focal point of machine learning research. *Pre-training* highly capable general-purpose LLMs is extremely expensive due to large model sizes (Chowdhery et al., 2022) and massive training corpora (Meta, 2025). Thus, sustainable advancement and adoption of LLMs hinges on designing novel algorithms that can reduce the overall pre-training compute cost and data requirement for LLM development.

This paper focuses on leveraging *small language models* (SLMs) for efficient LLM pre-training. A growing literature (see, e.g., Gupta et al., 2024; Chen et al., 2023; Yue et al., 2024) shows that SLMs can acquire a good understanding of pre-training data distribution despite their limited model capacity. Particularly, SLMs can perform well on a large portion of "*easy*" instances, and help identify the remaining "*hard*" instances, e.g., via confidence of their predictions. This prompts us to explore:

*Can we speed up pre-training of a high-quality* large *LM by transferring the predictive distribution from a lower-quality* small *LM?*

Suitable SLMs are often readily available during LLM development, either as previous generation models trained on similar pre-training corpora, or smaller models trained for initial exploration around architectural and algorithmic choices on the current pre-training corpora. Furthermore, the potential of SLMs to enhance LLM quality and efficiency, coupled with their relatively cheaper development cost, justifies training such models even to aid LLM training solely, especially since their training cost can be amortized by leveraging them to train multiple LLMs.

*Knowledge distillation* (KD; Bucilă et al., 2006; Hinton et al., 2015) is a natural candidate to achieve our objective by utilizing the SLM as a *teacher* model to transfer its predictive distribution to a *student* LLM during pre-training. However, it is unclear if KD can help realize our goal, as unlike a typical KD setup – where a larger or stronger teacher trains a smaller or weaker student (Team et al., 2025; Meta, 2025) – our proposal is to leverage a smaller and *weaker teacher* LM to improve the training efficiency and quality of a larger and *stronger student* LM.

We begin by developing a statistical framework to study KD for language modeling. We derive novel risk bounds that identify the desirable properties of the teacher LM-provided supervision for

enhancing the student LM's performance, even when employing a significantly weaker SLM as the teacher. To our knowledge, ours are the first such bounds for language modeling. Notably, our bounds subsume standard pre-training as a special case, and control LM generalization as one scales model size and pre-training data. These bounds may be of independent interest to the broader community.

Our statistical analysis lays the foundation for an adaptive pre-training method that leverages SLM via KD only in "easy" regions, where the SLM can approximate the ground truth next-token distribution well. Combining this with the tendency of neural network to learn easier examples first (Kalimeris et al., 2019; Refinetti et al., 2023), we propose *small model aided large model training* (SALT), a two-stage pre-training approach wherein the first phase employs KD from an SLM, and the sec-

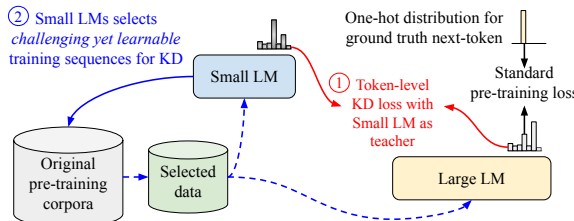

Figure 1: An overview of SALT, which utilizes an SLM in two ways to improve the pre-training of LLM: ① To perform KD with SLM as teacher in the early phase of LLM pre-training; and ② To obtain a valuable subset of pre-training corpora to be utilized during the KD.

ond phase reverts to standard (next-token prediction-based) pre-training. We then expand SALT by employing an SLM to additionally perform *data selection* for the KD phase. Our selection procedure focuses on identifying *challenging* yet *learnable* sequences from the easy region of the data distribution to ensure an effective transfer of SLM's predictive distribution during KD (see Fig. 1 for an overview). Our key contributions are:

(1) We present a statistical framework for KD in language modeling, which delineates how even a significantly weaker teacher LM can improve the quality of a larger student LM (§3).

(2) Guided by our analysis, we propose SALT, a two-stage pre-training method that employs KD with an SLM teacher in the first stage. We then extend SALT by using SLMs to perform data selection for the KD, facilitating effective transfer of predictive distribution from SLM to LLM (§4).

(3) We showcase the utility of SALT (with and without data selection) by training 2.8B and 8.6B LMs with the help of smaller LMs, on the Pile dataset (Gao et al., 2020). SALT-produced LLMs *outperform* the same-sized LMs trained via standard pre-training on a comprehensive set of few-shot benchmarks while utilizing less that $0.7\times$ training step budget, resulting in $\sim25\%$ and $\sim28\%$ wall-clock time reduction for 2.8B and 8.6B LMs, respectively. Moreover, SALT models demonstrate significant downstream performance gains after SFT on multiple domains (§5).

## 2 BACKGROUND

**Language modeling.** Given a large corpus, language modeling aims to train a model that can assign probabilities to each sequence $\mathbf{x} \in \mathcal{V}^\star$, where $\mathcal{V}$ denotes the vocabulary with $V = |\mathcal{V}|$ tokens. Given a $T$-token long sequence $\mathbf{x} = [x_1, x_2, \ldots, x_T]$, a language model (LM) parameterized by $\boldsymbol{\theta}$ assigns the probability $P_{\boldsymbol{\theta}}(\mathbf{x}) = P_{\boldsymbol{\theta}}(x_1)P_{\boldsymbol{\theta}}(x_2|x_1)\cdots P_{\boldsymbol{\theta}}(x_T|x_1,\ldots,x_{T-1})$ to $\mathbf{x}$. Transformer (Vaswani et al., 2017) is the most prominent architecture for modern LMs, which we briefly discuss in Appendix A.1.

**Standard LM pre-training.** Typically, LM pre-training utilizes the *next-token prediction* task: given a training sequence $\mathbf{x} = [x_1, \ldots, x_T]$, for each $t \in [T]$, one maximizes the log-likelihood $\log P_{\boldsymbol{\theta}}(x_t|\mathbf{x}_{\leq t-1})$. This amounts to minimizing the *cross-entropy* loss between the per-token LM prediction distribution $P_{\boldsymbol{\theta}}(\cdot|\mathbf{x}_{\leq t-1})$ and the one-hot distribution $\mathbb{1}_{x_t}(\cdot)$ defined by the *ground truth* next-token $x_t$ (for $v \in \mathcal{V}$, $\mathbb{1}_x(v) = 1$ iff $v = x$). Thus, the overall loss associated with $\mathbf{x}$ becomes

$$\ell(\mathbf{x}; \boldsymbol{\theta}) = 1/T \cdot \sum_{t \in [T]} -\log P_{\boldsymbol{\theta}}(x_t|\mathbf{x}_{\leq t-1}) = 1/T \cdot \sum_{t \in [T]} \mathsf{CE}\big(\mathbb{1}_{x_t}(\cdot), P_{\boldsymbol{\theta}}(\cdot|\mathbf{x}_{\leq t-1})\big), \quad (1)$$

where $\mathsf{CE}(P_1, P_2) = -\sum_{v \in \mathcal{V}} P_1(v) \log P_2(v)$ is the cross-entropy between distributions $P_1$ & $P_2$.

**Knowledge distillation for LMs.** Going beyond the ground truth next-token based loss in (1), one can utilize the per-token prediction distribution provided by another LM parameterized by $\boldsymbol{\zeta}$ as additional supervision. Formally, given the context $\mathbf{x}_{\leq t-1}$, one can train the LM parameterized by $\boldsymbol{\theta}$ via aligning its prediction distribution $P_{\boldsymbol{\theta}}(\cdot|\mathbf{x}_{\leq t-1})$ with $P_{\boldsymbol{\zeta}}(\cdot|\mathbf{x}_{\leq t-1})$. KL divergence is a common choice to promote such an alignment, which amounts to minimizing the following cross-entropy loss:

$$\ell^{\boldsymbol{\zeta}}(\mathbf{x}; \boldsymbol{\theta}) = 1/T \cdot \sum_t \mathsf{CE}\big(P_{\boldsymbol{\zeta}}(\cdot|\mathbf{x}_{\leq t-1}), P_{\boldsymbol{\theta}}(\cdot|\mathbf{x}_{\leq t-1})\big). \quad (2)$$

The objective in (2) corresponds to *token-level KD* (Kim & Rush, 2016), with $\boldsymbol{\zeta}$ and $\boldsymbol{\theta}$ serving as the *teacher* and *student* LMs, respectively. Appendix A.2 discusses KD for LM variants. *Temperature scaling* of teacher is a common strategy (Zheng & Yang, 2024) where, given a temperature $\rho > 0$, one utilizes $P_{\boldsymbol{\zeta}^\rho}(\cdot|\mathbf{x}_{\leq t-1}) = P_{\boldsymbol{\zeta}}(\cdot|\mathbf{x}_{\leq t-1})^\rho / \sum_{v' \in \mathcal{V}} P_{\boldsymbol{\zeta}}(v'|\mathbf{x}_{\leq t-1})^\rho$ during KD, resulting in the loss:

$$\ell^{\boldsymbol{\zeta}^\rho}(\mathbf{x};\boldsymbol{\theta}) = 1/T \cdot \sum_t \mathsf{CE}\big(P_{\boldsymbol{\zeta}^\rho}(\cdot|\mathbf{x}_{\leq t-1}), P_{\boldsymbol{\theta}}(\cdot|\mathbf{x}_{\leq t-1})\big). \tag{3}$$

In practice, one typically minimizes a weighted combination of both (1) and (3) with *distillation loss weight* $\omega \in [0,1]$, yielding the following loss for $\mathbf{x}$ (we omit the dependence on $\boldsymbol{\zeta}$ and $\rho$ for brevity):

$$\ell^\omega(\mathbf{x};\boldsymbol{\theta}) \triangleq (1-\omega) \cdot \ell(\mathbf{x};\boldsymbol{\theta}) + \omega \cdot \ell^{\boldsymbol{\zeta}^\rho}(\mathbf{x};\boldsymbol{\theta}). \tag{4}$$

# 3 THEORETICAL ANALYSIS: WHEN CAN KD HELP LANGUAGE MODELING?

Our goal is to utilize KD with an SLM as the teacher to improve LLM pre-training. Towards this, we develop a statistical framework to study KD for language modeling by building on Menon et al. (2021); Dao et al. (2021a); Ren et al. (2022a). The resulting novel risk bounds show how even a *weaker* teacher can aid student LLM by striking the right balance in terms of a bias-variance trade-off.

Notably, our analysis controls the student LM's generalization gap in terms of both the number of training sequences $N$, as well as the number of tokens $NT$. The latter is highly non-trivial due to possibly arbitrary dependence within a training sequence, and crucially leverages certain natural stability conditions on the underlying distribution and function class. Next, we setup necessary notation and then present our risk bounds as functions of $N$ and $NT$ in §3.1 and §3.2, respectively. §3.3 utilizes our bounds to justify the utility of SLMs for improving LLM model quality via KD.

Let $\mathcal{D}$ be the data distribution that generates $N$ independent training sequences $\mathcal{S}_N = \{\mathbf{x}^{(i)}\}_{i \in [N]} \subset \mathcal{V}^T$, i.e., $\mathbf{x}^{(i)} \sim \mathcal{D}$. (Our analysis can be extended to *varying* length sequences at the cost of increased notational complexity.) Given $\mathcal{S}_N$ and CE surrogate loss (cf. (1)), we define the *empirical surrogate risk*, i.e., standard training objective, and its population version for an LM parameterized by $\boldsymbol{\theta}$ as:

$$R_N(\boldsymbol{\theta}) = 1/N \cdot \sum_{\mathbf{x} \in \mathcal{S}_N} \ell(\mathbf{x};\boldsymbol{\theta}); \quad R(\boldsymbol{\theta}) = \mathbb{E}_{\mathbf{x}}\big[\ell(\mathbf{x};\boldsymbol{\theta})\big]. \tag{5}$$

On the other hand, the empirical surrogate risk for KD, i.e., the KD training objective, and its population version take the following form (note that we omit dependence on $\boldsymbol{\zeta}, \rho$):

$$R_N^\omega(\boldsymbol{\theta}) = 1/N \cdot \sum_{\mathbf{x} \in \mathcal{S}_N} \ell^\omega(\mathbf{x};\boldsymbol{\theta}); \quad R^\omega(\boldsymbol{\theta}) = \mathbb{E}_{\mathbf{x}}\big[\ell^\omega(\mathbf{x};\boldsymbol{\theta})\big]. \tag{6}$$

## 3.1 EXCESS SURROGATE RISK BOUND FOR LM IN TERMS OF NUMBER OF SEQUENCES

Given a potentially *infinite* function class $\Theta$ for student LMs, let $\hat{\boldsymbol{\theta}}$ and $\boldsymbol{\theta}^*$ be the minimizers of KD *training* objective in (6) and the *population* risk in (5), respectively:

$$\hat{\boldsymbol{\theta}} := \arg\min_{\boldsymbol{\theta} \in \Theta} R_N^\omega(\boldsymbol{\theta}); \quad \boldsymbol{\theta}^* = \arg\min_{\boldsymbol{\theta} \in \Theta} R(\boldsymbol{\theta}). \tag{7}$$

We want to compare the test performance (population risk) of $\hat{\boldsymbol{\theta}}$ with that of $\boldsymbol{\theta}^*$ – the optimal LM in $\Theta$. To achieve this, our analysis relies on the following assumption.

**Assumption 3.1.** The per-token log-loss with at most $T$-token long sequences for the function class $\Theta$ is bounded by a universal constant $M$, i.e., $\sup_{\boldsymbol{\theta} \in \Theta; \mathbf{x} \in \mathcal{V}^{\leq T-1}} \max_{v \in \mathcal{V}} |\log P_{\boldsymbol{\theta}}(v|\mathbf{x})| \leq M$.

Bounded loss assumption is common in the literature which holds, e.g., by clipping the loss by a large constant or by perturbing predictions with a small amount of noise (Lotfi et al., 2024a). We now state an informal version of our excess risk bound, with a formal statement and proof in Appendix B.1.

**Theorem 3.2** (Informal). *Let $\hat{\boldsymbol{\theta}}$ & $\boldsymbol{\theta}^*$ be as defined in (7) and $\delta \in (0,1)$. Define $f^{\boldsymbol{\theta}}(\mathbf{x}) := \ell^\omega(\mathbf{x};\boldsymbol{\theta})$, $\forall \mathbf{x} \in \mathcal{V}^T, \boldsymbol{\theta} \in \Theta$. Then, under Assumption 3.1, with probability at least $1-\delta$, we have*

$$R(\hat{\boldsymbol{\theta}}) - R(\boldsymbol{\theta}^*) \leq \frac{c_2 M}{N-1} \cdot \log(\mathcal{M}(N)/\delta) + \frac{4M\omega}{T} \sum_{t \in [T]} \mathbb{E}\Big[\mathsf{D}_{\mathrm{TV}}\Big(P_{\boldsymbol{\zeta}^\rho}(\cdot|\mathbf{x}_{\leq t-1}), \mathcal{D}(\cdot|\mathbf{x}_{\leq t-1})\Big)\Big]$$

$$+ c_1/\sqrt{N} \cdot \Big(\sqrt{U_N(f^{\hat{\boldsymbol{\theta}}}) \log(2\mathcal{M}(N)/\delta)} + \sqrt{U_N(f^{\boldsymbol{\theta}^*}) \log(4/\delta)}\Big),$$

*where $c_1$ & $c_2$ are universal constants; $U_N(f^{\boldsymbol{\theta}}) = \frac{1}{N(N-1)} \sum_{1 \leq i < j \leq N} \big(f^{\boldsymbol{\theta}}(\mathbf{x}^{(i)}) - f^{\boldsymbol{\theta}}(\mathbf{x}^{(j)})\big)^2$ is sample variance; $\mathsf{D}_{\mathrm{TV}}$ is TV distance; and $\mathcal{M}(N)$ depends on the growth function of $\{f^{\boldsymbol{\theta}} : \boldsymbol{\theta} \in \Theta\}$.*

## 3.2 EXCESS SURROGATE RISK BOUND FOR LM IN TERMS OF NUMBER OF TOKENS

For an LM $\boldsymbol{\theta} \in \Theta$ and a training sequence $\mathbf{x} = [x_1, x_2, \dots, x_T] \in \mathcal{V}^T$, define

$$\xi_t(\mathbf{x};\boldsymbol{\theta}) = \mathbb{E}_{\mathbf{z} \sim \mathcal{D}}\left[\ell^\omega(\mathbf{z};\boldsymbol{\theta})|\mathbf{z}_{\leq t-1} = \mathbf{x}_{\leq t-1}\right] - \mathbb{E}_{\mathbf{z} \sim \mathcal{D}}\left[\ell^\omega(\mathbf{z};\boldsymbol{\theta})|\mathbf{z}_{\leq t} = \mathbf{x}_{\leq t}\right], \quad t \in [T]. \tag{8}$$

Note that $\xi_t(\mathbf{x}; \boldsymbol{\theta})$ does not depend on $\mathbf{x}_{>t}$. For $t \in [T]$, $\xi_t(\mathbf{x}; \boldsymbol{\theta})$ measures the expected KD loss deviation for the student when we condition on the context up to $(t-1)$-th vs. $t$-th token, respectively, and sample the remaining tokens from $\mathcal{D}$. In general, the deviation can be large as changing a *single* token in the context can significantly alter LM's distribution. However, a well-behaved LM should be robust to such perturbations. Motivated by this, we introduce the following assumption.

**Assumption 3.3.** Given the data distribution $\mathcal{D}$ and a *finite* function class $\Theta$, $\exists \{C_t\}_{t \in [T]}$ and $\{V_t\}_{t \in [T]}$ such that the following holds for any $\mathbf{x} \in \text{Support}(\mathcal{D})$, $\boldsymbol{\theta} \in \Theta$, and $t \in [T]$:

$$|\xi_t(\mathbf{x}; \boldsymbol{\theta})| \le C_t \le C; \quad \mathbb{E}\left[\xi_t^2(\mathbf{x}; \boldsymbol{\theta})|\mathbf{x}_{\le t-1}\right] \le V_t. \tag{9}$$

We empirically validate that Assumption 3.3 is reasonable: Fig. 2 shows the distribution of $\widehat{\xi}_t(\mathbf{x}; \boldsymbol{\theta})$ – a plugin estimator of $\xi_t(\mathbf{x}; \boldsymbol{\theta})$ – for $\boldsymbol{\theta}$ denoting the BASE-LINE 2.8B LM in §5. Note that $\widehat{\xi}_t(\mathbf{x}; \boldsymbol{\theta})$ concentrates around $0$ as $t$ increases. Thus, the upper bounds $C_t$ and $V_t$ in (9) should also decrease with $t$, which, as discussed in Remark 3.5, would lead to favorable risk bounds for LM via our analysis. Appendix C provides the procedure for estimating $\xi_t(\mathbf{x}; \boldsymbol{\theta})$. We now state an excess risk bound for a student LM under Assumption 3.3; see Appendix B.2 for the proof.

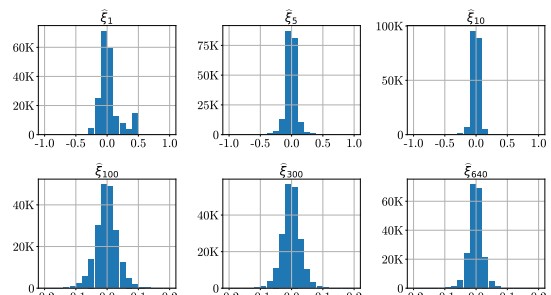

Figure 2: Histograms of $\widehat{\xi}_t$, $t \in \{1, 5, 10, 100, 300, 640\}$, for a 2.8B LM. In the bottom histograms, we reduce the x-axis range for better granularity. See Appendix C for details.

**Theorem 3.4.** *Let $\Theta$ be a finite function class. Under Assumptions 3.1 and 3.3, with probability at least $1 - \delta$, the following holds for the student LM $\hat{\boldsymbol{\theta}} \in \Theta$ obtained via KD:*

$$R(\hat{\boldsymbol{\theta}}) - R(\boldsymbol{\theta}^*) \le \frac{2C}{3N} \cdot \left(\log\left(\frac{2|\Theta|}{\delta}\right) + \log\left(\frac{2}{\delta}\right)\right) + \frac{4M\omega}{T} \cdot \sum_t \mathbb{E}\left[\text{D}_{\text{TV}}\left(P_{\boldsymbol{\zeta}^\rho}(\cdot|\mathbf{x}_{\le t-1}), \mathcal{D}(\cdot|\mathbf{x}_{\le t-1})\right)\right]$$

$$+ \sqrt{2\sum_t V_t/N} \cdot \left(\sqrt{\log\left(2|\Theta|/\delta\right)} + \sqrt{\log\left(2/\delta\right)}\right)$$

*Remark* 3.5 (Dependence of $C, \{V_t\}$ on $T$). Thm. 3.4 captures a fine-grained dependence on $C$ and $\{V_t\}$ from Assumption 3.3. For a robust LM, when $C$ is $\mathcal{O}(1/T)$ and $\{V_t\}$ are $\mathcal{O}(1/T^2)$ (note the scaling of $\ell^\omega$ by $T$), we get the tightest bound decaying with $NT$. For a non-robust LM, in the worst case, $C$ and $\{V_t\}$ can be as large as $\mathcal{O}(1)$. The bound is not tight anymore and we fall back to the bound in Thm. 3.2 that only decays with $N$ instead of $NT$.

*Remark* 3.6. Replacing Assumption 3.3 with a subgaussian assumption leads to an alternative bound that depends on the *mean* of variance proxies instead of the *maximal* quantities $C, V_t$ (Appendix B.3).

Recently, Lotfi et al. (2024b) obtained generalization bounds for LMs trained *without* KD in terms $NT$. However, specializing Thm. 3.4 to standard pre-training by setting $\omega$ to $0$ provides a bound that significantly differs from theirs both in terms of proof technique as well as its implications. Crucially, results in Lotfi et al. (2024b) only hold for the contexts seen during training whereas our bound holds for novel contexts generated from the data distribution during the test time. In a concurrent work, Zekri et al. (2024) provide generalization bounds for LMs by connecting auto-regressive models to finite state Markov chains. Notably, Zekri et al. (2024) do not explore KD for language modeling.

### 3.3 KD OUT-PERFORMING STANDARD PRE-TRAINING: A BIAS-VARIANCE TRADE-OFF

Empowered by our novel risk bounds, we now provide justification for why KD can outperform standard pre-training. Specifically, we base our analysis on Thm. 3.4, but similar conclusions also follow from Thm. 3.2. As per Thm. 3.4, three key quantities control the generalization of the student: (1) $\sum_t V_t$ which relates to loss variance; (2) $C$ which relates to extreme loss values; and (3) divergence between the teacher-provided distribution and the ground truth distribution: $\text{DIV}(\zeta, \omega) = \omega \cdot \sum_t \mathbb{E}\left[\text{D}_{\text{TV}}\left(P_{\boldsymbol{\zeta}, \rho}(\cdot|\mathbf{x}_{\le t-1}), \mathcal{D}(\cdot|\mathbf{x}_{\le t-1})\right)\right]$. Under Assumption 3.1, only $\sum_t V_t$ and $\text{DIV}(\zeta, \omega)$ are crucial in distinguishing KD and standard pre-training.

Since $\text{DIV}(\zeta, 0) = 0$, one may surmise that standard pre-training (i.e., $\omega = 0$) leads to a tighter bound. But as we detail in Appendix D due to the page limit, the variance term becomes smaller as we *increase* $\omega$. Thus, with a careful selection of $\omega$, the *variance reduction* via KD can offset the *bias*

---

**Algorithm 1** Small model aided large model training (SALT)

---

**Input:** Training data $\mathcal{S}_N = \{\mathbf{x}^{(i)}\}_{i \in [N]} \subset \mathcal{V}^T$, gradient-based optimization algorithm $\mathcal{A}$, SLM parameterized by $\boldsymbol{\zeta}$, distillation loss weight $\omega \in [0, 1]$, teacher temperature $\rho > 0$, batch size $B$, training step budget $n$, learning rate schedule $\{\eta_j\}_{j \in [n]}$, and $n_{\mathrm{KD}} \leq n$.

**Output:** Pre-trained LLM parameterized by $\hat{\boldsymbol{\theta}} \in \Theta$.

 1: Initialize $\boldsymbol{\theta}_0 \in \Theta$.
 2: **for** $j = 1, 2, \ldots, n_{\mathrm{KD}}$ **do**          // First stage of LLM pre-training via KD
 3:     Construct a new batch of $B$ training sequences $\mathcal{B}_j = \{\mathbf{x}^{(i)}\}_{i \in [B]} \subset \mathcal{S}_N$.
 4:     Update $\theta_{j+1}$ with step size $\eta_j$ via one step of $\mathcal{A}$ on $\mathcal{L}^{\mathrm{KD}}(\mathcal{B}_j) = \frac{1}{B} \sum_{\mathbf{x} \in \mathcal{B}_j} \ell^\omega(\mathbf{x}; \boldsymbol{\theta}_j)$.
 5: **end for**
 6: **for** $j = n_{\mathrm{KD}} + 1, n_{\mathrm{KD}} + 2, \ldots, n$ **do**      // Second stage: standard pre-training
 7:     Construct a new batch of $B$ training sequences $\mathcal{B}_j = \{\mathbf{x}^{(i)}\}_{i \in [B]} \subset \mathcal{S}_N$.
 8:     Update $\theta_{j+1}$ with step size $\eta_j$ via one step of $\mathcal{A}$ on $\mathcal{L}^{\mathrm{Std}}(\mathcal{B}_j) = \frac{1}{B} \sum_{\mathbf{x} \in \mathcal{B}_j} \ell(\mathbf{x}; \boldsymbol{\theta}_j)$.
 9: **end for**
10: $\hat{\boldsymbol{\theta}} \leftarrow \boldsymbol{\theta}_n$

---

$\mathrm{DIV}(\zeta, \omega)$. In particular, if the teacher closely approximates the ground truth distribution so that the bias $\mathrm{DIV}(\zeta, \omega)$ is small even for large $\omega$, then the variance reduction via KD becomes prominent, resulting in significantly smaller excess risk compared to standard pre-training.

**Performance gain from an SLM as teacher.** While small teacher LMs – the main interest of this work – also lead to variance reduction, they are typically not powerful enough to model the true distribution over the *entire* data domain very well. Thus, any effect of variance reduction via KD with such a teacher would be washed away by a large bias $\mathrm{DIV}(\zeta, \omega)$. This highlights the need for an *adaptive* form of KD from SLMs. Even SLMs with their limited capacity can approximate the true distribution well on certain regions of the data domain, which we call the "easy" regions. Thus, one can employ KD from SLMs on the easy regions to benefit from the variance reduction without incurring large bias and guarantee improved student LLM performance on these regions. For the remaining ("hard") regions, where the bias is large enough to overshadow the contributions of variance reduction, one should not perform KD from SLMs and utilize the standard pre-training loss.

## 4 SALT: S̲MALL MODEL A̲IDED L̲ARGE MODEL T̲RAINING

We now operationalize the key takeaway from §3 by proposing SALT – a simple yet effective two-stage pre-training method. SALT relies on the inherent tendency of a model to first focus on easier supervision before fitting more complex supervision during training (Kalimeris et al., 2019; Refinetti et al., 2023) to perform selective KD from a teacher SLM.

**Two-stage LLM pre-training via SALT.** Inspired by our analysis, we propose a two-stage pre-training method for LLMs in Alg. 1. The algorithm employs KD with SLM as a teacher in the first stage comprising $n_{\mathrm{KD}}$ training steps, and transitions to standard pre-training *without* KD in the second stage. We are interested in the selective transfer of predictive distribution from teacher SLM to student LLM in those regions where SLM performs well by capturing true distribution. By design, KD aims to align predictive distributions of the teacher and student. On the regions where SLM performs well, we expect it to exhibit reasonably confident predictive distribution that should align with ground truth next-token (Gupta et al., 2024), thereby constituting an easier supervision signal for the LLM. In contrast, on hard instances where SLM's predictive distribution is not confident enough or does not align well with the ground truth next-token, learning will be delayed to the later phase of the training (Kalimeris et al., 2019; Refinetti et al., 2023). Thus, SALT relies on the tendency of neural networks to focus on easier instances early during the training to perform desirable knowledge transfer from SLM in the first stage. Once the student LLM is sufficiently aligned with teacher SLM on easier regions, it starts utilizing its model capacity to further align with SLM on more complex regions where high divergence between SLM and ground truth distribution can become detrimental to the LLM's performance. Switching to standard pre-training in the second stage prevents this undesirable over-alignment. We empirically verify the above intuition behind SALT in §5.4.

SALT$_{\mathrm{DS}}$: SALT **with d̲ata s̲election.** We now endow SALT (cf. Alg. 1) with explicit selection of examples where we want to transfer teacher SLM's predictive distribution on, with SLM itself enabling the selection. In particular, we want to select the most informative (or challenging) examples

among the ones that SLM performs well on. Towards this, given the SLM $\boldsymbol{\zeta}$ and a positive integer $k$, we assign a score $S_{\boldsymbol{\zeta},k}(\mathbf{x})$ to training sequence $\mathbf{x}$, with a higher score indicating a higher likelihood to be selected for training. More specifically, we compute the per-token cross-entropy losses of SLM on $\mathbf{x}$ and aggregate them into a sequence-level score. This encourages selecting more challenging examples. However, in the spirit of selecting examples that are still *learnable*, we remove all losses where the ground-truth token is not in top-$k$ outputs of the SLM before aggregating:

$$S_{\boldsymbol{\zeta},k}(\mathbf{x}) = \text{median}\big(\big\{ -\mathbb{1}\{x_t \in \text{argtop}_k(P_{\boldsymbol{\zeta}}(\cdot|\mathbf{x}_{<t}))\} \cdot \log P_{\boldsymbol{\zeta}}(x_t|\mathbf{x}_{<t}) : t \in [T]\big\}\big), \quad (10)$$

where $\text{argtop}_k(P_{\boldsymbol{\zeta}}(\cdot|\mathbf{x}_{<i}))$ denotes the top-$k$ scoring tokens at position $i$ as per SLM. Given a scored pool of sequences, we select the top-$m$ scoring sequences so that $m$ suffices to complete the first stage of Alg. 1. The reader may note that if the SLM has been trained with the same dataset that it is scoring, then the computed sequence score may be biased. To circumvent that, we use an "early checkpoint" of the SLM after $n_0$ steps, i.e., $\boldsymbol{\zeta}_{n_0}$, which has trained on a small number of examples from the overall training set with $n_0 B \ll N$. We then sample from the remainder of the training examples using score $S_{\boldsymbol{\zeta}_{n_0},k}(\cdot)$. Although $\boldsymbol{\zeta}_{n_0}$ may have lower quality, it is only the relative ordering of examples that is important when computing a score, rather than the absolute score.

## 5 EXPERIMENTS

We now showcase the potential of SALT for improving LLM pre-training, by realizing both better quality and improved training efficiency. In our study, we compare with a natural baseline (denoted BASELINE) where one pre-trains an LLM in a standalone manner with a self-supervised objective over a pre-training set. We also compare SALT with RKD (standing for *reverse KD*) where we perform KD with teacher SLM *throughout* pre-training.

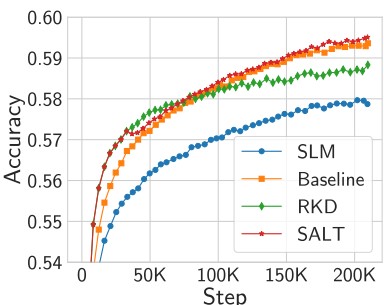

Figure 3: Next-token prediction accuracy on the Pile training set for 2.8B LMs trained with the help of a 1.5B SLM.

The key takeaways from our study are: **(1)** SALT & SALT$_{\text{DS}}$ attain BASELINE performance with less than 70% of training steps, and significantly outperform BASELINE with the same number of training steps (§ 5.2). Additionally, SALT leverages an improved teacher SLM to further enhance LLM performance (Appendix K). **(2)** SALT & SALT$_{\text{DS}}$ pre-trained LMs outperform BASELINE after SFT (§ 5.3). **(3)** SALT is robust to key design choices. *Step* transition from KD to standard training in Alg. 1 is a good design choice as it outperforms other natural alternatives. Moreover, SALT continues to outperform BASELINE for a wide range of values for $n_{\text{KD}}$ (Appendix K).

### 5.1 EXPERIMENTAL SETUP

**Model architectures, pre-training data, and hyperparameters.** We pre-train decoder-only Transformer-based LMs with 2.8B and 8.6B parameters. For the small teacher model (SLM), we primarily utilize 1.5B and 2.8B parameter LMs. We pre-train all LMs on the Pile dataset (Gao et al., 2020) for 545 billion tokens via UL2 objective (Tay et al., 2023) with a mixture of causal LM, prefix LM, and span corruption tasks. For SALT$_{\text{DS}}$, we score examples with SLM trained for $n_0 = 26\text{K}$ steps and set $k = 10$ in (10). Based on our hyperparameter search at small scale (see Appendix G), we set the distillation weight $\omega = 0.667$ and teacher temperature $\rho = 0.25$ in all of our experiments. As for the remaining hyperparameters such as peak LR and LR schedule, we do not optimize those for our proposed method and use the same values for all of our experiments involving different model sizes. This along with the consistent quality & efficiency improvements realized via SALT in § 5.2 shows the robustness of SALT to various hyperparameter choices. See Appendix E for further details.

**Few-shot evaluation tasks.** As per the literature (see, e.g., Anil et al., 2023; Touvron et al., 2023), we perform few-shot evaluation of pre-trained LMs on a wide range of benchmarks, which can be categorized into: (1) world knowledge, (2) reading comprehension, (3) commonsense reasoning, (4) natural language generation (NLG), and (5) SuperGLUE. We also consider LAMBADA (Paperno et al., 2016) and MBPP (Austin et al., 2021) which are Cloze and code generation tasks, respectively. See Appendix F for the full list of benchmarks and the corresponding evaluation metrics. We focus on English benchmarks as the Pile data is mostly English.

**Fine-tuning tasks.** We focus on (arithmetic) reasoning, summarization, and NLI tasks during SFT, covering GSM8K (Nie et al., 2020), XSum (Narayan et al., 2018), CNN/DailyMail (Nallapati et al., 2016), and ANLI-R1/R2/R3 (Nie et al., 2020) benchmarks.

## 5.2 RESULTS: SALT IMPROVES BOTH QUALITY AND EFFICIENCY

Table 1: **Accuracy and log-perplexity** on a held-out set of Pile while training 2.8B LM by using a 1.5B SLM. We evaluate 2.8B LMs at an early and the final step. At the final step, SALT & SALT$_{DS}$ outperform BASELINE in terms of both next-token prediction accuracy and log-perplexity.

| Model | Evaluation stage (steps) | Accuracy ($\uparrow$) | Log ($\downarrow$) perplexity |
|---|---|---|---|
| SLM | Final (208K) | 57.70 | 1.951 |
| BASELINE | Early (36K) | 56.68 | 2.011 |
| RKD | | 57.21 | 2.160 |
| SALT | | 57.21 | 2.160 |
| SALT$_{DS}$ | | 56.47 | 2.188 |
| BASELINE | Final (208K) | 58.99 | 1.868 |
| RKD | | 58.46 | 2.071 |
| SALT | | 59.10 | 1.863 |
| SALT$_{DS}$ | | **59.17** | **1.857** |

As reflected in the *next-token prediction accuracy* in Fig. 3, KD from the seemingly weaker 1.5B SLM does improve 2.8B larger LM training in the beginning compared to BASELINE (cf. Fig. 8 in Appendix L for the *log-perplexity* plot). However, continuing KD from the weaker teacher eventually become detrimental. As evident in Fig. 3 and Tab. 1, RKD significantly underperforms BASELINE on both training and validation set. In contrast, SALT leverages KD from SLM only during first $n_{KD}$ training steps (cf. Alg. 1).

**Quality improvements via SALT.** Unlike RKD, SALT (with $n_{KD}$=36K steps) yields a pre-trained LLM that improves upon BASELINE on both training set (Fig. 3) and held-out validation set (Tab. 1). Tab. 2 presents domain-wise few-shot performance of BASELINE, RKD, SALT, and SALT$_{DS}$ (see Tab. 12 in Appendix H for per-task performance). Both SALT & SALT$_{DS}$ consistently outperform BASELINE (and RKD) @100% training steps, i.e., 208K steps. Besides improving overall average, SALT and SALT$_{DS}$ outperform BASELINE in 5 out of 7 domains and 6 out of 7 domains, respectively. Similar gains are evident for training an 8.6B LM with the aid of a 2.8B SLM from Tab. 13 & 14 in Appendix I. This establishes the utility of SALT approaches in successfully leveraging SLMs to boost the quality of LLMs.

*Remark* 5.1. One can assess the significance of the performance improvements via SALT by contextualizing the improvements relative to the gains from model scaling. From Tab. 2, @100% steps, SALT has a gain of $47.94 - 47.32 = 0.62$ in terms of overall average over BASELINE, which is $\sim$13% of the gain from nearly doubling the model size, from 42.56 for 1.5B to 47.32 for 2.8B. Similarly, from Tab. 13 in Appendix I, SALT realizes a gain of $52.96 - 51.73 = 1.23$, which is $\sim$28% of the gain of $51.73 - 47.32 = 4.41$ realized by increasing model size $\sim$3x from 2.8B to 8.6B.

Table 2: **Domain-wise few-shot performance** of 2.8B pre-trained LMs. Using a 1.5B SLM as a teacher in the KD phase, SALT and SALT$_{DS}$ already outperform BASELINE in terms of average few-shot performance at 70% of the training step budget, thereby improving both training efficiency *and* model quality. RKD (i.e., naïvely distilling from the 1.5B SLM throughout pre-training) performs much worse than BASELINE. The (second-)best results for each domain are (underlined) **boldfaced**.

| Domain (# Tasks) | SLM | BASELINE @100% steps | RKD @100% steps | SALT @70% steps | SALT @100% steps | SALT$_{DS}$ @70% steps | SALT$_{DS}$ @100% steps |
|---|---|---|---|---|---|---|---|
| **World Knowledge (4)** | 15.90 | 22.19 | 18.69 | 21.59 | **22.70** | 20.64 | 21.72 |
| **Reading Comprehension (4)** | 46.30 | 53.00 | 51.00 | 53.55 | 54.55 | 54.35 | **54.93** |
| **Commonsense Reasoning (7)** | 57.76 | 61.99 | 58.30 | 61.27 | 61.67 | 62.00 | **62.10** |
| **LAMBADA (1)** | 26.90 | 36.20 | 31.10 | 50.70 | 48.30 | 48.00 | **53.00** |
| **SuperGLUE (8)** | 61.59 | 65.53 | 62.91 | **66.30** | 65.28 | 65.99 | 65.58 |
| **NLG (3)** | 3.13 | 4.60 | 3.40 | 4.63 | 4.73 | 4.80 | **4.83** |
| **MBPP (1)** | 9.60 | 16.20 | 11.40 | 15.60 | 17.00 | 16.60 | **17.80** |
| **Average (28)** | 42.56 | 47.32 | 44.39 | 47.86 | 47.94 | 47.89 | **48.26** |

**Training efficiency via SALT.** As per Tab. 2, SALT *surpasses* BASELINE at 146K steps on average performance, suggesting a savings of 30% training steps. While $n_{KD} = 36K$ of those 146K steps involve KD (which is computationally costlier than standard training), as shown in Appendix J, we still realize efficiency gains via SALT as our teacher is an SLM. With our implementation based on rematerialization (Chen et al., 2016), SALT realizes a wall clock saving of $\sim 25\%$ for training a 2.8B LM, and $\sim 28\%$ for training an 8.6B LM with the aid of a 2.8B LM (see Appendix J). In summary, SALT results in improved training efficiency while still outperforming the (overtrained) BASELINE.

*Remark* 5.2 (Training cost of SLM). As discussed in the introduction, suitable SLMs for SALT are often available without incurring additional training cost. Even if one has to train an SLM specifically for SALT, its training cost can be amortized across multiple LLM training runs as the *same* SLM can

Table 4: **Supervised fine-tuning (SFT) results** for 2.8B LMs. Performance of pre-trained LMs on downstream tasks after SFT. For each benchmark, pre-trained 2.8B models are fine-tuned on the corresponding train split and evaluated on the validation split (test split in case of GSM8K). *Acc*, *Rg1*, *Rg2*, and *RgL* represent the *Accuracy*, *Rouge-1*, *Rouge-2*, and *Rouge-Lsum* metrics, respectively.

| | GSM8K | XSum | | | CNN/DailyMail | | | ANLI-R1 | ANLI-R2 | ANLI-R3 |
|---|---|---|---|---|---|---|---|---|---|---|
| | *Acc* | *Rg1* | *Rg2* | *RgL* | *Rg1* | *Rg2* | *RgL* | *Acc* | *Acc* | *Acc* |
| Baseline | 31.84 | 43.39 | 21.09 | 35.91 | 42.84 | 20.43 | 40.38 | 63.70 | 56.90 | 57.83 |
| SALT | 34.87 | 43.45 | 21.21 | 36.04 | 43.19 | 20.65 | 40.74 | 67.00 | **57.80** | **59.67** |
| SALT_{DS} | **35.25** | **43.77** | **21.44** | **36.24** | **43.41** | **20.87** | **40.95** | **67.30** | 57.70 | 59.58 |

be used to train *multiple* larger LMs, potentially of varying sizes; e.g., we used the same 1.5B LM to train 2.8B LM (cf. Tab. 2) and 8.6B LM (cf. Tab. 16 in Appendix I.3).

**Ablation on student-teacher size ratios.** We now explore the effectiveness of SALT as the gap between the sizes of the teacher and student increases. Table 3 shows the average few-shot performance for 8.6B student LM trained via SALT with 0.5B, 1.5B, and 2.8B teacher LMs (see Appendix I for full results). As expected, the performance of SALT improves as the gap between student and teacher sizes decreases. SALT leads to both quality and efficiency gains with $3\times$ and $5.5\times$ smaller teacher. At the *extreme* end, with $\sim17\times$ smaller 0.5B teacher, the average score does not improve with SALT, but notably,

Table 3: **Average few-shot performance** for 8.6B student LM trained via SALT with teacher models of different sizes.

| Method | Teacher size (student-teacher size ratio) | @70% steps | @100% steps |
|---|---|---|---|
| Baseline | – | 51.21 | 51.73 |
| SALT | 0.5B (17.2) | 50.68 | 51.62 |
| | 1.5B (5.5) | 51.82 | 52.80 |
| | 2.8B (3.0) | 52.24 | 52.96 |

the performance improves on 4 out of 7 domains (cf. Tab. 18, Appendix I). This hints at the utility of multiple small teachers – each an expert in its own domain.

## 5.3 IMPROVED POST SFT PERFORMANCE REALIZED VIA SALT

Tab. 2 and Tab. 13 (in Appendix I) already show the utility of SALT in out-performing Baseline. However, all the pre-trained LMs (including Baseline) exhibit relatively poor few-shot performance on certain benchmarks, e.g., NLG or summarization tasks (Tab. 2), MATH (Hendrycks et al., 2021), and ANLI (Nie et al., 2020). To establish the value of SALT for these domains, we employ SFT on downstream tasks covering arithmetic reasoning, NLG, and NLI domains.

For each task, we perform SFT on the pre-trained LMs obtained via Baseline, SALT, and SALT_{DS}. During SFT, we train for 10K steps by using Adafactor (Shazeer & Stern, 2018) and cosine learning rate schedule (Loshchilov & Hutter, 2017) with a peak learning rate of $10^{-4}$ and linear warm-up for 200 steps. These hyperparameters were optimized for Baseline, without further tuning for SALT and SALT_{DS}. Finally, we employ greedy decoding during the evaluation of the fine-tuned LMs. Post-SFT results for 2.8B LMs in Tab. 4 show that SALT consistently outperforms Baseline on all benchmarks, and is often the best-performing method. (See Tab. 15 in Appendix I for 8.6B results.) Thus, SALT (*with* or *without* data selection) enables significant improvements for several *difficult* downstream domains.

## 5.4 SLM ENABLES FAST LEARNING ON EASY EXAMPLES

Recall that SALT aims to improve quality and efficiency of LLM pre-training by quickly transferring the predictive distribution of an SLM to the LLM via KD, focusing on the "easy" regions of the data distribution where the SLM performs well. Subsequently, SALT falls back on ground truth next-token-based supervision to refine LLM's performance on the 'hard' regions where the SLM fares poorly. Here, we empirically demonstrate that this key intuition behind SALT is indeed borne out in practice. Furthermore, to highlight the importance of leveraging teacher-provided supervision on the "easy" regions, we consider another baseline, namely BaselineEZ, that *explicitly* trains on only the "easy" regions of

Table 5: **Per-bucket few-shot evaluation on XLSum-EN** (Rouge-2). Gray, green, and red mark results on-par, better than, and worse than Baseline, respectively.

| | Evaluation stage | Easy | Medium | Hard |
|---|---|---|---|---|
| SLM | Final (208K) | 8.04 | 0.43 | 0.00 |
| Baseline | Early (36K steps) | 6.15 | 1.61 | 0.71 |
| BaselineEZ | | 6.43 | 1.54 | 0.69 |
| RKD | | 6.76 | 1.40 | 0.58 |
| SALT | | 6.76 | 1.40 | 0.58 |
| Baseline | Final (208K steps) | 8.80 | 2.52 | 0.97 |
| BaselineEZ | | 9.37 | 2.51 | 0.91 |
| RKD | | 7.87 | 1.68 | 0.74 |
| SALT | | 9.68 | 2.67 | 0.99 |

the training data during the first stage (i.e., first $n_{KD}$ steps) *without* performing distillation (see Appendix M for a formal description of BASELINEEZ).

Focusing on various few-shot benchmarks, we partition instances in each benchmark into "easy", "medium", and "hard" buckets based on the teacher SLM's performance (see Appendix M for details). We then evaluate BASELINE, BASELINEEZ, SALT, and RKD pre-trained LMs on these buckets after $n_{KD} = 36K$ training steps when the KD phase of SALT ends as well as at the end of the pre-training, i.e., after 208K steps. Tab. 5 presents these results on XLSum-EN (see Appendix M for results on other benchmarks), which validate: (1) KD from SLM quickly enables LLM to perform well on "easy" instances; and (2) standard pre-training after KD phase ending at $n_{KD}$-th step helps LLM performance on 'hard' instances the most. In contrast, BASELINEEZ does not exhibit the same trend, and focusing only on the easy instances during the first stage without KD seems to hurt the final performance on non-easy slices. In fact, Tab. 27 in Appendix M shows that BASELINEEZ is even worse than BASELINE in terms of general few-shot performance.

## 6 RELATED WORK

**Aiding large model training with small models.** Small models often help inform hyper-parameter selection for large model training (Yang et al., 2021). Progressive or stage-wise training (Gong et al., 2019; Reddi et al., 2023; Yao et al., 2023; Li et al., 2023; Du et al., 2024; Agarwal et al., 2024a) develops a large model in stages, with model parameters at a stage getting initialized based on the parameters of a smaller model from the previous stage. Chen et al. (2022); Trockman & Kolter (2023); Wang et al. (2023b;a); Samragh et al. (2024) initialize large model based on a smaller model *without* employing progressive training. Most such works crucially rely on *architectural overlaps* between the small and large models. That said, such approaches are complementary to SALT, and can be explored together to boost LLM performance, e.g., by progressively growing an LLM during pre-training while utilizing an SLM as a teacher during the early stages of the progressive growth. Closer to our work, Yuan et al. (2020); Xie et al. (2020) consider KD from a weaker model, *albeit* for image classification. In the work that is closest to our proposal, Qin et al. (2022) distill an LM from a smaller LM during the early phase of pre-training. We show the utility of such an approach with larger models, larger datasets, and a wider range of evaluation benchmarks. We also provide a statistical framework to rigorously justify the value of a weaker teacher during pre-training. Other recent efforts on using SLMs to boost LLMs mainly consider fine-tuning (Yang et al., 2024; Mitchell et al., 2024) or alignment (Burns et al., 2023), while we focus on compute and data intensive pre-training.

**Data Selection.** Ankner et al. (2024) perform sequence selection by aggregating per-token log-perplexities of a reference model by taking the mean, whereas we take the median and crucially exclude noisy (unlearnable) tokens from the aggregation. Mindermann et al. (2022) select sequences and Lin et al. (2024) select tokens based on *excess* training loss over a reference model. Unlike our *offline* data selection approach, these works select data from training batches on the fly. While Gu et al. (2025) also perform offline data selection, unlike our work, they rely on two models for data selection – a smaller reference model and a *larger* teacher model. SALT$_{DS}$ can further benefit from diversity encouraging methods (see, e.g., Abbas et al., 2023; Tirumala et al., 2024). Please refer to Albalak et al. (2024) for a comprehensive survey of data selection techniques for LMs.

Due to page limit, we defer the discussion of literature on theoretical results for KD to Appendix A. Existing works *do not* provide a statistical treatment of KD in a *sequence* learning setting such as language modeling. We provide the first generalization bounds for KD in such a setting.

## 7 CONCLUSION

We explored the utility of SLMs for improving the LLM pre-training. Towards this, we introduced a statistical framework for KD in language modeling, which guided the design of SALT to selectively transfer predictive distribution from an SLM to LLM. We further enhanced SALT by performing data selection via SLMs to effectively transfer knowledge from SLMs to LLMs. SALT significantly reduces the pre-training time for LLMs while ensuring good overall quality as measured by the LLM's few-shot performance as well as downstream performance after fine-tuning.

An interesting direction for future work is efficiently improving LLMs by leveraging multiple SLMs, each an expert in its own domain. Building on our positive results with SALT$_{DS}$, exploring and extending data selection approaches tailored to SALT$_{DS}$ is also a promising avenue to enhance LLM quality. Another fruitful direction is to study SALT in conjunction with progressive training to leverage their complementary nature for LLM pre-training.

REPRODUCIBILITY STATEMENT

Our submission provides the details required to reproduce our pre-training results, including model architectures, pre-training data, hyperparameters, and training setup in § 5.1 and Appendix E. Appendix F describes the few-shot evaluation benchmarks considered in our submission along with the corresponding evaluation metrics. § 5.3 discusses the hyperparameters for the SFT experiments. As for our theoretical results, § 3 clearly states various assumptions and Appendix B provides detailed proofs of the claims. Appendix C details our methodology to verify the feasibility of Assumption 3.3. Appendix M provides details to reproduce the per-bucket analysis in § 5.4.

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

# Appendix

## A   LANGUAGE MODELING AND KNOWLEDGE DISTILLATION

### A.1   LANGUAGE MODELING VIA TRANSFORMER-BASED MODELS

Here, we briefly discuss how Transformers are typically employed for language modeling in modern systems. Given a context $\mathbf{x}_{\leq t} = [x_1, \ldots, x_t] \in \mathcal{V}^t$, a Transformer-based LM first produces a sequence of $d$-dimensional token embeddings $E(\mathbf{x}_{\leq t}) = [\mathbf{e}_{x_1}^\top, \mathbf{e}_{x_2}^\top, \ldots, \mathbf{e}_{x_t}^\top] \in \mathbb{R}^{d \times t}$, where $\mathbf{e}_v \in \mathbb{R}^d$ denotes the token embedding for $v \in \mathcal{V}$. A Transformer network $f_{\boldsymbol{\psi}}$ then processes $E(\mathbf{x}_{\leq t})$ to produce a target embedding $f_{\boldsymbol{\psi}}\big(E(\mathbf{x}_{\leq t})\big) \in \mathbb{R}^d$, which is multiplied by $W \in \mathbb{R}^{V \times d}$, namely a classification layer, to obtain a logit vector $\mathbf{u}_{\mathbf{x}_{\leq t}} = (u_{\mathbf{x}_{\leq t}}(v))_{v \in \mathcal{V}} := W f_{\boldsymbol{\psi}}\big(E(\mathbf{x}_{\leq t})\big) \in \mathbb{R}^V$. Accordingly, we have $\boldsymbol{\theta} = \{E, \boldsymbol{\psi}, W\}$ as the parameters of the LM. Applying softmax operation on the logit vector produces the probability that the LM assigns to each token in $\mathcal{V}$ as the possible continuation (also known as next token) to the context $\mathbf{x}_{\leq t}$:

$$P_{\boldsymbol{\theta}}(v|\mathbf{x}_{\leq t}) = \frac{\exp(u_{\mathbf{x}_{\leq t}}(v)/\tau)}{\sum_{v' \in \mathcal{V}} \exp(u_{\mathbf{x}_{\leq t}}(v')/\tau)}, \qquad \forall v \in \mathcal{V}. \tag{11}$$

Here, $\tau$ denotes the (inverse) temperature associated with the softmax operation. Unless stated otherwise, we assume that $\tau = 1$.

### A.2   OTHER COMMON VARIANTS OF KNOWLEDGE DISTILLATION FOR LM

**Top-$k$ token-level KD.** Instead of aligning the teacher and student's full per-token predictive distributions, one could only match these distribution on $\mathcal{T} \subset \mathcal{V}$, e.g., $k \ll V$ elements of $\mathcal{V}$ that receive the highest scores from the teacher:

$$\ell^{\boldsymbol{\zeta}}(\mathbf{x}; \boldsymbol{\theta}) = -\sum_{t=1}^{T} \Big( \sum_{v \in \mathcal{T}} P_{\boldsymbol{\zeta}}^{\mathcal{T}}(v|\mathbf{x}_{\leq t-1}) \cdot \log P_{\boldsymbol{\theta}}^{\mathcal{T}}(v|\mathbf{x}_{\leq t-1}) \Big), \tag{12}$$

where $P^{\mathcal{T}}$ denotes the restriction of $P$ (defined over $\mathcal{V}$) to $\mathcal{T}$:

$$P^{\mathcal{T}}(v) = \begin{cases} \frac{P(v)}{\sum_{v' \in \mathcal{T}} P(v')} & \text{if } v \in \mathcal{T}, \\ 0 & \text{otherwise.} \end{cases} \tag{13}$$

**Sequence-level KD.** Unlike token-level KD, sequence-level KD aims to align teacher and student's distributions on all sequences up to sequence length $T$. In particular, the sequence-level KD loss takes the form:

$$\ell^{\boldsymbol{\zeta}}(\mathbf{x}; \boldsymbol{\theta}) = -\sum_{\tilde{\mathbf{x}} \in \mathcal{V}^{\leq n}} P_{\boldsymbol{\zeta}}(\tilde{\mathbf{x}}) \cdot \log P_{\boldsymbol{\theta}}(\tilde{\mathbf{x}}) \tag{14}$$

In practice, it's natural to focus on a subset of all candidate target sequences $\mathcal{U} \subset \mathcal{V}^{\leq T}$:

$$\ell^{\boldsymbol{\zeta}}(\mathbf{x}; \boldsymbol{\theta}) = -\sum_{\tilde{\mathbf{x}} \in \mathcal{U}} P_{\boldsymbol{\zeta}}(\tilde{\mathbf{x}}) \cdot \log P_{\boldsymbol{\theta}}(\tilde{\mathbf{x}})$$

A common choice for $\mathcal{U}$ is the set of say $k$ most likely sequences under the teacher's distribution $P_{\boldsymbol{\zeta}}$.

### A.3   RECENT LITERATURE ON KNOWLEDGE DISTILLATION (KD) FOR LANGUAGE MODELING

A large body of literature focuses on utilizing KD (Bucilă et al., 2006; Hinton et al., 2015) as a core technique to improve LMs (Kim & Rush, 2016; Gou et al., 2021; Xu et al., 2024). For instance, Sanh et al. (2019); Turc et al. (2019); Wang et al. (2020); Sun et al. (2019); Jiao et al. (2019) relied on KD to compress BERT-style LMs during pre-training, fine-tuning, or both. More recently, KD has been primarily employed in the instruction-tuning or fine-tuning phase where a general purpose LM is adapted to a specific collection of tasks (Xu et al., 2024). Black-box KD methods for LM only assume

access to training sequences sampled from a teacher LM (Taori et al., 2023; Fu et al., 2023; Peng et al., 2023). With access to token-level distributions from teacher LM, *token-level distillation* from teacher LM to student LM is possible (Kim & Rush, 2016). In contrast, *sequence-level distillation* involves sampling training sequences from the teacher LM, the student LM, or both before aligning teacher and student's predictive distribution on such sequences (Kim & Rush, 2016; Agarwal et al., 2024b; Gu et al., 2024; Wen et al., 2023).

### A.4 PRIOR WORK ON THEORETICAL UNDERSTANDING OF KD

Multiple works focus on explaining the widespread success of KD. Phuong & Lampert (2019) connect the effectiveness of KD to data geometry and optimization bias. Menon et al. (2021); Ren et al. (2022b) show that KD leads to reduced variance of training objective. Dao et al. (2021b) study KD from the lens of semiparameteric inference. Mobahi et al. (2020) show that KD enhances regularization. Allen-Zhu & Li (2023) explain the utility of KD via better feature learning. Harutyunyan et al. (2023) attribute the success of KD to the reduced supervision complexity. More recently, Safaryan et al. (2023) argue that KD can act as a form of partial variance reduction to improve convergence. Notably, existing literature *does not* provide a statistical treatment of KD in a *sequence* learning setting such as language modeling, and we provide the first generalization bounds for KD in such a setting. Similar to our work, Xu et al. (2020); Nagarajan et al. (2023) also explore the utility of KD only during an early-phase of student training, albeit not in a language modeling setting.

## B PROOFS DEFERRED FROM §3

### B.1 PROOF OF THEOREM 3.2

Before stating the formal version of Theorem 3.2 and its proof, let us recall the necessary notation. Given a function class for student LMs $\Theta$, $\hat{\boldsymbol{\theta}}$ denotes the LM obtained by minimizing the training objective for KD in (6), i.e.,

$$\hat{\boldsymbol{\theta}} := \arg\min_{\boldsymbol{\theta} \in \Theta} R_N^{\omega}(\boldsymbol{\theta}). \tag{15}$$

Further, $\boldsymbol{\theta}^*$ represents the optimal or best performing LM in $\Theta$, i.e.,

$$\boldsymbol{\theta}^* = \arg\min_{\boldsymbol{\theta} \in \Theta} R(\boldsymbol{\theta}). \tag{16}$$

Finally, recall our assumption regarding the bounded loss values.

**Assumption B.1.** Given a function class $\Theta$ for (student) LM, the per-token log-loss with at most $T$-token long sequences for the underlying function class $\Theta$ is bounded by a universal constant $M$, i.e.,

$$\sup_{\boldsymbol{\theta} \in \Theta; \mathbf{x} \in \mathcal{V}^{\leq T-1}} \max_{v \in \mathcal{V}} |\log P_{\boldsymbol{\theta}}(v|\mathbf{x})| \leq M. \tag{17}$$

Towards establishing Theorem 3.2, we first state the following intermediate result.

**Proposition B.2.** *Let $\hat{\boldsymbol{\theta}}$ and $\boldsymbol{\theta}^*$ be as defined in* (15) *and* (16)*, respectively. Then, under Assumption B.1, the excess surrogate risk for $\hat{\boldsymbol{\theta}}$ satisfies the following.*

$$R(\hat{\boldsymbol{\theta}}) - R(\boldsymbol{\theta}^*) \leq \frac{4M\omega}{T} \cdot \sum_{t=1}^{T} \mathbb{E}_{\mathbf{x}_{\leq t-1} \sim \mathcal{D}} D_{\mathrm{TV}}\Big( P_{\boldsymbol{\zeta}^\rho}(\cdot|\mathbf{x}_{\leq t-1}), \mathcal{D}(\cdot|\mathbf{x}_{\leq t-1}) \Big)$$
$$+ \Big( R^{\omega}(\hat{\boldsymbol{\theta}}) - R_N^{\omega}(\hat{\boldsymbol{\theta}}) \Big) + (R_N^{\omega}(\boldsymbol{\theta}^*) - R^{\omega}(\boldsymbol{\theta}^*)), \tag{18}$$

*where $D_{\mathrm{TV}}(\cdot, \cdot)$ denotes the total-variation distance between two probability distributions.*

*Proof of Proposition B.2.* For convenience, recall that

$$R^\omega(\boldsymbol{\theta}) = \mathbb{E}_\mathbf{x}\Big[(1-\omega)\cdot \ell(\mathbf{x};\boldsymbol{\theta}) + \omega\cdot \ell^{\boldsymbol{\zeta}^\rho}(\mathbf{x};\boldsymbol{\theta})\Big]$$

$$= \mathbb{E}_{\mathbf{x}\sim\mathcal{D}}\left[\frac{1}{T}\sum_{t=1}^{T}\mathsf{CE}\big(P_{\boldsymbol{\zeta}^\rho}^{(x_t,\omega)}, P_{\boldsymbol{\theta}}(\cdot|\mathbf{x}_{\leq t-1})\big)\right]$$

$$= \frac{1}{T}\sum_{t=1}^{T}\mathbb{E}_{\mathbf{x}_{\leq t-1}\sim\mathcal{D}}\Big[\mathsf{CE}\big(\underbrace{(1-\omega)\cdot\mathcal{D}(\cdot|\mathbf{x}_{\leq t-1}) + \omega\cdot P_{\boldsymbol{\zeta}^\rho}(\cdot|\mathbf{x}_{\leq t-1})}_{P_{\boldsymbol{\zeta}^\rho}^{(\mathcal{D},\omega)}(\cdot|\mathbf{x}_{\leq t-1})}, P_{\boldsymbol{\theta}}(\cdot|\mathbf{x}_{\leq t-1}))\big]$$

$$= \frac{1}{T}\sum_{t=1}^{T}\mathbb{E}_{\mathbf{x}_{\leq t-1}\sim\mathcal{D}}\big[\mathsf{CE}\big(P_{\boldsymbol{\zeta}^\rho}^{(\mathcal{D},\omega)}(\cdot|\mathbf{x}_{\leq t-1}), P_{\boldsymbol{\theta}}(\cdot|\mathbf{x}_{\leq t-1}))\big].$$

Note that we have

$$R(\hat{\boldsymbol{\theta}}) - R(\boldsymbol{\theta}^*)$$

$$\overset{(i)}{=} R(\hat{\boldsymbol{\theta}}) - R(\boldsymbol{\theta}^*) - \Big(R^\omega(\hat{\boldsymbol{\theta}}) - R^\omega(\boldsymbol{\theta}^*)\Big) + \Big(R^\omega(\hat{\boldsymbol{\theta}}) - R^\omega(\boldsymbol{\theta}^*)\Big)$$

$$\overset{(ii)}{=} \frac{1}{T}\sum_{t=0}^{T-1}\mathbb{E}_{\mathbf{x}_{1:t}\sim\mathcal{D}}\Big[\sum_{v\in\mathcal{V}}\Big(P_{\boldsymbol{\zeta}^\rho}^{(\mathcal{D},\omega)}(v|\mathbf{x}_{1:t}) - \mathcal{D}(v|\mathbf{x}_{1:t})\Big)\cdot\Big(\log P_{\hat{\boldsymbol{\theta}}}(v|\mathbf{x}_{1:t}) - \log P_{\boldsymbol{\theta}^*}(v|\mathbf{x}_{1:t})\Big)\Big] +$$

$$R^\omega(\hat{\boldsymbol{\theta}}) - R^\omega(\boldsymbol{\theta}^*)$$

$$\overset{(iii)}{\leq} \frac{1}{T}\sum_{t=0}^{T-1}\mathbb{E}_{\mathbf{x}_{1:t}\sim\mathcal{D}}\Big[\Big\|P_{\boldsymbol{\zeta}^\rho}^{(\mathcal{D},\omega)}(\cdot|\mathbf{x}_{1:t}) - \mathcal{D}(\cdot|\mathbf{x}_{1:t})\Big\|_1\cdot\Big\|\log P_{\hat{\boldsymbol{\theta}}}(\cdot|\mathbf{x}_{1:t}) - \log P_{\boldsymbol{\theta}^*}(\cdot|\mathbf{x}_{1:t})\Big\|_\infty\Big] +$$

$$R^\omega(\hat{\boldsymbol{\theta}}) - R^\omega(\boldsymbol{\theta}^*)$$

$$\overset{(iv)}{\leq} \frac{4M\omega}{T}\cdot\sum_{t=0}^{T-1}\mathbb{E}_{\mathbf{x}_{1:t}\sim\mathcal{D}}\Big[\mathsf{D}_{\mathrm{TV}}\Big(P_{\boldsymbol{\zeta}^\rho}(\cdot|\mathbf{x}_{1:t}), \mathcal{D}(\cdot|\mathbf{x}_{1:t})\Big)\Big] + \underbrace{R^\omega(\hat{\boldsymbol{\theta}}) - R^\omega(\boldsymbol{\theta}^*)}_{(\mathrm{I})}, \tag{19}$$

where $(i)$ follows from adding and subtracting $R^\omega(\hat{\boldsymbol{\theta}}) - R^\omega(\boldsymbol{\theta}^*)$; $(ii)$ employs the definition of $R(\hat{\boldsymbol{\theta}})$, $R(\boldsymbol{\theta}^*)$, $R^\omega(\hat{\boldsymbol{\theta}})$, and $R^\omega(\boldsymbol{\theta}^*)$; $(iii)$ invokes Hölder's inequality; and $(iv)$ follows from the definition of total-variation distance $\mathsf{D}_{\mathrm{TV}}(\cdot,\cdot)$ and the fact that underlying per-token loss terms are bounded by $M$.

Next, we focus on the term (I) in (19):

$$R^\omega(\hat{\boldsymbol{\theta}}) - R^\omega(\boldsymbol{\theta}^*) \overset{(i)}{=} R^\omega(\hat{\boldsymbol{\theta}}) - R_N^\omega(\hat{\boldsymbol{\theta}}) + R_N^\omega(\hat{\boldsymbol{\theta}}) - R_N^\omega(\boldsymbol{\theta}^*) + R_N^\omega(\boldsymbol{\theta}^*) - R^\omega(\boldsymbol{\theta}^*)$$

$$= \Big(R^\omega(\hat{\boldsymbol{\theta}}) - R_N^\omega(\hat{\boldsymbol{\theta}})\Big) + \Big(R_N^\omega(\boldsymbol{\theta}^*) - R^\omega(\boldsymbol{\theta}^*)\Big) + R_N^\omega(\hat{\boldsymbol{\theta}}) - R_N^\omega(\boldsymbol{\theta}^*)$$

$$\overset{(ii)}{\leq} \Big(R^\omega(\hat{\boldsymbol{\theta}}) - R_N^\omega(\hat{\boldsymbol{\theta}})\Big) + \Big(R_N^\omega(\boldsymbol{\theta}^*) - R^\omega(\boldsymbol{\theta}^*)\Big) \tag{20}$$

where $(i)$ follows by adding and subtracting $R_N^\omega(\hat{\boldsymbol{\theta}})$ and $R_N^\omega(\boldsymbol{\theta}^*)$; and $(ii)$ holds as $\hat{\boldsymbol{\theta}}$ is the minimizer of $R_N^\omega(\cdot)$ in $\Theta$ which implies that $R_N^\omega(\hat{\boldsymbol{\theta}}) - R_N^\omega(\boldsymbol{\theta}^*) \leq 0$. Now, the statement in Proposition B.2 follows by combining (19) and (20). $\qquad\square$

Note that the bound on excess surrogate risk in Proposition B.2 decomposes into three terms:

- First term captures the *divergence* between the ground truth per-token distribution and the teacher-induced per-token distribution leveraged during KD; and
- The last two terms corresponds to the deviation between empirical and population surrogate risks for the empirical risk minimizer $\hat{\boldsymbol{\theta}}$ and population risk minimzer $\boldsymbol{\theta}^*$ within the function class $\Theta$. Note that since, $\hat{\boldsymbol{\theta}}$ is a a random variable in itself (which depends on the training sample $\mathcal{S}_N$), one typically needs to bound the deviation uniformly over all functions $\boldsymbol{\theta}\in\Theta$. As

we will see next, one can bound these deviations in terms of the properties of both model class $\Theta$ as well as the teacher-induced per-token distributions.

In order to make the excess surrogate risk bound in Proposition B.2 explicit, we need to bound the third term via a computable quantity. We apply sample variance-based bounds from (Maurer & Pontil, 2009) to get the following result.

**Theorem B.3** (Formal version of Theorem 3.2). *Suppose Assumption B.1 holds. Let $\mathcal{F}^{\zeta,\rho,\omega}$ be a function class that maps elements in $\mathcal{V}^T$ to $[0, M]$ as defined below:*

$$\mathcal{F}^{\omega} := \mathcal{F}^{\zeta,\rho,\omega} \triangleq \left\{ \mathbf{x} \mapsto \frac{1}{T} \sum_{t=1}^{T} \mathsf{CE}\big(P_{\zeta^\rho}^{(\mathcal{D},\omega)}(\cdot|\mathbf{x}_{\leq t-1}), P_{\boldsymbol{\theta}}(\cdot|\mathbf{x}_{\leq t-1})\big), \, \forall \mathbf{x} \in \mathcal{V}^T, \boldsymbol{\theta} \in \Theta \right\}. \quad (21)$$

*For $\epsilon > 0$, let $\mathcal{N}_\infty(\epsilon, \mathcal{F}^{\zeta,\rho,\omega}, N)$ denote the growth function for the function class $\mathcal{F}^{\zeta,\rho,\omega}$, i.e.,*

$$\mathcal{N}_\infty(\epsilon, \mathcal{F}^{\zeta,\rho,\omega}, N) \triangleq \sup_{\mathbf{X} = (\mathbf{x}^{(1)}, \ldots, \mathbf{x}^{(N)}) \in \mathcal{V}^{T \times N}} \mathcal{N}(\epsilon, \mathcal{F}^{\zeta,\rho,\omega}(\mathbf{X}), \|\cdot\|_\infty), \quad (22)$$

*where $\mathcal{N}(\epsilon, \mathcal{F}^{\zeta,\rho,\omega}(\mathbf{X}), \|\cdot\|_\infty)$ denotes the smallest $\epsilon$-cover of the set*

$$\mathcal{F}^{\zeta,\rho,\omega}(\mathbf{X}) = \left\{ \big(f(\mathbf{x}^{(1)}), f(\mathbf{x}^{(2)}), \ldots, f(\mathbf{x}^{(N)})\big) : f \in \mathcal{F}^{\zeta,\rho,\omega} \right\} \subseteq \mathbb{R}^N$$

*with respect to $\|\cdot\|_\infty$ norm. Then, with probability at least $1 - \delta$, for all $\boldsymbol{\theta} \in \Theta$, we have*

$$R(\hat{\boldsymbol{\theta}}) - R(\boldsymbol{\theta}^*) \leq \frac{4M\omega}{T} \cdot \sum_{t=1}^{T} \mathbb{E}_{\mathbf{x}_{\leq t-1} \sim \mathcal{D}} \mathsf{D}_{\mathsf{TV}}\Big( P_{\zeta^\rho}(\cdot|\mathbf{x}_{\leq t-1}), \mathcal{D}(\cdot|\mathbf{x}_{\leq t-1}) \Big)$$

$$+ \sqrt{\frac{18 U_N(f^{\hat{\boldsymbol{\theta}}}, \mathcal{S}_N) \log\left(\frac{2\mathcal{M}(N)}{\delta}\right)}{N}} + \frac{15M \log\left(\frac{2\mathcal{M}(N)}{\delta}\right)}{N - 1}$$

$$+ \sqrt{\frac{2 U_N(f^{\boldsymbol{\theta}^*}, \mathcal{S}_N) \log\left(\frac{4}{\delta}\right)}{N}} + \frac{7M \log\left(\frac{4}{\delta}\right)}{3(N - 1)}, \quad (23)$$

*where $\mathcal{M}(N) \triangleq 10 \cdot \mathcal{N}_\infty(1/N, \mathcal{F}^{\zeta,\rho,\omega}, 2N)$; $f^{\boldsymbol{\theta}}$ denotes the function in $\mathcal{F}^{\zeta,\rho,\omega}$ that corresponds to $\boldsymbol{\theta}$, as per (21); and $U_N(f^{\boldsymbol{\theta}}, \mathcal{S}_N)$ denotes the sample variance*

$$U_N(f^{\boldsymbol{\theta}}, \mathcal{S}_N) = \frac{1}{N(N-1)} \sum_{1 \leq i < j \leq N} \big(f^{\boldsymbol{\theta}}(\mathbf{x}^{(i)}) - f^{\boldsymbol{\theta}}(\mathbf{x}^{(j)})\big)^2. \quad (24)$$

*Proof of Theorem B.3.* As discussed earlier, in light of Proposition B.2, we only need to bound two terms $R^\omega(\hat{\boldsymbol{\theta}}) - R_N^\omega(\hat{\boldsymbol{\theta}})$ and $R_N^\omega(\boldsymbol{\theta}^*) - R^\omega(\boldsymbol{\theta}^*)$ to obtain the desired result. Now utilizing Theorem 6 and Theorem 4 (with $\delta$ replaced with $\delta/2$) in (Maurer & Pontil, 2009) to bound the two terms, respectively, completes the proof of Theorem B.3. $\qquad\square$

### B.2 PROOF OF TOKEN-LEVEL EXCESS RISK BOUND IN THEOREM 3.4

Before providing a proof of Theorem 3.4, we first introduce some intermediate results that are needed to prove the theorem. Recall that our training sample $\mathcal{S}_N = \{\mathbf{x}^{(i)} = [x_1^{(i)}, \ldots, x_T^{(i)}]\}_{i \in [N]}$ comprises $N$ independent sequences such that $\mathbf{x}^{(i)} \sim \mathcal{D}, \forall i \in [N]$. With $\ell^\omega(\mathbf{x}^{(i)}; \boldsymbol{\theta})$ representing the KD loss on $i$-th sequence, we define the random variables

$$Z_0^{(i)} = \mathbb{E}[\ell^\omega(\mathbf{x}^{(i)}; \boldsymbol{\theta})],$$
$$Z_t^{(i)} = \mathbb{E}\left[\ell^\omega(\mathbf{x}^{(i)}; \boldsymbol{\theta}) \mid \mathbf{x}_{\leq t}^{(i)}\right], \text{ for } 1 \leq t \leq T, \quad (25)$$

where $Z_T^{(i)} = \mathbb{E}\left[\ell^\omega(\mathbf{x}^{(i)}; \boldsymbol{\theta}) \mid \mathbf{x}^{(i)}\right] = \ell^\omega(\mathbf{x}^{(i)}; \boldsymbol{\theta})$. Note that $\{Z_t^{(i)}\}_{0 \leq t \leq T}$ is a *Doob martingale sequence* with respect to the natural filtration $\{\mathcal{F}_t^{(i)}\}_{0 \leq t \leq T}$ of the random variables

$\{x_1^{(i)}, \ldots, x_t^{(i)}\}$ (Ross, 1983, pg 297). Accordingly, we define a *martingale difference sequence* $\{\xi_t^{(i)}, \mathcal{F}_t^{(i)}\}_{t \in [T]}$ such that for $t \in [T]$,

$$\xi_t^{(i)} := \xi_t(\mathbf{x}^{(i)}; \boldsymbol{\theta}) = Z_{t-1}^{(i)} - Z_t^{(i)} = \mathbb{E}\left[\ell^\omega(\mathbf{x}; \boldsymbol{\theta})|\mathbf{x}_{\leq t-1}^{(i)}\right] - \mathbb{E}\left[\ell^\omega(\mathbf{x}; \boldsymbol{\theta})|\mathbf{x}_{\leq t}^{(i)}\right]. \qquad (26)$$

As per Assumption 3.3, the following holds for each $t \in [T]$:

$$|\xi_t^{(i)}(\mathbf{x}; \boldsymbol{\theta})| \leq C_t \leq C, \qquad (27)$$

$$\mathbb{E}\left[\left(\xi_t^{(i)}\right)^2 |\mathbf{x}_{\leq t-1}\right] \leq V_t. \qquad (28)$$

We are ready to state the first intermediate result which bounds the moment generating function for the following random variable associated with the KD loss on the $i$-th training sequence:

$$Z_0^{(i)} - Z_T^{(i)} = \mathbb{E}\left[\ell^\omega(\mathbf{x}^{(i)}; \boldsymbol{\theta})\right] - \ell^\omega(\mathbf{x}^{(i)}; \boldsymbol{\theta}).$$

**Lemma B.4.** *Under Assumption 3.3, the following holds for each $i \in [N]$:*

$$\mathbb{E}\left[e^{\lambda \cdot (Z_0^{(i)} - Z_T^{(i)})/C}\right] \leq \exp\left(T \cdot f\left(\lambda, \frac{1}{T}\sum_{t=1}^T \frac{V_t}{C^2}\right)\right), \qquad (29)$$

*where, for $\lambda \geq 0$ and $s \geq 0$,*

$$f(\lambda, s) \triangleq \log\left(\frac{1}{1+s} \cdot \exp(-\lambda s) + \frac{s}{1+s} \cdot \exp(\lambda)\right). \qquad (30)$$

*Proof.* Note that

$$\begin{aligned}
\mathbb{E}\left[e^{\lambda \cdot (Z_0^{(i)} - Z_T^{(i)})/C}\right] &= \mathbb{E}\left[e^{\lambda \cdot \sum_{t=1}^T \xi_t^{(i)}/C}\right] \\
&= \mathbb{E}\left[\mathbb{E}\left[e^{\lambda \cdot \sum_{t=1}^T \xi_t^{(i)}/C}|\mathbf{x}_{\leq T-1}^{(i)}\right]\right] \\
&\overset{(i)}{=} \mathbb{E}\left[e^{\lambda \cdot \sum_{t=1}^{T-1} \xi_t^{(i)}/C} \cdot \mathbb{E}\left[e^{\lambda \cdot \xi_T^{(i)}/C}|\mathbf{x}_{\leq T-1}^{(i)}\right]\right] \\
&\overset{(ii)}{\leq} \mathbb{E}\left[e^{\lambda \cdot \sum_{t=1}^{T-1} \xi_t^{(i)}/C} \cdot e^{f\left(\lambda, \frac{1}{C^2} \cdot \mathbb{E}\left[\left(\xi_T^{(i)}\right)^2 |\mathbf{x}_{\leq T-1}^{(i)}\right]\right)}\right] \\
&\overset{(iii)}{\leq} \mathbb{E}\left[e^{\lambda \cdot \sum_{t=1}^{T-1} \xi_t^{(i)}/C} \cdot e^{f(\lambda, \frac{V_T}{C^2})}\right] \\
&= \mathbb{E}\left[e^{\lambda \cdot \sum_{t=1}^{T-1} \xi_t^{(i)}/C}\right] \cdot e^{f(\lambda, \frac{V_T}{C^2})} \qquad (31)
\end{aligned}$$

where $(i)$ follows as $e^{\lambda \cdot \sum_{t=1}^{T-1} \xi_t^{(i)}}$ is $\mathcal{F}_{T-1}^{(i)}$-measurable; $(ii)$ follows from (Fan et al., 2012, Lemma 3.1); and $(iii)$ follows from Assumption 3.3 and the fact that, for $\lambda > 0$ and $s \geq 0$, $f(\lambda, s)$ is an increasing function in its second argument (Fan et al., 2012, Lemma 3.2). By following the similar steps in (31) for $\xi_{i,T-1}, \xi_{i,T-2}, \ldots, \xi_{i,1}$, we obtain that

$$\mathbb{E}\left[e^{\lambda \cdot (Z_0^{(i)} - Z_T^{(i)})/C}\right] \leq e^{\sum_{t=1}^T f(\lambda, \frac{V_t}{C^2})}. \qquad (32)$$

According to (Fan et al., 2012, Lemma 3.2) that, for $\lambda \geq 0$ and $s \geq 0$, $f(\lambda, s)$ is a concave function in its second argument. Thus, it follows from Jensen's inquality that

$$\frac{1}{T}\sum_{t=1}^T f\left(\lambda, \frac{V_t}{C^2}\right) \leq f\left(\lambda, \frac{1}{T}\sum_{t=1}^T \frac{V_t}{C^2}\right). \qquad (33)$$

By combining (32) and (33), we have

$$\mathbb{E}\left[e^{\lambda \cdot (Z_0^{(i)} - Z_T^{(i)})/C}\right] \leq e^{T \cdot f\left(\lambda, \frac{1}{T}\sum_{t=1}^T \frac{V_t}{C^2}\right)}, \qquad (34)$$

which completes the proof. $\qquad \square$

Now we can leverage Lemma B.4 to obtain the following concentration inequality for the KD training objective.

**Lemma B.5.** *Let $\zeta$ and $\theta \in \Theta$ denote the teacher and student LM, respectively. Then, for $\epsilon > 0$, the following holds under Assumption 3.3.*

$$\mathbb{P}\left(\sum_{i=1}^{N}\left(\mathbb{E}\left[\ell^{\omega}(\mathbf{x}^{(i)}; \boldsymbol{\theta})\right] - \ell^{\omega}(\mathbf{x}^{(i)}; \boldsymbol{\theta})\right)/C \geq N\epsilon\right) \leq \exp\left(-\frac{N\epsilon^2}{2(\sum_t \frac{V_t}{C^2} + \frac{1}{3}\epsilon)}\right). \quad (35)$$

*Proof.* Recall that, as per our notation, we have

$$\mathbb{E}\left[\ell^{\omega}(\mathbf{x}^{(i)}; \boldsymbol{\theta})\right] - \ell^{\omega}(\mathbf{x}^{(i)}; \boldsymbol{\theta}) = Z_0^{(i)} - Z_T^{(i)}.$$

Thus,

$$\mathbb{P}\left(\sum_{i=1}^{N}\left(\mathbb{E}\left[\ell^{\omega}(\mathbf{x}^{(i)}; \boldsymbol{\theta})\right] - \ell^{\omega}(\mathbf{x}^{(i)}; \boldsymbol{\theta})\right)/C \geq N\epsilon\right) = \mathbb{P}\left(\sum_{i=1}^{N}\left(Z_0^{(i)} - Z_T^{(i)}\right)/C \geq N\epsilon\right). \quad (36)$$

It follows from Markov's inequality that, for $\lambda \geq 0$,

$$\mathbb{P}\left(\sum_{i=1}^{N}\left(Z_0^{(i)} - Z_T^{(i)}\right)/C \geq N\epsilon\right) = \mathbb{P}\left(e^{\lambda \cdot \sum_{i=1}^{N}\left(Z_0^{(i)} - Z_T^{(i)}\right)/C} \geq e^{N\lambda\epsilon}\right)$$

$$\leq \frac{\mathbb{E}\left[e^{\lambda \cdot \sum_{i=1}^{N}\left(Z_0^{(i)} - Z_T^{(i)}\right)/C}\right]}{e^{N\lambda\epsilon}}$$

$$\overset{(i)}{=} \frac{\prod_{i\in[N]}\mathbb{E}\left[e^{\lambda \cdot \left(Z_0^{(i)} - Z_T^{(i)}\right)/C}\right]}{e^{N\lambda\epsilon}}, \quad (37)$$

where $(i)$ follows as $\{Z_0^{(i)} - Z_T^{(i)}\}_{i\in[N]}$ are independent random variables. By combining (37) with Lemma B.4, we obtain that

$$\mathbb{P}\left(\sum_{i=1}^{N}\left(Z_0^{(i)} - Z_T^{(i)}\right)/C \geq N\epsilon\right) \leq e^{-N\cdot\left(\lambda\epsilon - T\cdot f(\lambda, \frac{1}{T}\sum_{t=1}^{T}\frac{V_t}{C^2})\right)}. \quad (38)$$

Since (38) holds for each $\lambda \geq 0$, we have

$$\mathbb{P}\left(\sum_{i=1}^{N}\left(Z_0^{(i)} - Z_T^{(i)}\right)/C \geq N\epsilon\right) \leq \inf_{\lambda \geq 0} e^{-N\cdot\left(\lambda\epsilon - T\cdot f(\lambda, \frac{1}{T}\sum_{t=1}^{T}\frac{V_t}{C^2})\right)} \quad (39)$$

Now as argued in the Proof of Remark 2.1 in (Fan et al., 2012), for $0 \leq \lambda < 3, s \geq 0$, we have

$$f(\lambda, s) \leq (e^{\lambda} - 1 - \lambda)s \leq \frac{\lambda^2 s}{2(1 - \frac{1}{3}\lambda)}. \quad (40)$$

Thus, it follows from (39) that

$$\mathbb{P}\left(\sum_{i=1}^{N}\left(Z_0^{(i)} - Z_T^{(i)}\right)/C \geq N\epsilon\right) \leq \inf_{0 \leq \lambda < 3} \exp\left(-N \cdot \left(\lambda\epsilon - \frac{\lambda^2}{2(1 - \frac{1}{3}\lambda)} \cdot \sum_t \frac{V_t}{C^2}\right)\right)$$

$$\leq \exp\left(-\frac{N\epsilon^2}{2(\sum_t \frac{V_t}{C^2} + \frac{1}{3}\epsilon)}\right). \quad (41)$$

This completes the proof. $\qquad \square$

Equipped with Lemma B.5, we are now ready to prove Theorem 3.4 below.

*Proof of Theorem 3.4.* Note that, we have the following from Proposition B.2.

$$R(\hat{\boldsymbol{\theta}}) - R(\boldsymbol{\theta}^*) \leq \frac{4M\omega}{T} \cdot \sum_{t=1}^{T} \mathbb{E}_{\mathbf{x}_{\leq t-1} \sim \mathcal{D}} \mathsf{D}_{\mathrm{TV}}\Big(P_{\boldsymbol{\zeta}^p}(\cdot|\mathbf{x}_{\leq t-1}), \mathcal{D}(\cdot|\mathbf{x}_{\leq t-1})\Big)$$

$$+ \underbrace{\Big(R^\omega(\hat{\boldsymbol{\theta}}) - R_N^\omega(\hat{\boldsymbol{\theta}})\Big)}_{\text{(I)}} + \underbrace{(R_N^\omega(\boldsymbol{\theta}^*) - R^\omega(\boldsymbol{\theta}^*))}_{\text{(II)}}. \tag{42}$$

Next, we focus on bounding the term (I). As per notation, for any $\boldsymbol{\theta} \in \Theta$, we have

$$R^\omega(\boldsymbol{\theta}) - R_N^\omega(\boldsymbol{\theta}) = \frac{1}{N} \sum_{i=1}^{N} \Big(\mathbb{E}\big[\ell^\omega(\mathbf{x}^{(i)}; \boldsymbol{\theta})\big] - \ell^\omega(\mathbf{x}^{(i)}; \boldsymbol{\theta})\Big). \tag{43}$$

Thus, for a fixed $\boldsymbol{\theta} \in \Theta$, we have

$$\mathbb{P}\left(R^\omega(\boldsymbol{\theta}) - R_N^\omega(\boldsymbol{\theta}) \geq \gamma\right) = \mathbb{P}\left(\frac{1}{N} \sum_{i=1}^{N} \Big(\mathbb{E}\big[\ell^\omega(\mathbf{x}^{(i)}; \boldsymbol{\theta})\big] - \ell^\omega(\mathbf{x}^{(i)}; \boldsymbol{\theta})\Big) \geq \gamma\right)$$

$$= \mathbb{P}\left(\sum_{i=1}^{N} \Big(\mathbb{E}\big[\ell^\omega(\mathbf{x}^{(i)}; \boldsymbol{\theta})\big] - \ell^\omega(\mathbf{x}^{(i)}; \boldsymbol{\theta})\Big)/C \geq N \cdot \frac{\gamma}{C}\right)$$

$$\overset{(i)}{\leq} \exp\left(-\frac{N\gamma^2}{2(\sum_t V_t + \frac{1}{3}C\gamma)}\right), \tag{44}$$

where $(i)$ follows from (35) with $\epsilon = \frac{\gamma}{C}$. With some algebra, one can see that the right hand side of (44) is bounded by $\delta/(2|\Theta|)$ when

$$\gamma \geq \frac{2C}{3N} \cdot \log\left(2|\Theta|/\delta\right) + \sqrt{\frac{2}{N} \cdot \sum_t V_t \cdot \log\left(2|\Theta|/\delta\right)}. \tag{45}$$

(To see this, set the right hand side of (44) to $\delta/(2|\Theta|)$ to get a quadratic of the form $\gamma^2 = a\gamma + b$ with $a, b \geq 0$ and note that its non-negative root is $\leq a + \sqrt{b}$. All $\gamma \geq a + \sqrt{b}$ will make the right hand side of (44) $\leq \delta/(2|\Theta|)$.)

Now, by taking union bound, with probability at least $1 - \delta/2$, for all $\boldsymbol{\theta} \in \Theta$, we have the following.

$$R^\omega(\boldsymbol{\theta}) - R_N^\omega(\boldsymbol{\theta}) \leq \frac{2C}{3N} \cdot \log\left(2|\Theta|/\delta\right) + \sqrt{\frac{2}{N} \cdot \sum_t V_t \cdot \log\left(2|\Theta|/\delta\right)}. \tag{46}$$

Since the minimizer of the KD training objective $\hat{\boldsymbol{\theta}}$ is in $\Theta$, with probablity at least $1 - \delta/2$, we have

$$\text{(I)} = R^\omega(\hat{\boldsymbol{\theta}}) - R_N^\omega(\hat{\boldsymbol{\theta}}) \leq \frac{2C}{3N} \cdot \log\left(2|\Theta|/\delta\right) + \sqrt{\frac{2}{N} \cdot \sum_t V_t \cdot \log\left(2|\Theta|/\delta\right)}. \tag{47}$$

As for the term (II), one can follow the arguments in Lemma B.4 and B.5 with $Z_T^{(i)} - Z_0^{(i)}$ instead of $Z_0^{(i)} - Z_T^{(i)}$ to obtain that, with probability at least $1 - \delta/2$, we have

$$\text{(II)} = R_N^\omega(\boldsymbol{\theta}^*) - R^\omega(\boldsymbol{\theta}^*) \leq \frac{2C}{3N} \cdot \log\left(2/\delta\right) + \sqrt{\frac{2}{N} \cdot \sum_t V_t \cdot \log\left(2/\delta\right)}. \tag{48}$$

Note that, since $\boldsymbol{\theta}^*$ is a fixed element in $\Theta$, we do not need to take a union bound over all elements in $\Theta$ (as in (47)) to obtain (48). Now, the statement of Theorem 3.4 follows by combining (42), (47), and (48). $\square$

### B.3   ALTERNATIVE EXCESS SURROGATE RISK BOUND

In this section, we derive an excess risk bound similar to Theorem 3.4 with a subgaussian assumption on $\xi_t(\mathbf{x}; \boldsymbol{\theta})$ (Assumption B.6) instead of Assumption 3.3. The derived bound depends on the *mean* of certain subgaussian variance proxies, instead of the *supremal* quantities $C, V_t$ from Assumption 3.3. The subgaussian assumption is as follows.

**Assumption B.6.** For $i \in [N], t \in [T]$, for a all $\boldsymbol{\theta}$ in a finite class $\Theta$, let $\xi_t(\mathbf{x}^{(i)}; \boldsymbol{\theta})$ satisfy:

$$\mathbb{E}\left[e^{\lambda \xi_t(\mathbf{x}^{(i)}; \boldsymbol{\theta})} | \mathbf{x}_{\leq t-1}^{(i)}\right] \leq e^{\frac{1}{2}\lambda^2 \sigma_t^{(i)}} \quad \text{for all } \lambda \geq 0, \tag{49}$$

for a finite variance proxy $\sigma_t^{(i)} \geq 0$.

Now we state the alternative excess risk bound based on Assumption B.6.

**Theorem B.7.** *Let $\Theta$ be the finite function class in Assumption B.6. Under Assumption B.6, with probability at least $1 - \delta$, the following holds for the student LM $\hat{\boldsymbol{\theta}} \in \Theta$ obtained via KD:*

$$R(\hat{\boldsymbol{\theta}}) - R(\boldsymbol{\theta}^*) \leq \frac{1}{N} \sqrt{2 \sum_{i=1}^{N} \sum_{t=1}^{T} \sigma_t^{(i)} \cdot \left(\sqrt{\log\left(2|\Theta|/\delta\right)} + \sqrt{\log\left(2/\delta\right)}\right)} +$$

$$(4M\omega)/T \cdot \sum_{t \in [T]} \mathbb{E}_{\mathbf{x}_{\leq t-1} \sim \mathcal{D}} \mathsf{D}_{\mathrm{TV}}\left(P_{\boldsymbol{\zeta}^\rho}(\cdot | \mathbf{x}_{\leq t-1}), \mathcal{D}(\cdot | \mathbf{x}_{\leq t-1})\right). \tag{50}$$

*Remark* B.8 (Dependence on mean subgaussian variance proxy). Theorem B.7 gives a bound that depends on the *mean* subgaussian variance proxy $\sigma := \frac{1}{NT} \sum_{i,t} \sigma_t^{(i)}$ instead of the *supremal* quantities $C, V_t$ that appear in Theorem 3.4. Therefore, this bound is potentially sharper than the one in Theorem 3.4.

The proof of Theorem B.7 follows the same steps as that of Theorem 3.4, except in some arguments where we naturally replace Assumption 3.3 with Assumption B.6. However, we write down almost all the steps for the reader's convenience.

*Proof of Theorem B.7.* Note that, we have the following from Proposition B.2.

$$R(\hat{\boldsymbol{\theta}}) - R(\boldsymbol{\theta}^*) \leq \frac{4M\omega}{T} \cdot \sum_{t=1}^{T} \mathbb{E}_{\mathbf{x}_{\leq t-1} \sim \mathcal{D}} \mathsf{D}_{\mathrm{TV}}\left(P_{\boldsymbol{\zeta}^\rho}(\cdot | \mathbf{x}_{\leq t-1}), \mathcal{D}(\cdot | \mathbf{x}_{\leq t-1})\right)$$

$$+ \underbrace{\left(R^\omega(\hat{\boldsymbol{\theta}}) - R_N^\omega(\hat{\boldsymbol{\theta}})\right)}_{(\mathrm{I})} + \underbrace{\left(R_N^\omega(\boldsymbol{\theta}^*) - R^\omega(\boldsymbol{\theta}^*)\right)}_{(\mathrm{II})}. \tag{51}$$

Next, we focus on bounding the term (I). As per notation, for any $\boldsymbol{\theta} \in \Theta$, we have

$$R^\omega(\boldsymbol{\theta}) - R_N^\omega(\boldsymbol{\theta}) = \frac{1}{N} \sum_{i=1}^{N} \left(\mathbb{E}\left[\ell^\omega(\mathbf{x}^{(i)}; \boldsymbol{\theta})\right] - \ell^\omega(\mathbf{x}^{(i)}; \boldsymbol{\theta})\right). \tag{52}$$

Thus, for a fixed $\boldsymbol{\theta} \in \Theta$, we have

$$\mathbb{P}\left(R(\boldsymbol{\theta}) - R_N^\omega(\boldsymbol{\theta}) \geq \gamma\right) = \mathbb{P}\left(\frac{1}{N} \sum_{i=1}^{N} \left(\mathbb{E}\left[\ell^\omega(\mathbf{x}^{(i)}; \boldsymbol{\theta})\right] - \ell^\omega(\mathbf{x}^{(i)}; \boldsymbol{\theta})\right) \geq \gamma\right)$$

$$\leq \exp\left(-\frac{N^2 \gamma^2}{2 \sum_{i,t} \sigma_t^{(i)}}\right) \tag{53}$$

where the inequality follows from Lemma B.9 below (we utilize (58) with $\epsilon = \gamma$). One can see that the right hand side of (53) is bounded by $\delta/(2|\Theta|)$ when

$$\gamma \geq \frac{1}{N} \sqrt{2 \sum_{i,t} \sigma_t^{(i)} \cdot \log\left(2|\Theta|/\delta\right)}. \tag{54}$$

Now, by taking union bound, with probability at least $1 - \delta/2$, for all $\boldsymbol{\theta} \in \Theta$, we have

$$R^\omega(\boldsymbol{\theta}) - R_N^\omega(\boldsymbol{\theta}) \leq \frac{1}{N} \sqrt{2 \sum_{i,t} \sigma_t^{(i)} \cdot \log\left(2|\Theta|/\delta\right)}. \tag{55}$$

Since the minimizer of the KD training objective $\hat{\boldsymbol{\theta}}$ is in $\Theta$, with probablity at least $1 - \delta/2$, we have

$$(\mathrm{I}) = R^\omega(\hat{\boldsymbol{\theta}}) - R_N^\omega(\hat{\boldsymbol{\theta}}) \leq \frac{1}{N} \sqrt{2 \sum_{i,t} \sigma_t^{(i)} \cdot \log\left(2|\Theta|/\delta\right)}. \tag{56}$$

As for the term (II), one can follow the arguments in Lemma B.10 and B.9 with $Z_T^{(i)} - Z_0^{(i)}$ instead of $Z_0^{(i)} - Z_T^{(i)}$ to obtain that, with probability at least $1 - \delta/2$, we have

$$(\text{II}) = R_N^\omega(\boldsymbol{\theta}^*) - R^\omega(\boldsymbol{\theta}^*) \leq \frac{1}{N} \sqrt{2 \sum_{i,t} \sigma_t^{(i)} \cdot \log(2/\delta)}. \tag{57}$$

Now, the statement of Theorem B.7 follows by combining (51), (56), and (57). □

**Lemma B.9.** *Let $\zeta$ and $\boldsymbol{\theta} \in \Theta$ denote the teacher and student LM, respectively. Then, under the Assumption B.6, for any $\epsilon > 0$, the following holds:*

$$\mathbb{P}\left( \sum_{i=1}^N \left( \mathbb{E}[\ell^\omega(\mathbf{x}^{(i)}; \boldsymbol{\theta})] - \ell^\omega(\mathbf{x}^{(i)}; \boldsymbol{\theta}) \right) \geq N\epsilon \right) \leq \exp\left( -\frac{N^2 \epsilon^2}{2 \sum_i \sum_t \sigma_t^{(i)}} \right). \tag{58}$$

*Proof of Lemma B.9.* Recall that, as per our notation, we have

$$\mathbb{E}\left[ \ell^\omega(\mathbf{x}^{(i)}; \boldsymbol{\theta}) \right] - \ell^\omega(\mathbf{x}^{(i)}; \boldsymbol{\theta}) = Z_0^{(i)} - Z_T^{(i)}.$$

Thus,

$$\mathbb{P}\left( \sum_{i=1}^N \left( \mathbb{E}[\ell^\omega(\mathbf{x}^{(i)}; \boldsymbol{\theta})] - \ell^\omega(\mathbf{x}^{(i)}; \boldsymbol{\theta}) \right) \geq N\epsilon \right) = \mathbb{P}\left( \sum_{i=1}^N \left( Z_0^{(i)} - Z_T^{(i)} \right) \geq N\epsilon \right).$$

It follows from Markov's inequality that, for $\lambda \geq 0$,

$$\mathbb{P}\left( \sum_{i=1}^N \left( Z_0^{(i)} - Z_T^{(i)} \right) \geq N\epsilon \right) = \mathbb{P}\left( e^{\lambda \cdot \sum_{i=1}^N \left( Z_0^{(i)} - Z_T^{(i)} \right)} \geq e^{N\lambda\epsilon} \right)$$

$$\leq \frac{\mathbb{E}\left[ e^{\lambda \cdot \sum_{i=1}^N \left( Z_0^{(i)} - Z_T^{(i)} \right)} \right]}{e^{N\lambda\epsilon}}$$

$$\overset{(i)}{=} \frac{\prod_{i \in [N]} \mathbb{E}\left[ e^{\lambda \cdot \left( Z_0^{(i)} - Z_T^{(i)} \right)} \right]}{e^{N\lambda\epsilon}}, \tag{59}$$

where $(i)$ follows as $\{ Z_0^{(i)} - Z_T^{(i)} \}_{i \in [N]}$ are independent random variables. By combining (59) with Lemma B.10, we write

$$\mathbb{P}\left( \sum_{i=1}^N \left( Z_0^{(i)} - Z_T^{(i)} \right) \geq N\epsilon \right) \leq e^{-N \cdot \left( \lambda\epsilon - \frac{1}{2}\lambda^2 \sum_{i,t} \sigma_t^{(i)} \right)}. \tag{60}$$

Since (60) holds for each $\lambda \geq 0$, we get the desired bound by minimizing the right hand side with respect to $\lambda \geq 0$. This completes the proof. □

**Lemma B.10.** *Under Assumption B.6, the following holds for any $i \in [N]$, $\lambda \geq 0$:*

$$\mathbb{E}\left[ e^{\lambda \cdot (Z_0^{(i)} - Z_T^{(i)})} \right] \leq \exp\left( \frac{1}{2}\lambda^2 \sum_{t=1}^T \sigma_t^{(i)} \right). \tag{61}$$

*Proof of Lemma B.10.* Note that

$$\mathbb{E}\left[ e^{\lambda \cdot (Z_0^{(i)} - Z_T^{(i)})} \right] = \mathbb{E}\left[ e^{\lambda \cdot \sum_{t=1}^T \xi_t^{(i)}} \right]$$

$$= \mathbb{E}\left[ \mathbb{E}\left[ e^{\lambda \cdot \sum_{t=1}^T \xi_t^{(i)}} | \mathbf{x}_{\leq T-1}^{(i)} \right] \right]$$

$$\overset{(i)}{=} \mathbb{E}\left[ e^{\lambda \cdot \sum_{t=1}^{T-1} \xi_t^{(i)}} \cdot \mathbb{E}\left[ e^{\lambda \cdot \xi_T^{(i)}} | \mathbf{x}_{\leq T-1}^{(i)} \right] \right]$$

$$\overset{(ii)}{\leq} \mathbb{E}\left[ e^{\lambda \cdot \sum_{t=1}^{T-1} \xi_t^{(i)}} \cdot e^{\frac{1}{2}\lambda^2 \sigma_T^{(i)}} \right]$$

$$= \mathbb{E}\left[ e^{\lambda \cdot \sum_{t=1}^{T-1} \xi_t^{(i)}} \right] \cdot e^{\frac{1}{2}\lambda^2 \sigma_T^{(i)}} \tag{62}$$

where $(i)$ follows as $e^{\lambda \cdot \sum_{t=1}^{T-1} \xi_t^{(i)}}$ is $\mathcal{F}_{T-1}^{(i)}$-measurable; $(ii)$ follows from (49). Repeatedly peeling out the terms for $\xi_{i,T-1}, \xi_{i,T-2}, \ldots, \xi_{i,1}$, we obtain

$$\mathbb{E}\left[e^{\lambda \cdot (Z_0^{(i)} - Z_T^{(i)})}\right] \leq e^{\frac{1}{2}\lambda^2 \sum_{t=1}^{T} \sigma_t^{(i)}}. \qquad \square$$

### B.4 BOUNDING EXCESS RISK FOR KD

Different from the surrogate (empirical or population) risks utilized in the main text (cf. §3), which employs the cross-entropy loss as a surrogate loss, one could directly work with the risk defined with respect to a particular evaluation metric (and the corresponding loss) that one cares about. Since our training focuses on correct next-token prediction, we can focus on the accuracy of the next-token prediction under greedy-decoding as one such metric. This amounts to the following (population) risk with respect to $0/1$-*loss*.

$$R_{0/1}(\boldsymbol{\theta}) := \mathbb{E}_{\mathbf{x}\sim\mathcal{D}}\left[\sum_{t=1}^{T} \mathbb{1}\{\arg\max_{v} P_{\boldsymbol{\theta}}(v|\mathbf{x}_{\leq t-1}) \neq x_t\right]$$

$$= \sum_{t=1}^{T} \mathbb{E}_{\mathbf{x}_{\leq t-1}\sim\mathcal{D}}\left[\sum_{v\in\mathcal{V}} \mathcal{D}(v|\mathbf{x}_{\leq t-1}) \cdot \mathbb{1}\{\arg\max_{v'} P_{\boldsymbol{\theta}}(v'|\mathbf{x}_{\leq t-1}) \neq v\}\right], \qquad (63)$$

where $\mathbb{1}\{\cdot\}$ denotes the indicator function. A large body of literature (see, e.g., Bartlett et al., 2006; Zhang, 2004; Steinwart, 2007; Pires & Szepesvári, 2016, and references therein) has studied *calibration functions* that enable converting bounds on *excess surrogate risk* to control the *excess risk*. Applying the calibration functions for the cross-entropy loss (Pires & Szepesvári, 2016), we obtain the following bound on the excess risk for next-token prediction:

$$R_{0/1}(\hat{\boldsymbol{\theta}}) - R_{0/1}(\boldsymbol{\theta}^*) \leq \mathrm{g}^{-1}\left(R(\hat{\boldsymbol{\theta}}) - R(\boldsymbol{\theta}^*)\right), \qquad (64)$$

where $\mathrm{g}^{-1}(\cdot)$ denotes the inverse of the function $\mathrm{g} : \epsilon \mapsto \frac{1}{2}\big((1-\epsilon)\log(1-\epsilon) + (1+\epsilon)\log(1+\epsilon)\big)$.

# C    DISTRIBUTION OF $\xi_t$ AND $V_t$

In this section, we attempt to understand the distribution of $\xi_t(\mathbf{x}; \boldsymbol{\theta})$ as $\mathbf{x} \sim \mathcal{D}$ for a learned model parameterized by $\boldsymbol{\theta}$ to validate Assumption 3.3 in §3. Recall from (8) that

$$\xi_t(\mathbf{x}; \boldsymbol{\theta}) = \mathbb{E}_{\mathbf{z} \sim \mathcal{D}}\left[\ell^\omega(\mathbf{z}; \boldsymbol{\theta}) | \mathbf{z}_{\leq t-1} = \mathbf{x}_{\leq t-1}\right] - \mathbb{E}_{\mathbf{z} \sim \mathcal{D}}\left[\ell^\omega(\mathbf{z}; \boldsymbol{\theta}) | \mathbf{z}_{\leq t} = \mathbf{x}_{\leq t}\right], \quad t \in [T]. \quad (65)$$

In order to estimate $\xi_t(\mathbf{x}; \boldsymbol{\theta})$, we intend to use a plugin estimator for the two expectations in this equation. To estimate, say, the second expectation above with a Monte-Carlo average, we need to be able to sample *completions* of $\mathbf{x}_{<t}$ so that the completed sequence follows the distribution $\mathcal{D}$. The best access to data distribution $\mathcal{D}$ is via the training data; however, it is generally not possible to sample *multiple* completions starting with the *same* prefix $\mathbf{x}_{<t}$ from the training data. Due to this difficulty, we sample completions from an *oracle* language model, as an approximation to the true data distribution. We use the BASELINE 8.6B model described in Appendix I as our oracle.

For a sequence $\mathbf{x}$ and prefix length $t \in [T]$, we employ the plugin estimate

$$\widehat{\xi}_t(\mathbf{x}; \boldsymbol{\theta}) := \frac{1}{n_{\mathrm{com}}} \sum_{i=1}^{n_{\mathrm{com}}} \ell^\omega([\mathbf{x}_{1:t-1}, \mathbf{y}^i(\mathbf{x}_{1:t-1})]; \boldsymbol{\theta}) - \frac{1}{n_{\mathrm{com}}} \sum_{i=1}^{n_{\mathrm{com}}} \ell^\omega([\mathbf{x}_{1:t}, \mathbf{y}^i(\mathbf{x}_{1:t})]; \boldsymbol{\theta}) \quad (66)$$

where $\mathbf{y}^i(\mathbf{x}_{1:s})$, for $s \in [T]$, is a completion of $\mathbf{x}_{1:s}$, generated by the oracle, with a length of $|\mathbf{y}^i(\mathbf{x}_{1:s})| = (T - s)$ so that the concatenation $[\mathbf{x}_{1:s}, \mathbf{y}^i(\mathbf{x}_{1:s})]$ has length $T$. $n_{\mathrm{com}}$ denotes the number of completions sampled from the oracle for estimating the expectations. We also compute the estimate $\widehat{V}_t(\mathbf{x}; \boldsymbol{\theta})$ of $V_t(\mathbf{x}; \boldsymbol{\theta}) := \mathbb{E}[\xi_t^2(\mathbf{x}; \boldsymbol{\theta}) | x_{\leq t-1}]$ from $\widehat{\xi}_t(\mathbf{x}; \boldsymbol{\theta})$.

We compute $\widehat{\xi}_t(\mathbf{x}; \boldsymbol{\theta})$ and $\widehat{V}_t(\mathbf{x}; \boldsymbol{\theta})$ for two models: BASELINE 1.5B and BASELINE 2.8B LMs. As for $\mathbf{x}$, we employ sequences in the validation set (held out from training any model, including the oracle). The number of completions is $n_{\mathrm{com}} = 64$. The validation set size $n_{\mathrm{val}} \approx 200K$.

In Figure 4, we observe that for 2.8B LM the estimates of $\widehat{\xi}_t$ increasingly concentrate around 0 as $t$ increases. We compute the mean $|\widehat{\xi}_t|$ with $\frac{1}{n_{\mathrm{val}}} \sum_{i=1}^{n_{\mathrm{val}}} |\widehat{\xi}_t(x^{(i)}; \boldsymbol{\theta})|$ where $\{x^{(i)}, i \in [n_{\mathrm{val}}]\}$ is the validation set. From the rightmost panel of Figure 4, we see that the mean $|\widehat{\xi}_t|$ decreases quickly with $t$. For the first few tokens of the sequence, it is hard to predict the next token, because the context is not sufficient to make a good prediction. Hence the loss and variation are high for small $t$. These observations suggest that the magnitude of $\xi_t$ decreases with $t$. Intuitively, the upper bounds $C_t$ and $V_t$ defined in Assumption 3.3 should also decrease with $t$. Similar results hold for 1.5B LM in Figure 6. The plots in Figure 5 for 2.8B model (and Figure 7 for 1.5B model) show that the variance is small and decreases with $t$ as well.

Table 6 shows that for the 2.8B LM, the mean $|\widehat{\xi}_t|$ decreases as the sequence length $T$ increases from 64 to 1280. For modern LMs configured to train on much longer sequences, the table indicates that the mean $|\xi_t|$ are likely to be small. Table 8 shows similar behavior for 1.5B LM. Similar comments apply to the variance terms in Tables 7 and 9.

Table 6: Mean $|\widehat{\xi}_t|$ for **2.8B parameter** model decreases as we increase the sequence length $T$. Conversely, for a fixed sequence length $T$, mean $|\widehat{\xi}_t|$ decreases with prefix length $t$. "−" indicates that the entry is not meaningful because the prefix length $t$ is more than the sequence length.

| $T$ | $t = 1$ | $t = 5$ | $t = 10$ | $t = 30$ | $t = 100$ | $t = 300$ | $t = 640$ |
|---|---|---|---|---|---|---|---|
| 64 | 0.338 | 0.133 | 0.106 | 0.087 | − | − | − |
| 128 | 0.261 | 0.099 | 0.074 | 0.057 | 0.042 | − | − |
| 256 | 0.188 | 0.082 | 0.060 | 0.044 | 0.033 | − | − |
| 512 | 0.134 | 0.071 | 0.053 | 0.039 | 0.031 | 0.019 | − |
| 1280 | 0.109 | 0.058 | 0.041 | 0.031 | 0.027 | 0.023 | 0.016 |

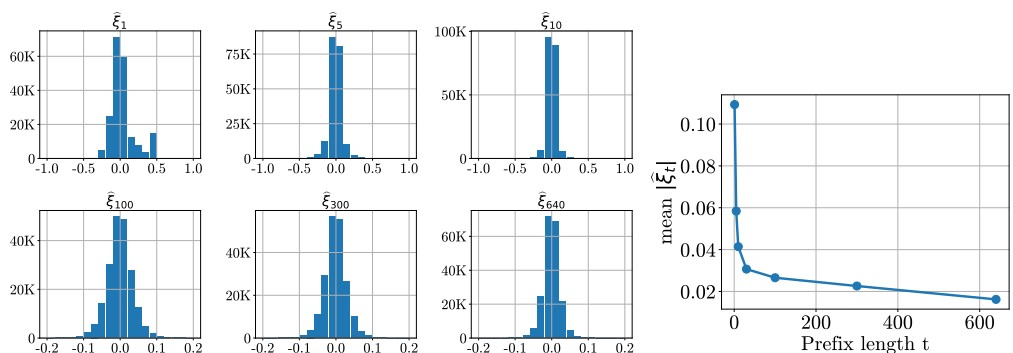

Figure 4: Histograms of $\widehat{\bar{\xi}}_t$, for prefix lengths $t \in \{1, 5, 10, 100, 300, 640\}$ for BASELINE **2.8B parameter** model, estimated with $n_{\text{com}} = 64$ completions for each expectation in (65). The sequence length is 1280. The distribution gets concentrated around 0 as $t$ increases. In the bottom histograms, we reduce the x-axis range to one-fifth that of the top histograms, to focus on the trend within the bottom row. In the rightmost plot, the mean of $|\widehat{\bar{\xi}}_t|$ over validation set decreases rapidly with $t$.

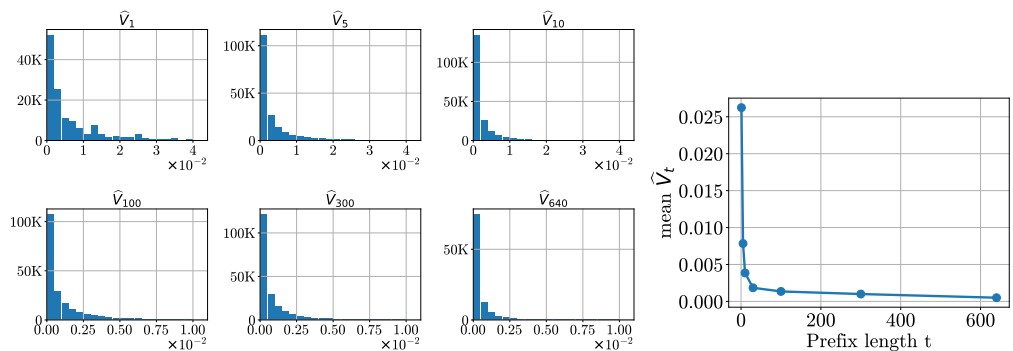

Figure 5: Histograms of $\widehat{V}_t(x; \boldsymbol{\theta})$, for prefix lengths $t \in \{1, 5, 10, 100, 300, 640\}$ for BASELINE **2.8B parameter** model, estimated from $\widehat{\bar{\xi}}_t(x; \boldsymbol{\theta})$. The sequence length is 1280. The distribution gets concentrated around 0 as $t$ increases. In the bottom histograms, we reduce the x-axis range to one-fourth that of the top histograms, to focus on the trend within the bottom row. In the rightmost plot, the mean of $\widehat{V}_t$ over validation set decreases rapidly with $t$.

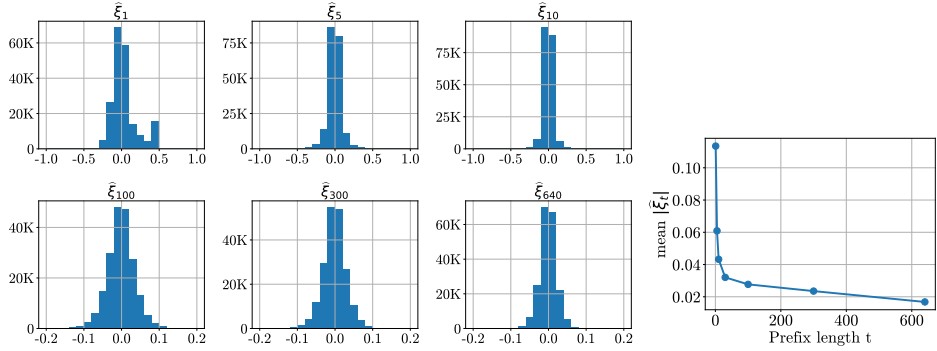

Figure 6: Histograms of $\widehat{\bar{\xi}}_t$, $t \in \{1, 5, 10, 100, 300, 640\}$ for BASELINE **1.5B parameter** model, estimated with $n_{\text{com}} = 64$ completions for each expectation in (65). The sequence length is 1280. The distribution gets concentrated around 0 as $t$ increases. In the bottom histograms, we reduce the x-axis range to about one-fifth that of the top histograms, to focus on the trend within the bottom row. In the rightmost plot, the mean of $|\widehat{\bar{\xi}}_t|$ over validation set decreases rapidly with $t$.

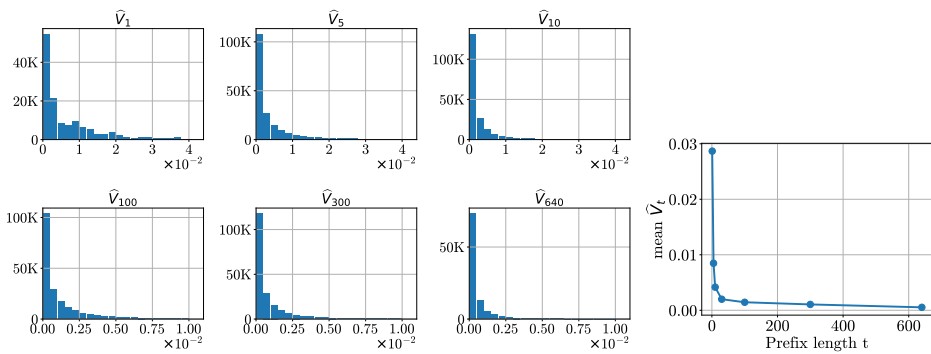

Figure 7: Histograms of $\widehat{V}_t(x; \boldsymbol{\theta})$, for prefix lengths $t \in \{1, 5, 10, 100, 300, 640\}$ for BASELINE **1.5B parameter** model, estimated from $\widehat{\xi}_t(x; \boldsymbol{\theta})$. The sequence length is 1280. The distribution gets concentrated around 0 as $t$ increases. In the bottom histograms, we reduce the x-axis range to one-fourth that of the top histograms, to focus on the trend within the bottom row. In the rightmost plot, the mean of $\widehat{V}_t$ over validation set decreases rapidly with $t$.

Table 7: Mean $\widehat{V}_t$ for **2.8B parameter** model decreases as we increase the sequence length $T$. Conversely, for a fixed sequence length $T$, mean $V_t$ decreases with prefix length $t$. "$-$" indicates that the entry is not meaningful because the prefix length $t$ is more than the sequence length.

| $T$ | $t = 1$ | $t = 5$ | $t = 10$ | $t = 30$ | $t = 100$ | $t = 300$ | $t = 640$ |
|---|---|---|---|---|---|---|---|
| 64 | 0.1597 | 0.0351 | 0.0221 | 0.0154 | $-$ | $-$ | $-$ |
| 128 | 0.1015 | 0.0209 | 0.0114 | 0.0065 | 0.0038 | $-$ | $-$ |
| 256 | 0.0613 | 0.0146 | 0.0078 | 0.0039 | 0.0022 | $-$ | $-$ |
| 512 | 0.0386 | 0.0111 | 0.0060 | 0.0030 | 0.0019 | 0.0007 | $-$ |
| 1280 | 0.0262 | 0.0078 | 0.0039 | 0.0018 | 0.0013 | 0.0010 | 0.0005 |

Table 8: Mean $|\widehat{\xi}_t|$ for **1.5B parameter** model decreases as we increase the sequence length $T$. Conversely, for a fixed sequence length $T$, mean $|\widehat{\xi}_t|$ decreases with prefix length $t$. "$-$" indicates that the entry is not meaningful because the prefix length $t$ is more than the sequence length.

| $T$ | $t = 1$ | $t = 5$ | $t = 10$ | $t = 30$ | $t = 100$ | $t = 300$ | $t = 640$ |
|---|---|---|---|---|---|---|---|
| 64 | 0.345 | 0.135 | 0.108 | 0.087 | $-$ | $-$ | $-$ |
| 128 | 0.267 | 0.102 | 0.076 | 0.058 | 0.042 | $-$ | $-$ |
| 256 | 0.194 | 0.085 | 0.062 | 0.045 | 0.033 | $-$ | $-$ |
| 512 | 0.138 | 0.074 | 0.055 | 0.040 | 0.032 | 0.019 | $-$ |
| 1280 | 0.113 | 0.061 | 0.043 | 0.032 | 0.028 | 0.024 | 0.017 |

Table 9: Mean $\widehat{V}_t$ for **1.5B parameter** model decreases as we increase the sequence length $T$. Conversely, for a fixed sequence length $T$, mean $V_t$ decreases with prefix length $t$. "$-$" indicates that the entry is not meaningful because the prefix length $t$ is more than the sequence length.

| $T$ | $t = 1$ | $t = 5$ | $t = 10$ | $t = 30$ | $t = 100$ | $t = 300$ | $t = 640$ |
|---|---|---|---|---|---|---|---|
| 64 | 0.1675 | 0.0366 | 0.0229 | 0.0154 | $-$ | $-$ | $-$ |
| 128 | 0.1082 | 0.0221 | 0.0120 | 0.0066 | 0.0038 | $-$ | $-$ |
| 256 | 0.0665 | 0.0156 | 0.0083 | 0.0041 | 0.0022 | $-$ | $-$ |
| 512 | 0.0422 | 0.0120 | 0.0065 | 0.0032 | 0.0020 | 0.0007 | $-$ |
| 1280 | 0.0287 | 0.0085 | 0.0042 | 0.0020 | 0.0014 | 0.0011 | 0.0005 |

## D  KD CAN IMPROVE GENERALIZATION VIA VARIANCE REDUCTION

Here, we leverage our novel risk bounds to provide a theoretical justification for why KD can result in better generalization behavior compared to standard pre-training. In particular, we will focus on our bound in Theorem 3.4.[*] Note that, besides $|\Theta|$, $N$, and $T$ which are independent of the underlying training approach, there are three key quantities that dictate the generalization gap: (1) $\sum_t V_t$ which is related to the loss variance; (2) $C$ which is related to the extreme values that loss can take; and (3) the divergence between the teacher-provided next-token predictive distribution and the ground truth next-token distribution:

$$\text{DIV}(\boldsymbol{\zeta}, \omega) := \omega \cdot \sum_{t=1}^{T} \mathbb{E}\left[\text{D}_{\text{TV}}\left(P_{\boldsymbol{\zeta}^\rho}(\cdot|\mathbf{x}_{\leq t-1}), \mathcal{D}(\cdot|\mathbf{x}_{\leq t-1})\right)\right].$$

Note that, under Assumption 3.1, both KD and standard pre-training loss terms are bounded by $M$, allowing us to provide the same $C$ (as a function of $M$ and $T$) for *both* KD and standard pre-training. Thus, we focus on the remaining two terms which relate to $\sum_t V_t$ and $\text{DIV}(\boldsymbol{\zeta}, \omega)$.

Note that standard pre-training, i.e., training without KD, corresponds to $\omega = 0$, which leads to $\text{DIV}(\boldsymbol{\zeta}, \omega = 0) = 0$. In contrast, with $\omega > 0$, KD would incur a non-zero value for $\text{DIV}(\boldsymbol{\zeta}, \omega)$. On the other hand, as we will argue next, KD can lead to smaller value of the variance term $\sum_t V_t$. Thus, as long as the underlying teacher LM approximates the true next-token distribution well enough, it can lead to improved (student) performance or equivalently smaller generalization gap by striking a balance between the divergence (or bias) $\text{DIV}(\boldsymbol{\zeta}, \omega)$ and variance $\sum_t V_t$; as a result, realizing a form of *bias vs. variance* trade-off for LM pre-training.

The variance reduction in the case of KD is the cleanest to observe by focusing on the last summand in $\sum_t V_t$, i.e., $V_T$. Towards this, recall from Assumption 3.3 that, for each $\boldsymbol{\theta} \in \Theta$, $V_T$ bounds the second-order moment of $\xi_T(\mathbf{x}; \boldsymbol{\theta})$. Define the short-hand

$$P_{\boldsymbol{\zeta}^\rho}^{(x_t, \omega)}(\cdot|\mathbf{x}_{\leq t-1}) := (1 - \omega) \cdot \mathbb{1}_{x_t}(\cdot) + \omega \cdot P_{\boldsymbol{\zeta}^\rho}(\cdot|\mathbf{x}_{\leq t-1}) \tag{67}$$

and write

$$\xi_T(\mathbf{x}; \boldsymbol{\theta}) = \mathbb{E}\left[\ell^\omega(\mathbf{x}; \boldsymbol{\theta})|\mathbf{x}_{\leq T-1}\right] - \mathbb{E}\left[\ell^\omega(\mathbf{x}; \boldsymbol{\theta})|\mathbf{x}_{\leq T}\right]$$

$$= \mathbb{E}\left[\frac{1}{T}\sum_{t=1}^{T} \text{CE}\left(P_{\boldsymbol{\zeta}^\rho}^{(x_t, \omega)}(\cdot|\mathbf{x}_{\leq t-1}), P_{\boldsymbol{\theta}}(\cdot|\mathbf{x}_{\leq t-1})\right)|\mathbf{x}_{\leq T-1}\right] -$$

$$\mathbb{E}\left[\frac{1}{T}\sum_{t=1}^{T} \text{CE}\left(P_{\boldsymbol{\zeta}^\rho}^{(x_t, \omega)}(\cdot|\mathbf{x}_{\leq t-1}), P_{\boldsymbol{\theta}}(\cdot|\mathbf{x}_{\leq t-1})\right)|\mathbf{x}_{\leq T}\right]$$

$$\overset{(i)}{=} \frac{1}{T}\sum_{t=1}^{T-1} \text{CE}\left(P_{\boldsymbol{\zeta}^\rho}^{(x_t, \omega)}(\cdot|\mathbf{x}_{\leq t-1}), P_{\boldsymbol{\theta}}(\cdot|\mathbf{x}_{\leq t-1})\right) + \mathbb{E}\left[\frac{1}{T}\text{CE}\left(P_{\boldsymbol{\zeta}^\rho}^{(x_T, \omega)}(\cdot|\mathbf{x}_{\leq T-1}), P_{\boldsymbol{\theta}}(\cdot|\mathbf{x}_{\leq T-1})\right)|\mathbf{x}_{\leq T-1}\right]$$

$$- \frac{1}{T}\sum_{t=1}^{T} \text{CE}\left(P_{\boldsymbol{\zeta}^\rho}^{(x_t, \omega)}(\cdot|\mathbf{x}_{\leq t-1}), P_{\boldsymbol{\theta}}(\cdot|\mathbf{x}_{\leq t-1})\right)$$

$$\overset{(ii)}{=} \mathbb{E}\left[\frac{1}{T}\text{CE}\left(P_{\boldsymbol{\zeta}^\rho}^{(x_T, \omega)}(\cdot|\mathbf{x}_{\leq T-1}), P_{\boldsymbol{\theta}}(\cdot|\mathbf{x}_{\leq T-1})\right)|\mathbf{x}_{\leq T-1}\right] - \frac{1}{T}\text{CE}\left(P_{\boldsymbol{\zeta}^\rho}^{(x_T, \omega)}(\cdot|\mathbf{x}_{\leq T-1}), P_{\boldsymbol{\theta}}(\cdot|\mathbf{x}_{\leq t-1})\right)$$

$$= (1 - \omega) \cdot \left(\mathbb{E}\left[-\frac{1}{T}\cdot \log P_{\boldsymbol{\theta}}(x_T|\mathbf{x}_{\leq t-1})|\mathbf{x}_{\leq T-1}\right] + \frac{1}{T}\cdot \log P_{\boldsymbol{\theta}}(x_T|\mathbf{x}_{\leq T-1})\right) \tag{68}$$

where $(i)$ follows we can remove expectation for those terms that are functions of those random variables that we condition on; and $(ii)$ follows by removing the terms that cancel each other; and the last line follows as we have

$$P_{\boldsymbol{\zeta}^\rho}^{(x_T, \omega)}(\cdot|\mathbf{x}_{\leq T-1}) = (1 - \omega) \cdot \mathbb{1}_{x_T}(\cdot) + \omega \cdot P_{\boldsymbol{\zeta}^\rho}(\cdot|\mathbf{x}_{\leq T-1}).$$

---

[*]One could draw similar conclusion from Theorem 3.2 by extending the arguments from Menon et al. (2021) to the language modeling setting.

It follows from (68) that, for any $\boldsymbol{\theta} \in \Theta$, we have

$$\mathbb{E}\left[\xi_T^2(\mathbf{x}; \boldsymbol{\theta}, \boldsymbol{\zeta})\right] = (1 - \omega) \cdot \operatorname{Var}\left[\frac{1}{T} \cdot \log P_{\boldsymbol{\theta}}(x_T | \mathbf{x}_{\leq t-1}) \, \Big| \, \mathbf{x}_{\leq T-1}\right], \tag{69}$$

where $\operatorname{Var}\left[\cdot | \cdot\right]$ denotes conditional variance. Note that (69) shows that $V_T$ decreases with $\omega$ in $[0, 1]$. This highlights that KD, i.e., $\omega > 0$, would realize a smaller variance than standard pre-training, i.e., $\omega = 0$. Thus, to realize improved generalization via KD, one needs to select the distillation weight $\omega$ so that the *variance reduction* via KD offsets the divergence term $\operatorname{DIV}(\boldsymbol{\zeta}, \omega)$. In particular, when the teacher LM approximates the ground truth next-token distribution very well, i.e., $\operatorname{DIV}(\boldsymbol{\zeta}, \omega)$ term is small even for a relatively large value of $\omega$, the variance reduction via KD becomes prominent, ensuring significant improvement over standard pre-training in terms of generalization performance.

## E  EXPERIMENTAL SETUP DETAILS

**Model architectures.** We work with standard decoder-only Transformer-based LMs. In our experimental evaluations, we collectively explored student LMs with 1.5B, 2.8B and 8.6B parameters. As for the small teacher LMs, our results covered LMs with 0.5B, 1.5B, and 2.8B parameters. Table 10 presents the key design parameters regarding the architecture of all these models. Note that, through our experiments we cover a wide range of model architecture, including deep & narrow models, shallow & wide models, multi-head attention (Vaswani et al., 2017), and multi-query attention (Shazeer, 2019). This demonstrates that the proposed SALT method is robust to the underlying model architecture.

Table 10: Model architecture parameters.

|  | Model size | | | |
| --- | --- | --- | --- | --- |
|  | 0.5B | 1.5B | 2.8B | 8.6B |
| **Number of Layers** | 32 | 44 | 92 | 32 |
| **Model dimension** | 768 | 1024 | 1024 | 4096 |
| **MLP hidden dimension** | 3072 | 8192 | 8192 | 16384 |
| **Number of attention heads** | 6 | 4 | 4 | 32 |
| **Attention type** | Multi-head | Multi-head | Multi-query | Multi-query |

We use a SentencePiece tokenizer (Kudo & Richardson, 2018) from Du et al. (2022) with a vocabulary size of $256K$. Similar vocabulary size was also utilized in (Chowdhery et al., 2022; Team et al., 2025). We employ weight tying (Press & Wolf, 2017), i.e., the same vocabulary embedding parameters are used for input token embedding and output embedding layers.

**Pre-training data.** We pre-train all LMs on ThePile dataset (Gao et al., 2020) by minimizing the UL2 objective (Tay et al., 2023) with a mixture of four tasks: (1) *causal LM* task; (2) *prefix LM* task with mean prefix length of $1/4$th the sequence length, (3) *span corruption task* with $r = 15\%$ of the tokens corrupted and mean corrupted span length $\mu = 3$; and (4) *span corruption task* with $r = 50\%$ of the tokens corrupted and mean corrupted span length $\mu = 32$. The four tasks are mixed at a ratio of 6:2:1:1.

**Training setup.** We pre-train LMs for approximately 545 billion tokens, with a batch size of 2048 and input sequence length of 1280. This translates to a little over two epochs on ThePile data. As for the optimization method, we utilize Adafactor algorithm (Shazeer & Stern, 2018), as per (Chowdhery et al., 2022; Du et al., 2022). We use a cosine learning rate decay schedule with a peak learning rate of 0.001, 4000 warmup steps and final learning rate of 0.0001. Training is done on 1024 TPU-v5e chips with JAX (Bradbury et al., 2018) and SeqIO (Roberts et al., 2022).

## F  FEW-SHOT EVALUATION TASKS AND METRICS

We performed a comprehensive few-shot evaluation of pre-trained LMs on 28 benchmarks. Below, we list these by organizing them according to the corresponding domain.

**World Knowledge**: NQ-Open (Lee et al., 2019), TriviaQA (Joshi et al., 2017), TyDiQA-NoContext (English)(Clark et al., 2020), Web Questions (Berant et al., 2013).

**Reading Comprehension:** RACE-M, RACE-H (Lai et al., 2017), SQuADv2 (Lee et al., 2020), TyDiQA-GoldP (English)(Clark et al., 2020).

**Commonsense Reasoning:** ARC (Easy) and ARC (Challenge) (Clark et al., 2018), HellaSwag (Zellers et al., 2019), OpenBookQA (Mihaylov et al., 2018), PiQA (Bisk et al., 2020), StoryCloze (Mostafazadeh et al., 2016), Winogrande (Sakaguchi et al., 2020).

**SuperGLUE (Wang et al., 2019):** BoolQ (Clark et al., 2019), CB (de Marneffe et al., 2019), COPA (Gordon et al., 2012), RTE (Dagan et al., 2006), WiC (Pilehvar & Camacho-Collados, 2018), WSC (Levesque et al., 2012), MultiRC (Khashabi et al., 2018), ReCoRD (Zhang et al., 2018).

**Natural Language Generation (NLG):** English portions of the three benchmarks – XLSum (Hasan et al., 2021), XSum (Narayan et al., 2018) and WikiLingua (Ladhak et al., 2020).

**Open-ended Cloze task:** LAMBADA Paperno et al. (2016).

**Code generation:** Mostly Basic Python Problems (MBPP) (Austin et al., 2021).

In our exploration, we conduct 1-shot evaluation for all the above benchmarks, except for MBPP which is 3-shot.

For MBPP, the metric is the fraction of success ignoring challenge problems. For each benchmark, we typically report the corresponding prevalent metric in the literature. For TyDiQA benchmarks, we report the F1 score as opposed to *EM* as it is the primary metric in (Clark et al., 2020). For MultiRC in SuperGLUE, we report F1 metric as per (Du et al., 2022).

## G    HYPERPARAMETER SEARCH FOR DISTILLATION WEIGHT AND TEACHER TEMPERATURE

In the section, we conduct a search to identify two key hyperparameters related to the knowledge distillation process, namely distillation weight $\omega$ and teacher temperature $\rho$ (cf. (2) and (3)). Given the resource extensive nature of pre-training experiments, we perform the hyperparameter search with relatively smaller scale experiments. In particular, we employed a 0.5B teacher LM to train a larger student LM with 1.5B parameters. We train the 0.5B teacher with configuration specified in Appendix E. For each $(\omega, \rho)$ pair, we train the student LM for 40K steps with a batch size of 1024 and sequence length of 1280, amounting to $\sim$52B tokens. As for the evaluation criterion, we track the average few-shot performance on a smaller set of evaluation tasks, namely NQ-Open, TriviaQA, Web Questions, SQuADv2, LAMBADA, BoolQ, CB, COPA, RTE, WiC, WSC, MultiRC, ReCoRD, and HellaSwag. Table 11 shows the average 1-shot performance for different choices of $(\omega, \rho)$ as the training progresses.

Table 11: Average 1-shot performance as training progresses for different values of distillation weight $\omega$ and teacher temperature $\rho$.

| | | Evaluation step | | | | | |
|---|---|---|---|---|---|---|---|
| $\omega$ | $\rho$ | 8000 | 12000 | 16000 | 20000 | 24000 | 40000 |
| | 0.1 | 36.90 | 38.28 | 39.83 | 39.86 | 40.78 | 41.69 |
| | 0.25 | 37.43 | 39.05 | 40.36 | 40.73 | 41.01 | 41.96 |
| 0.333 | 1 | 37.82 | 39.21 | 38.86 | 40.66 | **41.57** | 42.04 |
| | 2 | 37.84 | 39.64 | 38.79 | 41.00 | **41.57** | 42.12 |
| | 0.1 | 37.59 | 39.32 | 40.13 | 39.64 | 40.90 | 41.92 |
| | 0.25 | **39.43** | **39.73** | **40.50** | **41.02** | 41.56 | 41.87 |
| 0.667 | 1 | 38.53 | 39.49 | 40.27 | 40.76 | 40.97 | 41.83 |
| | 2 | 38.28 | 39.39 | 40.17 | 39.96 | 41.05 | **42.38** |
| | 0.1 | 37.01 | 38.05 | 38.47 | 39.14 | 38.92 | 39.46 |
| | 0.25 | 37.94 | 38.18 | 39.10 | 39.38 | 39.68 | 40.39 |
| 1 | 1 | 38.88 | 39.60 | 39.03 | 39.64 | 40.72 | 40.03 |
| | 2 | 38.18 | 39.38 | 38.84 | 38.16 | 39.68 | 39.63 |

Note that $\omega = 0.667$ and $\rho = 0.25$ ensure the best performance during most of the training. Based on this observation and the fact that the proposed SALT method (cf. Algorithm 1) employs KD only during the early phase of pre-training, we work with $\omega = 0.667$ and $\rho = 0.25$ in our empirical exploration of SALT. As evident from our experimental results (cf. §5), SALT with this choice of hyperparameters consistently leads to both improved quality and efficiency for different teacher-student pairs and longer pre-training. This suggests the robustness of SALT with a particular choice of $\omega$ and $\rho$.

## H   ADDITIONAL FEW-SHOT EVALUATION RESULTS FOR 2.8B LMS

Table 12 is an expansion of Table 2 in the main text. All evaluations are 1-shot, except for MBPP which is 3-shot. In the metric column, *EM*, *Acc*, and *Rg2* are abbreviations for *Exact Match*, *Accuracy*, and *Rouge2*, respectively.

Table 12: **Comprehensive few-shot performance of 2.8B pre-trained LMs.** A 1.5B SLM serves as the teacher LM for SALT & SALT$_{DS}$ during the KD phase of their pre-training and for RKD throughout its pre-training. BASELINE employs standard pre-training *without* KD from SLM. SALT and SALT$_{DS}$ already outperform BASELINE in terms of average few-shot performance at 70% of their training step budget, thereby improving both training efficiency and model quality. RKD, i.e., naively preforming KD from the small model through the pre-training, performs much worse than BASELINE. The best and second-best results for each domain are **boldfaced** and underlined, respectively.

| Domain | Dataset | Metric | SLM | BASELINE | RKD | SALT | | SALT$_{DS}$ | |
|---|---|---|---|---|---|---|---|---|---|
| | | | | @100% steps | @100% steps | @70% steps | @100% steps | @70% steps | @100% steps |
| **World Knowledge** | NaturalQuestions-Open | *EM* | 5.90 | 8.70 | 6.70 | 9.40 | **10.10** | 8.40 | 9.00 |
| | TriviaQA | *EM* | 30.09 | 43.15 | 34.87 | 39.87 | **43.71** | 39.37 | 41.27 |
| | TyDiQA-NoContext | *F1* | 22.20 | **28.20** | 26.10 | 27.90 | 27.10 | 25.90 | 27.20 |
| | WebQuestions | *EM* | 5.40 | 8.70 | 7.10 | 9.20 | **9.90** | 8.90 | 9.40 |
| | **Domain average** | | 15.90 | 22.19 | 18.69 | 21.59 | **22.70** | 20.64 | 21.72 |
| **Reading Comprehension** | RACE-M | *Acc* | 52.60 | 57.00 | 54.00 | 58.60 | **58.90** | 57.90 | 58.40 |
| | RACE-H | *Acc* | 37.50 | **42.30** | 39.70 | 42.20 | **42.30** | 42.10 | **42.30** |
| | SQuADv2 | *EM* | 43.30 | 54.80 | 50.90 | 54.60 | 55.90 | 57.60 | **57.90** |
| | TyDiQA-GoldP | *F1* | 51.80 | 57.90 | 59.40 | 58.80 | **61.10** | 59.80 | **61.10** |
| | **Domain average** | | 46.30 | 53.00 | 51.00 | 53.55 | 54.55 | 54.35 | **54.93** |
| **Commonsense Reasoning** | ARC-E | *Acc* | 64.60 | 68.40 | 66.00 | 67.60 | 67.60 | **69.40** | 69.00 |
| | ARC-C | *Acc* | 32.40 | 37.10 | 33.70 | 38.00 | **38.40** | 38.10 | 37.30 |
| | HellaSwag | *Acc* | 56.00 | 62.80 | 56.20 | 62.00 | 63.30 | 63.10 | **63.80** |
| | OpenBookQA | *Acc* | 48.00 | **50.00** | 45.80 | 47.20 | 48.20 | 47.60 | 48.20 |
| | PiQA | *Acc* | 72.00 | **75.40** | 72.60 | 73.20 | 73.70 | 74.10 | 73.90 |
| | StoryCloze | *Acc* | 73.10 | **77.20** | 73.70 | 76.90 | 76.80 | 77.00 | 77.10 |
| | WinoGrande | *Acc* | 58.20 | 63.00 | 60.10 | 64.00 | 63.70 | 64.70 | **65.40** |
| | **Domain average** | | 57.76 | 61.99 | 58.30 | 61.27 | 61.67 | 62.00 | **62.10** |
| | LAMBADA | *Acc* | 26.90 | 36.20 | 31.10 | 50.70 | 48.30 | 48.00 | **53.00** |
| **SuperGLUE** | BoolQ | *Acc* | 63.40 | 64.30 | 62.50 | 64.10 | 62.30 | **65.50** | 64.30 |
| | CB | *Acc* | 37.50 | 58.90 | 50.00 | **60.70** | 53.60 | 55.40 | 53.60 |
| | COPA | *Acc* | 77.00 | 79.00 | 71.00 | 76.00 | 77.00 | **81.00** | 77.00 |
| | MultiRC | *F1* | 53.80 | 54.20 | 53.50 | 57.50 | **58.60** | 50.70 | 53.00 |
| | RTE | *Acc* | 55.20 | 55.60 | **59.90** | 57.80 | 58.50 | 54.20 | 58.50 |
| | ReCoRD | *Acc* | 84.80 | 87.10 | 85.20 | 86.60 | 86.90 | 87.20 | **87.30** |
| | WiC | *Acc* | 48.40 | 47.20 | 47.20 | 49.80 | 48.10 | 50.00 | **50.90** |
| | WSC | *Acc* | 72.60 | 77.90 | 74.00 | 77.90 | 77.20 | **83.90** | 80.00 |
| | **Domain average** | | 61.59 | 65.53 | 62.91 | **66.30** | 65.28 | 65.99 | 65.58 |
| **NLG** | *GEM*-XLSum | *Rg2* | 2.80 | 4.10 | 3.40 | 4.40 | 4.40 | **4.60** | **4.60** |
| | *GEM*-XSum | *Rg2* | 2.80 | 5.10 | 3.20 | 5.00 | 5.10 | **5.40** | **5.40** |
| | WikiLingua | *Rg2* | 3.80 | 4.60 | 3.60 | 4.50 | **4.70** | 4.40 | 4.50 |
| | **Domain average** | | 3.13 | 4.60 | 3.40 | 4.63 | 4.73 | 4.80 | **4.83** |
| | MBPP | *Acc* | 9.60 | 16.20 | 11.40 | 15.60 | 17.00 | 16.60 | **17.80** |
| | **Average (28 tasks)** | | 42.56 | 47.32 | 44.39 | 47.86 | 47.94 | 47.89 | **48.26** |

# I  RESULTS FOR 8.6B PARAMETER LM PRE-TRAINING

In order to further validate the utility of the proposed SALT framework, we utilize it to train a larger LM. In particular, we train a 8.6B parameter LM on the Pile dataset with the help of a 2.8B parameter small LM via SALT with $n_{KD} = 36K$. In addition, we also explore $SALT_{DS}$ in this setting where, as per the discussion in §4, we utilize an early checkpoint (corresponding to $n_0 = 26K$ steps) of the 2.8B parameter model for data selection with $k = 10$ in (10).

Appendix I.1 and Appendix I.2 present the few-shot performance and post-SFT performance, respectively, for the 8.6B LMs trained via SALT and $SALT_{DS}$ while contrasting those with the performance of the natural baseline – an 8.6B LM trained via the standard pre-training approach. See §5.3 for more details about the SFT procedure.

Finally, we also explore the utility of SALT as we scale the student-teacher size ration. Appendix I.3 and I.4 present few-shot evaluations for the 8.6B LMs trained via 1.5B ($n_{KD} = 14K$) and 0.5B ($n_{KD} = 7K$) SLM teachers, respectively.

## I.1  FEW-SHOT EVALUATIONS FOR 8.6B MODELS PRE-TRAINED VIA 2.8B SLM TEACHER

Please see Table 16 for domain-wise few-shot performance results and Table 17 for the full few-shot performance results.

Table 13: **Domain-wise few-shot performance of pre-trained 8.6B parameter LMs.** SALT and $SALT_{DS}$ utilize **a 2.8B parameter** SLM during their pre-training. Note that SALT and $SALT_{DS}$ already outperform BASELINE in terms of average few-shot performance at 70% of their training step budget, thereby improving both training efficiency and model quality. The best and second-best results for each domain are **boldfaced** and underlined, respectively.

| Domain | # Tasks | SLM | BASELINE | SALT | | $SALT_{DS}$ | |
|---|---|---|---|---|---|---|---|
| | | | @100% steps | @70% steps | @100% steps | @70% steps | @100% steps |
| **World Knowledge** | 4 | 22.19 | 26.91 | 27.66 | **28.97** | 28.04 | 28.47 |
| **Reading Comprehension** | 4 | 53.00 | 56.40 | 56.83 | 57.42 | 56.10 | **57.48** |
| **Commonsense Reasoning** | 7 | 61.99 | 66.01 | 66.89 | 67.09 | 66.61 | **67.24** |
| **LAMBADA** | 1 | 36.20 | 58.70 | **65.50** | 64.80 | 54.30 | 55.00 |
| **SuperGLUE** | 8 | 65.53 | 69.69 | 69.19 | 70.38 | 71.06 | **71.26** |
| **NLG** | 3 | 4.60 | 5.40 | **5.97** | **5.97** | 5.23 | 5.30 |
| **MBPP** | 1 | 16.20 | 20.80 | 19.80 | 22.00 | 22.80 | **23.20** |
| **Average** | 28 | 47.32 | 51.73 | 52.24 | **52.96** | 52.29 | 52.81 |

Table 14: **Comprehensive few-shot performance of pre-trained 8.6B parameter LMs.** SLM is a **2.8B parameter** model that serves as the teacher LM for SALT & SALT$_{DS}$ during the KD phase of their pre-training and for RKD throughout its pre-training. BASELINE employs standard pre-training *without* KD from SLM. SALT and SALT$_{DS}$ already outperform BASELINE in terms of average few-shot performance at 70% of their training step budget, thereby improving both training efficiency and model quality. The best and second-best results for each domain are **boldfaced** and underlined, respectively.

| Domain | Dataset | Metric | SLM | BASELINE @100% steps | SALT @70% steps | SALT @100% steps | SALT$_{DS}$ @70% steps | SALT$_{DS}$ @100% steps |
|---|---|---|---|---|---|---|---|---|
| **World Knowledge** | NaturalQuestions-Open | *EM* | 8.70 | 10.50 | 11.80 | 11.50 | 12.00 | **12.30** |
| | TriviaQA | *EM* | 43.15 | 54.86 | 55.16 | 57.07 | 56.85 | **58.99** |
| | TyDiQA-NoContext | *F1* | 28.20 | 30.40 | 29.70 | **32.30** | 32.00 | 30.60 |
| | WebQuestions | *EM* | 8.70 | 11.90 | 14.00 | **15.00** | 11.30 | 12.00 |
| | **Domain average** | | 22.19 | 26.91 | 27.66 | **28.97** | 28.04 | 28.47 |
| **Reading Comprehension** | RACE-M | *Acc* | 57.00 | 60.70 | 61.70 | **62.40** | 59.60 | 60.70 |
| | RACE-H | *Acc* | 42.30 | 45.40 | 44.90 | **45.70** | 43.30 | 44.20 |
| | SQuADv2 | *EM* | 54.80 | 61.20 | 56.50 | 57.00 | 56.90 | **61.50** |
| | TyDiQA-GoldP | *F1* | 57.90 | 58.30 | 64.20 | **64.60** | **64.60** | 63.50 |
| | **Domain average** | | 53.00 | 56.40 | 56.83 | 57.42 | 56.10 | **57.48** |
| **Commonsense Reasoning** | ARC-E | *Acc* | 68.40 | 73.30 | 74.10 | **74.60** | 73.00 | 74.00 |
| | ARC-C | *Acc* | 37.10 | 42.70 | 45.00 | **46.20** | 43.90 | 44.50 |
| | HellaSwag | *Acc* | 62.80 | 70.40 | 70.00 | 70.80 | 70.70 | **71.60** |
| | OpenBookQA | *Acc* | 50.00 | 51.20 | **53.40** | 53.20 | 53.00 | 52.80 |
| | PiQA | *Acc* | 75.40 | 77.30 | 76.60 | 76.40 | 76.80 | **77.70** |
| | StoryCloze | *Acc* | 77.20 | 80.00 | 80.20 | 80.00 | 80.20 | **80.30** |
| | WinoGrande | *Acc* | 63.00 | 67.20 | 68.90 | 68.40 | 68.70 | **69.80** |
| | **Domain average** | | 61.99 | 66.01 | 66.89 | 67.09 | 66.61 | **67.24** |
| | LAMBADA | *Acc* | 36.20 | 58.70 | **65.50** | 64.80 | 54.30 | 55.00 |
| **SuperGLUE** | BoolQ | *Acc* | 64.30 | 70.70 | 74.20 | 74.70 | **76.80** | 76.00 |
| | CB | *Acc* | 58.90 | 60.70 | 53.60 | 58.90 | **64.30** | **64.30** |
| | COPA | *Acc* | 79.00 | **87.00** | **87.00** | 84.00 | 85.00 | 86.00 |
| | MultiRC | *F1* | 54.20 | 55.90 | 52.60 | 55.80 | 59.90 | **61.70** |
| | RTE | *Acc* | 55.60 | 61.40 | 64.60 | **67.10** | 62.50 | 62.50 |
| | ReCoRD | *Acc* | 87.10 | 89.20 | 89.40 | 89.30 | **89.50** | 89.20 |
| | WIC | *Acc* | 47.20 | **50.50** | 50.00 | 50.00 | 47.30 | 47.20 |
| | WSC | *Acc* | 77.90 | 82.10 | 82.10 | **83.20** | **83.20** | **83.20** |
| | **Domain average** | | 65.53 | 69.69 | 69.19 | 70.38 | 71.06 | **71.26** |
| **NLG** | GEM-XLSum | *Rg2* | 4.10 | 4.90 | 5.30 | **5.40** | 4.80 | 4.70 |
| | GEM-XSum | *Rg2* | 5.10 | 6.10 | **7.20** | 7.00 | 6.00 | 6.20 |
| | WikiLingua | *Rg2* | 4.60 | 5.20 | 5.40 | **5.50** | 4.90 | 5.00 |
| | **Domain average** | | 4.60 | 5.40 | **5.97** | **5.97** | 5.23 | 5.30 |
| | MBPP | *Acc* | 16.20 | 20.80 | 19.80 | 22.00 | 22.80 | **23.20** |
| | **Average (28 tasks)** | | 47.32 | 51.73 | 52.24 | **52.96** | 52.29 | 52.81 |

## I.2 POST SFT RESULTS FOR 8.6B MODELS PRE-TRAINED VIA 2.8B SLM TEACHER

Please see Table 15 for post-SFT performance of various 8.6B LMs.

Table 15: **Supervised fine-tuning (SFT) results.** Performance of various **8.6B pre-trained LMs** on downstream tasks after SFT. SALT and SALT$_{DS}$ employ an **2.8B SLM** as teacher. For each benchmark, pre-trained 8.6B models are fine-tuned on the corresponding train split and evaluated on the validation split (test split in case of GSM8K). *Acc*, *Rg1*, *Rg2*, and *RgL* represent the *Accuracy*, *Rouge-1*, *Rouge-2*, and *Rouge-Lsum* metrics, respectively.

| | GSM8K | XSum | | | CNN/DailyMail | | | ANLI-R1 | ANLI-R2 | ANLI-R3 |
|---|---|---|---|---|---|---|---|---|---|---|
| | *Acc* | *Rg1* | *Rg2* | *RgL* | *Rg1* | *Rg2* | *RgL* | *Acc* | *Acc* | *Acc* |
| BASELINE | 41.85 | 45.10 | 22.68 | 37.36 | 43.73 | 21.19 | 41.29 | 68.80 | 58.90 | 60.58 |
| SALT | **42.84** | 45.37 | 23.04 | 37.69 | 43.69 | 21.16 | 41.22 | **70.20** | 59.30 | **63.25** |
| SALT$_{DS}$ | 42.23 | **45.81** | **23.34** | **38.14** | **43.80** | **21.28** | **41.35** | 69.30 | **59.50** | 62.17 |

## I.3 FEW-SHOT EVALUATIONS FOR 8.6B MODELS PRE-TRAINED VIA 1.5B SLM TEACHER

Here we tabulate the results for the 8.6B model trained via SALT from a 1.5B teacher. Please see Table 16 for domain-wise few-shot performance results and Table 17 for the full few-shot performance results.

Table 16: **Domain-wise few-shot performance of 8.6B LMs (1.5B SLM teacher).** SALT already outperforms BASELINE in terms of average few-shot performance at 70% of their training step budget, thereby improving both training efficiency and model quality. The best and second-best results for each domain are **boldfaced** and underlined, respectively.

| Domain | # Tasks | SLM | BASELINE | SALT | |
|---|---|---|---|---|---|
| | | | *@100% steps* | *@70% steps* | *@100% steps* |
| **World Knowledge** | 4 | 15.90 | 26.91 | 27.96 | **28.57** |
| **Reading Comprehension** | 4 | 46.30 | 56.40 | 55.67 | **57.10** |
| **Commonsense Reasoning** | 7 | 57.76 | 66.01 | 66.39 | **66.73** |
| **LAMBADA** | 1 | 26.90 | 58.70 | **66.30** | 65.80 |
| **SuperGLUE** | 8 | 61.59 | 69.69 | 68.67 | **70.38** |
| **NLG** | 3 | 3.13 | **5.40** | 4.77 | 4.87 |
| **MBPP** | 1 | 9.60 | 20.80 | 21.60 | **25.20** |
| **Average** | 28 | 42.56 | 51.73 | 51.82 | **52.80** |

Table 17: **Comprehensive few-shot performance of 8.6B LMs (1.5B SLM teacher).** SLM serves as the teacher LM for SALT during the KD phase of their pre-training. BASELINE employs standard pre-training *without* KD from SLM. SALT outperforms BASELINE in terms of average few-shot performance at 70% of training step budget, thereby improving both training efficiency and model quality. The best and second-best results for each task are **boldfaced** and underlined, respectively.

| Domain | Dataset | Metric | SLM | BASELINE @100% steps | SALT @70% steps | SALT @100% steps |
|---|---|---|---|---|---|---|
| **World Knowledge** | NaturalQuestions-Open | *EM* | 5.90 | 10.50 | 11.50 | **12.10** |
| | TriviaQA | *EM* | 30.09 | 54.86 | 55.05 | **57.38** |
| | TyDiQA-NoContext | *F1* | 22.20 | 30.40 | 31.40 | **31.60** |
| | WebQuestions | *EM* | 5.40 | 11.90 | **13.90** | 13.20 |
| | **Domain average** | | 15.90 | 26.91 | 27.96 | **28.57** |
| **Reading Comprehension** | RACE-M | *Acc* | 52.60 | 60.70 | 61.00 | **61.40** |
| | RACE-H | *Acc* | 37.50 | **45.40** | 44.30 | 45.00 |
| | SQuADv2 | *EM* | 43.30 | **61.20** | 57.00 | 59.20 |
| | TyDiQA-GoldP | *F1* | 51.80 | 58.30 | 60.40 | **62.80** |
| | **Domain average** | | 46.30 | 56.40 | 55.67 | **57.10** |
| **Commonsense Reasoning** | ARC-E | *Acc* | 64.60 | 73.30 | 72.90 | **74.10** |
| | ARC-C | *Acc* | 32.40 | 42.70 | 44.80 | **45.30** |
| | HellaSwag | *Acc* | 56.00 | 70.40 | 69.70 | **70.70** |
| | OpenBookQA | *Acc* | 48.00 | 51.20 | 52.80 | **53.80** |
| | PiQA | *Acc* | 72.00 | 77.30 | **77.80** | 77.10 |
| | StoryCloze | *Acc* | 73.10 | **80.00** | 78.70 | 78.90 |
| | WinoGrande | *Acc* | 58.20 | 67.20 | **68.00** | 67.20 |
| | **Domain average** | | 57.76 | 66.01 | 66.39 | **66.73** |
| | LAMBADA | *Acc* | 26.90 | 58.70 | **66.30** | 65.80 |
| **SuperGLUE** | BoolQ | *Acc* | 63.40 | 70.70 | 72.60 | **74.80** |
| | CB | *Acc* | 37.50 | 60.70 | 55.40 | **64.30** |
| | COPA | *Acc* | 77.00 | **87.00** | 84.00 | **87.00** |
| | MultiRC | *F1* | 53.80 | 55.90 | 53.90 | **56.70** |
| | RTE | *Acc* | 55.20 | 61.40 | **63.20** | 62.50 |
| | ReCoRD | *Acc* | 84.80 | **89.20** | 89.10 | **89.20** |
| | WIC | *Acc* | 48.40 | **50.50** | 49.10 | 46.40 |
| | WSC | *Acc* | 72.60 | 82.10 | 82.10 | 82.10 |
| | **Domain average** | | 61.59 | 69.69 | 68.67 | **70.38** |
| **NLG** | GEM-XLSum | *Rg2* | 2.80 | **4.90** | 4.60 | 4.60 |
| | GEM-XSum | *Rg2* | 2.80 | **6.10** | 4.50 | 4.70 |
| | WikiLingua | *Rg2* | 3.80 | 5.20 | 5.20 | **5.30** |
| | **Domain average** | | 3.13 | **5.40** | 4.77 | 4.87 |
| | MBPP | *Acc* | 9.60 | 20.80 | 21.60 | **25.20** |
| | **Average (28 tasks)** | | 42.56 | 51.73 | 51.82 | **52.80** |

I.4 FEW-SHOT EVALUATIONS FOR 8.6B MODELS PRE-TRAINED VIA 0.5B SLM TEACHER

Here we tabulate the results for the 8.6B model trained via SALT with the help of a 0.5B teacher. Please see Table 18 for domain-wise few-shot performance results and Table 19 for the full few-shot performance results.

Table 18: **Domain-wise few-shot performance of 8.6B LMs (0.5B SLM teacher).** With $\sim17\times$ smaller teacher LM, SALT does not outperform the BASELINEon average few-shot performance. That said, it leads to improvement on 4 out 7 domains. The best and second-best results for each domain are **boldfaced** and underlined, respectively.

| Domain | # Tasks | SLM | BASELINE | SALT | |
| --- | --- | --- | --- | --- | --- |
| | | | *@100% steps* | *@70% steps* | *@100% steps* |
| **World Knowledge** | 4 | 8.18 | 26.91 | 26.96 | **28.01** |
| **Reading Comprehension** | 4 | 40.23 | 56.40 | 56.35 | **57.00** |
| **Commonsense Reasoning** | 7 | 50.26 | **66.01** | 65.99 | 65.94 |
| **LAMBADA** | 1 | 10.30 | 58.70 | 59.60 | **64.30** |
| **SuperGLUE** | 8 | 57.77 | **69.69** | 65.77 | 67.86 |
| **NLG** | 3 | 1.80 | **5.40** | 5.17 | 5.13 |
| **MBPP** | 1 | 3.60 | 20.80 | **22.60** | 21.00 |
| **Average** | 28 | 36.68 | **51.73** | 50.68 | 51.62 |

Table 19: **Comprehensive few-shot performance of 8.6B LMs (0.5B SLM teacher).** SLM serves as the teacher LM for SALTduring the KD phase of their pre-training.

| Domain | Dataset | Metric | SLM | BASELINE @100% steps | SALT @70% steps | SALT @100% steps |
|---|---|---|---|---|---|---|
| **World Knowledge** | NaturalQuestions-Open | *EM* | 2.40 | 10.50 | **11.70** | 11.10 |
| | TriviaQA | *EM* | 12.52 | 54.86 | 54.35 | **56.75** |
| | TyDiQA-NoContext | *F1* | 13.40 | 30.40 | 29.50 | **31.10** |
| | WebQuestions | *EM* | 4.40 | 11.90 | 12.30 | **13.10** |
| | **Domain average** | | 8.18 | 26.91 | 26.96 | **28.01** |
| **Reading Comprehension** | RACE-M | *Acc* | 44.20 | **60.70** | 60.00 | 60.00 |
| | RACE-H | *Acc* | 33.20 | **45.40** | 43.10 | 43.50 |
| | SQuADv2 | *EM* | 41.60 | 61.20 | 60.70 | **61.90** |
| | TyDiQA-GoldP | *F1* | 41.90 | 58.30 | 61.60 | **62.60** |
| | **Domain average** | | 40.23 | 56.40 | 56.35 | **57.00** |
| **Commonsense Reasoning** | ARC-E | *Acc* | 53.50 | **73.30** | 72.50 | 72.60 |
| | ARC-C | *Acc* | 28.40 | 42.70 | **43.30** | 42.70 |
| | HellaSwag | *Acc* | 42.40 | 70.40 | 69.90 | **70.70** |
| | OpenBookQA | *Acc* | 40.00 | 51.20 | 52.00 | **52.20** |
| | PiQA | *Acc* | 68.00 | **77.30** | 76.80 | 76.90 |
| | StoryCloze | *Acc* | 66.90 | **80.00** | 78.90 | 79.30 |
| | WinoGrande | *Acc* | 52.60 | 67.20 | **68.50** | 67.20 |
| | **Domain average** | | 50.26 | **66.01** | 65.99 | 65.94 |
| | LAMBADA | *Acc* | 10.30 | 58.70 | 59.60 | **64.30** |
| **SuperGLUE** | BoolQ | *Acc* | 51.70 | 70.70 | 70.30 | **71.30** |
| | CB | *Acc* | 42.90 | **60.70** | 33.90 | 41.10 |
| | COPA | *Acc* | 73.00 | **87.00** | 84.00 | 84.00 |
| | MultiRC | *F1* | 52.10 | 55.90 | 53.70 | **58.00** |
| | RTE | *Acc* | 49.10 | 61.40 | 59.20 | **65.00** |
| | ReCoRD | *Acc* | 78.70 | **89.20** | **89.20** | 89.10 |
| | WIC | *Acc* | **50.50** | **50.50** | 50.30 | 49.80 |
| | WSC | *Acc* | 64.20 | 82.10 | **85.60** | 84.60 |
| | **Domain average** | | 57.77 | **69.69** | 65.77 | 67.86 |
| **NLG** | GEM-XLSum | *Rg2* | 1.50 | 4.90 | **5.10** | 4.80 |
| | GEM-XSum | *Rg2* | 1.90 | **6.10** | 5.60 | 5.60 |
| | WikiLingua | *Rg2* | 2.00 | **5.20** | 4.80 | 5.00 |
| | **Domain average** | | 1.80 | **5.40** | 5.17 | 5.13 |
| | MBPP | *Acc* | 3.60 | 20.80 | **22.60** | 21.00 |
| | **Average (28 tasks)** | | 36.68 | **51.73** | 50.68 | 51.62 |

## J    WALL-CLOCK TIME SAVING VIA SALT

As evident in Tab. 2 and 13, SALT surpasses fully trained BASELINE after 146K training steps (i.e., 70% of the total step budget of 208K training steps) for both 2.8B and 8.6B LM pre-training. This suggests a saving of 30% training compute cost. That said, it is important to note that $n_{KD} = 36K$ of these 146K steps involve KD from a teacher SLM (cf. Algorithm 1). Each of these KD steps typically incurs higher computational cost compared to a single step of the standard pre-training as KD requires a forward pass of the teacher as an additional overhead. As we argue next, the fact that our teacher is a smaller LM ensures that we still realize training efficiency gains via SALT.

As a rule of thumb, depending on the application of gradient checkpointing/rematerialization Chen et al. (2016), the cost of a forward pass is $\alpha$ times the cost of a standard training step (that comprises both forward and backward passes) and such $\alpha \in [1/4, 1/3]$. In SALT, as the teacher is smaller than the student, the additional cost of teacher's forward pass is further smaller as a fraction of the student's standard step time (w/o KD). That said, the actual ratio between the teacher's forward pass time and student's standard training step time can depend on various implementation details.

For training a 2.8B LM by using a 1.5B teacher SLM, in our implementation on TPU-v5e chips, the wall-clock time of each KD step was $\sim 1.27$x that of a standard training step for 2.8B LM. Thus, the wall-clock time of 146K steps of SALT with $n_{KD} = 36K$ translates to a wall-clock time of approximately $(110 + 36 \times 1.27) = 155.7$K standard pre-training steps. Given that BASELINE has the training step budget of 208K steps, SALT realizes $\sim$ **25% savings in terms of wall-clock time** to surpass BASELINE while training a 2.8B LM with the help of a 1.5B teacher SLM.

As for training a 8.6B LM by using a 2.8B teacher SLM, a KD step incurs $\sim 1.12$x wall-clock time compared to a standard training step for 8.6B. Repeating the calculation given in the above paragraph, we see a $\sim$ **28% savings in terms of wall-clock time** to surpass BASELINE. With the 1.5B teacher, a KD step incurs $\sim 1.07$x time compared to a standard training step for 8.6B. Considering that the first stage of SALT is run for 14K steps, we get $\sim$ **29% savings** in terms of wall-clock time to surpass BASELINE. Please see Table 20 for a summary of the wall-clock time savings.

Table 20: **Wall-clock time savings with SALT.** The SALT trained student at 146K steps (70% training steps) surpasses BASELINE trained for 208K steps. We realize wall-clock time savings of 25-29%, after accounting for the additional cost of teacher forward pass during KD stage.

| Student size | Teacher size | $n_{KD}$ | KD step time / Regular step time | Wall-clock time savings to surpass BASELINE |
|---|---|---|---|---|
| 2.8B | 1.5B | 36K | 1.27 | 25% |
| 8.6B | 2.8B | 36K | 1.12 | 28% |
| 8.6B | 1.5B | 14K | 1.07 | 29% |

## K   ABLATION STUDY OF VARIOUS DESIGN CHOICES IN SALT

In this section, we explore how various design choices pertaining to SALT affect its final performance. Given the large cost of pre-training runs, we only conduct ablations for training 2.8B LMs with the help of 1.5B SLM teacher.

**Distillation from a better quality small model.** So far we assumed that SLM is also pre-trained for the same number of tokens as the LLM. Since training for SLM is relatively cheaper, one could consider a scenario where one invests more compute resources in improving the small model if it can eventually be beneficial in improving the LLM quality via SALT. Towards this, we employ a small LM that is trained for $\sim 2.5$ times longer – 498K steps vs. 208K steps in §5.2.[†] As evident in Table 21, SALT is indeed able to utilize the better small model as a teacher in the KD phase to further improve the LLM quality, as measured by the average few-shot performance.

**Varying transition point.** A key design choice for SALT is the selection of the transition point $n_{\mathrm{KD}}$ from KD phase (first stage) to standard training (second stage). Table 22 shows few-shot performance of SALT as we vary the transition point. Note that SALT ensures quality gains for LLM with a wide range of values for $n_{\mathrm{KD}}$ while demonstrating an inverted U-shape for LLM quality. We see consistent performance improvement from $n_{\mathrm{KD}} = 0$ (equivalent to BASELINE) to $n_{\mathrm{KD}} = 60$K which eventually degrades at $n_{\mathrm{KD}} = 208$K (equivalent to RKD). Given the training overhead of KD phase (see discussion in §5.2 and Appendix J), smaller value of $n_{\mathrm{KD}}$ helps ensure training efficiency gains via SALT. Thus, we worked with $n_{\mathrm{KD}} = 36$K in §5.2 as $n_{\mathrm{KD}} = 60$K only provides marginal quality gains if one takes into account the increased training cost due to longer KD phase.

**Different transition strategies.** In our study thus far, we have worked with *Step* transition between the two training stages in SALT where we abruptly stop performing KD after $n_{\mathrm{KD}}$ training steps. Looking at Figure 3, this causes an abrupt change in the model behavior during training, as observed in the next-token prediction accuracy curve for the training set (Figure 8 in Appendix L shows a similar behavior for log-perplexity). This raises a question if a smoother transition between the two stages can improve the training stability and thereby ensure higher final LLM quality. While there is a large space of potential choices of such smooth transition strategies, here we explore two natural candidates: (1) *Linear decay* where we linearly decrease the distillation loss weight to 0 between $n_{\mathrm{KD},1} = 32$K and $n_{\mathrm{KD},2} = 36$K steps; and (2) *Linear ratio decay* where we linearly decrease the ratio of distillation loss weight and standard loss weight $\frac{\omega}{1-\omega}$ to 0 between $n_{\mathrm{KD},1} = 32$K and $n_{\mathrm{KD},2} = 36$K training steps. As recorded in Table 23, the step transition constitutes a reasonable design choice for SALT as it outperforms both the considered alternatives in terms of average few-shot performance of the resulting pre-trained LLM.

---

[†]This approach aligns with the recent studies (Touvron et al., 2023; Gadre et al., 2024) that train small LMs well beyond the optimal compute budget predicted by neural scaling laws (Hoffmann et al., 2022).

Table 21: **Effect of improved** SLM **(comprehensive few-shot evaluation).** SALT with a better teacher – a 1.5B SLM trained for 498K steps as opposed to 208K steps – yields 2.8B LM with better average few-shot performance. For each benchmark, the best and second best results are **boldfaced** and underlined, respectively.

| Domain | Dataset | Metric | SLM trained for 208K steps | SLM trained for 498K steps | SALT w/ KD from SLM trained for 208K steps | SALT w/ KD from SLM trained for 498K steps |
|---|---|---|---|---|---|---|
| World Knowledge | NaturalQuestions-Open | *EM* | 5.90 | 6.30 | **10.10** | 9.00 |
| | TriviaQA | *EM* | 30.09 | 31.74 | **43.71** | 41.61 |
| | TyDiQA-NoContext | *F1* | 22.20 | 23.80 | **27.10** | 26.20 |
| | WebQuestions | *EM* | 5.40 | 7.60 | **9.90** | 9.10 |
| | **Domain average** | | 15.90 | 17.36 | **22.70** | 21.48 |
| Reading Comprehension | RACE-M | *Acc* | 52.60 | 54.40 | **58.90** | 57.00 |
| | RACE-H | *Acc* | 37.50 | 39.40 | **42.30** | 42.00 |
| | SQuADv2 | *EM* | 43.30 | 49.00 | 55.90 | **57.90** |
| | TyDiQA-GoldP | *F1* | 51.80 | 55.90 | **61.10** | 56.80 |
| | **Domain average** | | 46.30 | 49.67 | **54.55** | 53.43 |
| Commonsense Reasoning | ARC-E | *Acc* | 64.60 | 65.50 | 67.60 | **69.30** |
| | ARC-C | *Acc* | 32.40 | 34.30 | 38.40 | **39.10** |
| | HellaSwag | *Acc* | 56.00 | 57.80 | **63.30** | 63.20 |
| | OpenBookQA | *Acc* | 48.00 | 46.40 | 48.20 | **49.00** |
| | PiQA | *Acc* | 72.00 | 72.90 | 73.70 | **74.60** |
| | StoryCloze | *Acc* | 73.10 | 75.00 | 76.80 | **76.90** |
| | WinoGrande | *Acc* | 58.20 | 59.40 | 63.70 | **63.80** |
| | **Domain average** | | 57.76 | 58.76 | 61.67 | **62.27** |
| | LAMBADA | *Acc* | 26.90 | 37.80 | **48.30** | 47.80 |
| SuperGLUE | BoolQ | *Acc* | 63.40 | 61.40 | 62.30 | **65.80** |
| | CB | *Acc* | 37.50 | 42.90 | 53.60 | **73.20** |
| | COPA | *Acc* | 77.00 | 78.00 | 77.00 | **79.00** |
| | MultiRC | *F1* | 53.80 | 48.40 | **58.60** | 53.20 |
| | RTE | *Acc* | 55.20 | 52.30 | 58.50 | **61.70** |
| | ReCoRD | *Acc* | 84.80 | 85.50 | 86.90 | **87.10** |
| | WIC | *Acc* | 48.40 | 47.30 | 48.10 | **49.20** |
| | WSC | *Acc* | 72.60 | 72.30 | 77.20 | **79.30** |
| | **Domain average** | | 61.59 | 61.01 | 65.28 | **68.56** |
| NLG | *GEM*-XLSum | *Rg2* | 2.80 | 3.50 | **4.40** | 4.30 |
| | *GEM*-XSum | *Rg2* | 2.80 | 3.10 | 5.10 | **5.60** |
| | WikiLingua | *Rg2* | 3.80 | 3.80 | **4.70** | 4.40 |
| | **Domain average** | | 3.13 | 3.47 | 4.73 | **4.77** |
| | MBPP | *Acc* | 9.60 | 12.80 | 17.00 | **17.40** |
| | **Average (28 tasks)** | | 42.56 | 43.88 | 47.94 | **48.70** |

Table 22: **Effect of varying transitions step (comprehensive few-shot evaluation).** While training a 2.8B LM with the aid of a 1.5B SLM, the performance improvement via SALT over BASELINE is stable in a wide range of $n_{\text{KD}}$ (20k to 60k steps). Eventually, with much larger $n_{\text{KD}}$, SALT performance degrades significantly (208k steps). For each benchmark, the best and second best results are **boldfaced** and underlined, respectively.

| Domain | Dataset | Metric | SLM | BASELINE | SALT w/ $n_{\text{KD}}=20\text{K}$ | SALT w/ $n_{\text{KD}}=36\text{K}$ | SALT w/ $n_{\text{KD}}=60\text{K}$ | SALT w/ $n_{\text{KD}}=208\text{K}$ (RKD) |
|---|---|---|---|---|---|---|---|---|
| World Knowledge | NaturalQuestions-Open | *EM* | 5.90 | 8.70 | 8.90 | **10.10** | 9.30 | 6.70 |
| | TriviaQA | *EM* | 30.09 | 43.15 | 41.52 | **43.71** | 42.84 | 34.87 |
| | TyDiQA-NoContext | *F1* | 22.20 | **28.20** | 26.40 | 27.10 | 26.60 | 26.10 |
| | WebQuestions | *EM* | 5.40 | 8.70 | 8.20 | **9.90** | 8.60 | 7.10 |
| | **Domain average** | | 15.90 | 22.19 | 21.26 | **22.70** | 21.83 | 18.69 |
| Reading Comprehension | RACE-M | *Acc* | 52.60 | 57.00 | 58.70 | **58.90** | 58.60 | 54.00 |
| | RACE-H | *Acc* | 37.50 | **42.30** | 41.00 | **42.30** | 42.10 | 39.70 |
| | SQuADv2 | *EM* | 43.30 | 54.80 | 55.30 | **55.90** | 55.50 | 50.90 |
| | TyDiQA-GoldP | *F1* | 51.80 | 57.90 | 56.50 | **61.10** | 59.30 | 59.40 |
| | **Domain average** | | 46.30 | 53.00 | 52.88 | **54.55** | 53.88 | 51.00 |
| Commonsense Reasoning | ARC-E | *Acc* | 64.60 | **68.40** | 67.80 | 67.60 | **68.40** | 66.00 |
| | ARC-C | *Acc* | 32.40 | 37.10 | 38.10 | 38.40 | **38.70** | 33.70 |
| | HellaSwag | *Acc* | 56.00 | 62.80 | 62.80 | **63.30** | 62.90 | 56.20 |
| | OpenBookQA | *Acc* | 48.00 | **50.00** | 48.00 | 48.20 | 48.20 | 45.80 |
| | PiQA | *Acc* | 72.00 | **75.40** | **75.40** | 73.70 | 74.40 | 72.60 |
| | StoryCloze | *Acc* | 73.10 | **77.20** | 76.90 | 76.80 | 76.50 | 73.70 |
| | WinoGrande | *Acc* | 58.20 | 63.00 | 63.40 | **63.70** | 62.00 | 60.10 |
| | **Domain average** | | 57.76 | **61.99** | 61.77 | 61.67 | 61.59 | 58.30 |
| | LAMBADA | *Acc* | 26.90 | 36.20 | 44.70 | 48.30 | **53.30** | 31.10 |
| SuperGLUE | BoolQ | *Acc* | 63.40 | **64.30** | 63.90 | 62.30 | 63.80 | 62.50 |
| | CB | *Acc* | 37.50 | 58.90 | **60.70** | 53.60 | 55.40 | 50.00 |
| | COPA | *Acc* | 77.00 | **79.00** | 76.00 | 77.00 | 77.00 | 71.00 |
| | MultiRC | *F1* | 53.80 | 54.20 | 53.80 | **58.60** | 55.20 | 53.50 |
| | RTE | *Acc* | 55.20 | 55.60 | 52.30 | 58.50 | **59.90** | **59.90** |
| | ReCoRD | *Acc* | 84.80 | **87.10** | 86.90 | 86.90 | 86.70 | 85.20 |
| | WIC | *Acc* | 48.40 | 47.20 | **51.30** | 48.10 | 50.00 | 47.20 |
| | WSC | *Acc* | 72.60 | **77.90** | 77.50 | 77.20 | **77.90** | 74.00 |
| | **Domain average** | | 61.59 | 65.53 | 65.30 | 65.28 | **65.74** | 62.91 |
| NLG | *GEM*-XLSum | *Rg2* | 2.80 | 4.10 | 4.50 | 4.40 | **4.70** | 3.40 |
| | *GEM*-XSum | *Rg2* | 2.80 | 5.10 | **5.80** | 5.10 | 4.80 | 3.20 |
| | WikiLingua | *Rg2* | 3.80 | 4.60 | 4.30 | **4.70** | 4.60 | 3.60 |
| | **Domain average** | | 3.13 | 4.60 | **4.87** | 4.73 | 4.70 | 3.40 |
| | MBPP | *Acc* | 9.60 | 16.20 | 16.60 | **17.00** | 16.40 | 11.40 |
| | **Average (28 tasks)** | | 42.56 | 47.32 | 47.40 | 47.94 | **47.99** | 44.39 |

Table 23: **Effect of different transition strategies (comprehensive few-shot evaluation).** While training a 2.8B LM with the aid of a 1.5B SLM, the Step transition used in this work (cf. Algorithm 1) performs well compared to two natural alternative strategies, namely Linear decay and Linear ratio decay. For each benchmark, the best and second best results are **boldfaced** and underlined, respectively.

| Domain | Dataset | Metric | SLM | BASELINE | SALT w/ Step | SALT w/ Linear decay | SALT w/ Linear ratio decay |
|---|---|---|---|---|---|---|---|
| **World Knowledge** | NaturalQuestions-Open | *EM* | 5.90 | 8.70 | **10.10** | 8.20 | 8.10 |
| | TriviaQA | *EM* | 30.09 | 43.15 | **43.71** | 43.46 | 43.51 |
| | TyDiQA-NoContext | *F1* | 22.20 | 28.20 | 27.10 | **28.40** | 27.20 |
| | WebQuestions | *EM* | 5.40 | 8.70 | **9.90** | 8.20 | 8.40 |
| | **Domain average** | | 15.90 | 22.19 | **22.70** | 22.07 | 21.80 |
| **Reading Comprehension** | RACE-M | *Acc* | 52.60 | 57.00 | **58.90** | 57.90 | 57.40 |
| | RACE-H | *Acc* | 37.50 | 42.30 | 42.30 | 42.10 | **43.50** |
| | SQuADv2 | *EM* | 43.30 | 54.80 | 55.90 | 56.40 | **57.10** |
| | TyDiQA-GoldP | *F1* | 51.80 | 57.90 | **61.10** | 58.30 | 57.80 |
| | **Domain average** | | 46.30 | 53.00 | **54.55** | 53.68 | 53.95 |
| **Commonsense Reasoning** | ARC-E | *Acc* | 64.60 | 68.40 | 67.60 | 68.60 | **68.70** |
| | ARC-C | *Acc* | 32.40 | 37.10 | 38.40 | 38.60 | **39.80** |
| | HellaSwag | *Acc* | 56.00 | 62.80 | 63.30 | 63.30 | **63.50** |
| | OpenBookQA | *Acc* | 48.00 | **50.00** | 48.20 | 48.00 | 47.40 |
| | PiQA | *Acc* | 72.00 | **75.40** | 73.70 | 74.60 | 73.90 |
| | StoryCloze | *Acc* | 73.10 | **77.20** | 76.80 | 76.60 | 76.50 |
| | WinoGrande | *Acc* | 58.20 | 63.00 | **63.70** | 62.70 | 63.10 |
| | **Domain average** | | 57.76 | **61.99** | 61.67 | 61.77 | 61.84 |
| | LAMBADA | *Acc* | 26.90 | 36.20 | **48.30** | 40.50 | 42.60 |
| **SuperGLUE** | BoolQ | *Acc* | 63.40 | 64.30 | 62.30 | **67.90** | 66.50 |
| | CB | *Acc* | 37.50 | **58.90** | 53.60 | 44.60 | 46.40 |
| | COPA | *Acc* | 77.00 | 79.00 | 77.00 | 79.00 | **81.00** |
| | MultiRC | *F1* | 53.80 | 54.20 | 58.60 | 53.90 | **61.60** |
| | RTE | *Acc* | 55.20 | 55.60 | **58.50** | 56.30 | 55.20 |
| | ReCoRD | *Acc* | 84.80 | 87.10 | 86.90 | 87.00 | **87.30** |
| | WIC | *Acc* | 48.40 | 47.20 | 48.10 | 46.60 | **50.50** |
| | WSC | *Acc* | 72.60 | 77.90 | 77.20 | 78.60 | **78.90** |
| | **Domain average** | | 61.59 | 65.53 | 65.28 | 64.24 | **65.92** |
| **NLG** | *GEM*-XLSum | *Rg2* | 2.80 | 4.10 | 4.40 | **4.70** | 4.50 |
| | *GEM*-XSum | *Rg2* | 2.80 | **5.10** | **5.10** | 4.60 | **5.10** |
| | WikiLingua | *Rg2* | 3.80 | 4.60 | 4.70 | **4.80** | 4.60 |
| | **Domain average** | | 3.13 | 4.60 | **4.73** | 4.70 | **4.73** |
| | MBPP | *Acc* | 9.60 | 16.20 | **17.00** | 15.20 | **17.00** |
| **Average (28 tasks)** | | | 42.56 | 47.32 | **47.94** | 47.11 | 47.75 |

## L  LOG PERPLEXITY OF THE MODELS

For pre-training 2.8B LM, Figure 8 shows the log perplexity of the SALT and RKD pre-trained models along with BASELINE as the training progresses. We also provide the evolution of the log-perplexity of SLM during its training. Note that the log perplexity for RKD stays at a higher level than even SLM. Recall that RKD optimizes a sum of two losses – KD loss with weight $\omega = 0.667$ and the standard one-hot training loss with weight $1 - \omega$.

*Remark* L.1.  As the training log perplexity plotted in Figure 8 is the same as the standard hot training loss, the methods which directly optimize for that alone (BASELINE, SLM and in the second stage, SALT) have lower log perplexity on training set than RKD which optimizes additionally for distillation loss.

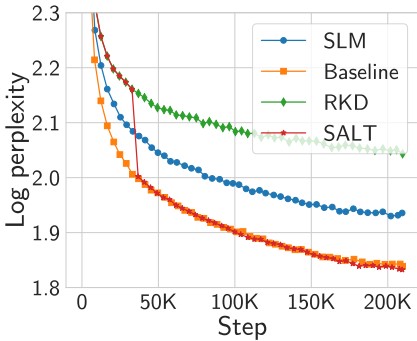

Figure 8: Log perplexity for different 2.8B LMs during their pre-training, as measured on a subset of the Pile training set. We also showcase the log-perplexity for 1.5B SLM (as its training progresses), which is utilized by SALT and RKD as a teacher.

## M  ADDITIONAL RESULTS: LEARNING EASY VS. HARD INSTANCE VIA SALT

**Creation of different hardness buckets.** For each evaluation benchmark, we first assign a relative rank to each test instance/example in the benchmark, representing its degree of difficulty. A test example with the lowest rank (easiest) is the one on which the small teacher LM achieves the largest task evaluation score, e.g., Rouge-2 metric for the XLSum task. Similarly, subsequent test examples are assigned ranks in descending order of the task evaluation score achieved by the small teacher LM. If two examples have the same evaluation score, the one with higher confidence score from the teacher (on its generated output) is deemed to have a lower rank. Each test example is assigned to one of the three buckets: 'easy', 'medium', or 'hard', according to whether its difficulty rank is in the first, second, or third tertile, respectively.

**Description of BASELINEEZ.** Recall that SALT aims to implicitly utilize small teacher LM provided supervision on the easy regions of the data distribution. To highlight the value of leveraging the teacher-provided supervision on these regions, we consider a natural baseline, namely BASELINEEZ, that explicitly train on only easy training instance during the early phase of pre-training *without* KD. In particular, while pre-training of 2.8B LM, BASELINEEZ consists of two stages: 1) Stage 1 (first 36K steps) trains on *only* easy instances *without* KD; and 2) Stage 2 (last 172K steps) performs standard pre-training. For Stage 1, we select easy instances as follows. Each sequence $\mathbf{x}$ in the Pile dataset is assigned a score equal to the fraction of times the ground truth token is in top-$k$ predictions of the selection model $\zeta_{n_0}$:

$$S_{\zeta_{n_0},k}(\mathbf{x}) = \text{mean}\Big(\mathbf{1}\Big\{x_t \in \text{argtop}_k(P_{\zeta_{n_0}}(\cdot|\mathbf{x}_{<t})); \ t \in [T]\Big\}\Big) \tag{70}$$

Then we select the top-40% sequences based on this score. As per the above selection criterion, a sequence is more likely to be selected for Stage 1 if a selection model finds it *easy to predict* most of the tokens in the sequence. The selection model $\zeta_{n_0}$ is an early checkpoint of 1.5B SLM, similar to SALT$_{\text{DS}}$ (cf. § 5.1).

**Additional results.** In Tables 24, 25 and 26, we report the results for SQuAD-v2, TriviaQA and LAMBADA respectively, sliced by difficulty level. Note that we focus on the 2.8B parameter LMs, with a 1.5B LM serving as the SLM teacher for SALT and RKD. See § 5.4 for the discussion on the key takeaways.

Table 24: **Few-shot evaluation on different buckets of SQuAD-v2 for 2.8B LMs.** Each number shows average Exact Match scores on the corresponding bucket. We use gray , green , and red to highlight the results similar to, better than, and worse than BASELINE performance, respectively.

|  | Evaluation stage (steps) | Easy | Medium | Hard |
|---|---|---|---|---|
| SLM | Final (208K) | 1.00 | 0.30 | 0.00 |
| BASELINE | | 0.86 | 0.41 | 0.23 |
| BASELINEEZ | Early (36K) | 0.72 | 0.30 | 0.13 |
| RKD | | 0.86 | 0.37 | 0.17 |
| SALT | | 0.86 | 0.37 | 0.17 |
| BASELINE | | 0.89 | 0.47 | 0.28 |
| BASELINEEZ | Final (208K) | 0.86 | 0.43 | 0.23 |
| RKD | | 0.91 | 0.42 | 0.20 |
| SALT | | 0.89 | 0.50 | 0.29 |

Table 25: **Few-shot evaluation on different buckets of TriviaQA** for 2.8B LMs. Each number shows average Exact Match scores on the corresponding bucket. We use gray , green , and red to highlight the results similar to, better than, and worse than BASELINE performance, respectively.

| | Evaluation stage (steps) | Easy | Medium | Hard |
|---|---|---|---|---|
| SLM | Final (208K) | 0.90 | 0.00 | 0.00 |
| BASELINE | | 0.63 | 0.11 | 0.08 |
| BASELINEEZ | Early (36K) | 0.28 | 0.11 | 0.05 |
| RKD | | 0.67 | 0.10 | 0.06 |
| SALT | | 0.67 | 0.10 | 0.06 |
| BASELINE | | 0.80 | 0.28 | 0.22 |
| BASELINEEZ | Final (208K) | 0.80 | 0.26 | 0.17 |
| RKD | | 0.79 | 0.14 | 0.11 |
| SALT | | 0.81 | 0.27 | 0.23 |

Table 26: **Few-shot evaluation on different buckets of LAMBADA** for 2.8B LMs. Each number shows average Accuracy on the corresponding bucket. We use gray , green , and red to highlight the results similar to, better than, and worse than BASELINE performance, respectively.

| | Evaluation stage (steps) | Easy | Medium | Hard |
|---|---|---|---|---|
| SLM | Final (208K) | 0.87 | 0.00 | 0.00 |
| BASELINE | | 0.47 | 0.12 | 0.12 |
| BASELINEEZ | Early (36K) | 0.31 | 0.08 | 0.07 |
| RKD | | 0.56 | 0.11 | 0.12 |
| SALT | | 0.56 | 0.11 | 0.12 |
| BASELINE | | 0.70 | 0.29 | 0.28 |
| BASELINEEZ | Final (208K) | 0.71 | 0.29 | 0.27 |
| RKD | | 0.65 | 0.17 | 0.17 |
| SALT | | 0.78 | 0.38 | 0.36 |

Table 27: **Domain-wise few-shot performance** of 2.8B pre-trained LMs. This table is an expansion of Table 2 as it also includes the performance of a 2.8B LM trained via BASELINEEZ. Notably, BASELINEEZ leads to significantly poorer performance even compared to BASELINE. In contrast, using a 1.5B SLM as a teacher in the KD phase, SALT and SALT$_{DS}$ already outperform BASELINE in terms of average few-shot performance at 70% of the training step budget, thereby improving both training efficiency *and* model quality. RKD (i.e., naïvely distilling from the 1.5B SLM throughout pre-training) performs much worse than BASELINE. The (second-)best results for each domain are (underlined) **boldfaced**.

| Domain (# Tasks) | SLM | BASELINE | BASELINEEZ | RKD | SALT | | SALT$_{DS}$ | |
|---|---|---|---|---|---|---|---|---|
| | | @100% steps | @100% steps | @100% steps | @70% steps | @100% steps | @70% steps | @100% steps |
| **World Knowledge (4)** | 15.90 | 22.19 | 19.19 | 18.69 | 21.59 | **22.70** | 20.64 | 21.72 |
| **Reading Comprehension (4)** | 46.30 | 53.00 | 51.38 | 51.00 | 53.55 | 54.55 | 54.35 | **54.93** |
| **Commonsense Reasoning (7)** | 57.76 | 61.99 | 60.64 | 58.30 | 61.27 | 61.67 | 62.00 | **62.10** |
| **LAMBADA (1)** | 26.90 | 36.20 | 40.40 | 31.10 | 50.70 | 48.30 | 48.00 | **53.00** |
| **SuperGLUE (8)** | 61.59 | 65.53 | 64.20 | 62.91 | **66.30** | 65.28 | 65.99 | 65.58 |
| **NLG (3)** | 3.13 | 4.60 | 4.47 | 3.40 | 4.63 | 4.73 | 4.80 | **4.83** |
| **MBPP (1)** | 9.60 | 16.20 | 13.40 | 11.40 | 15.60 | 17.00 | 16.60 | **17.80** |
| **Average (28)** | 42.56 | 47.32 | 45.98 | 44.39 | 47.86 | 47.94 | 47.89 | **48.26** |