# OpenReview forum: "A Little Help Goes a Long Way: Efficient LLM Training by Leveraging Small LMs"
_ICLR.cc/2026/Conference — Submitted to ICLR 2026_

### Official Review · Reviewer_SNmq · 2025-10-25

**Soundness:** 2
**Presentation:** 2
**Contribution:** 2
**Rating:** 4
**Confidence:** 3

**Summary:**

This paper proposes SALT (Small model Aided Large model Training), a two-stage pre-training method for Large Language Models (LLMs). It leverages a Small Language Model (SLM) as a teacher for Knowledge Distillation (KD) in the initial stage, followed by standard pre-training. An optional extension $SALT_{DS}$ uses the SLM to select "challenging yet learnable" data for the KD phase. The authors provide a theoretical analysis presenting risk bounds for KD in the language modeling context. Empirically, SALT is shown to achieve performance comparable to or better than a standard baseline with fewer training steps (70%), resulting in significant wall-clock time savings (25-29%) for 2.8B and 8.6B models, and also improves performance after SFT.

**Strengths:**

The paper tackles the critical and practical problem of reducing the high computational cost of LLM pre-training.

It demonstrates significant wall-clock time savings (25-29%) on large-scale models (2.8B, 8.6B) while maintaining or improving performance over the baseline.

The paper provides a theoretical analysis by developing risk bounds for knowledge distillation specifically in the autoregressive language modeling context.

The improvements from SALT pre-training are shown to carry over to downstream tasks after supervised fine-tuning.

**Weaknesses:**

The core components (small-to-large KD, early-stage KD, data selection via small models) are not new. The claimed novelty rests on a specific combination of these ideas, and the theoretical contribution appears to be an application of existing analyses to this specific setting.

The paper lacks direct empirical comparisons against stronger, contemporary baselines for both data selection and KD scheduling. It is also unclear if the standard baseline was sufficiently tuned.

**Questions:**

Can you clarify the specific novel contributions over prior work (like Qin et al., 2022, Ankner et al. 2024) beyond scale and the specific data selection heuristic?

The method performed poorly with a 0.5B teacher. What is the explanation for this performance degradation, and what does it imply for the method's practical limits?

If training step is longer, does the improvement disappear?

If training data is more recent clean one like fineweb-edu, does the improvement disappear?

How about the cost of hyper parameter search? And, generalization toward reuse of the selected hyperparameters to other settings?

---

> ### Author Response · Authors · 2025-11-23
> **Response to Reviewer SNmq (Part 1)**
>
> We thank the reviewer for their effort in reviewing our submission. We are happy to learn that the reviewer recognized the criticality of the problem we studied while also acknowledging the significance of our results towards improving both performance and training efficiency for large-scale models. Below, we provide a point-by-point response to the questions/concerns raised by the reviewer.
>
> > The core components (small-to-large KD, early-stage KD, data selection via small models) are not new. The claimed novelty rests on a specific combination of these ideas…Can you clarify the specific novel contributions over prior work (like Qin et al., 2022, Ankner et al. 2024) beyond scale and the specific data selection heuristic?
>
> Thank you for your questions. To the best of our knowledge, ours is the first work that provides a **rigorous statistical framework to study KD for language modeling** (see the discussion in L472-473 in Section 6 and L1035-1044 in Appendix A.4). Our work provides novel risk bounds for KD in the language modeling setting (see the next response regarding the novelty of our analysis) which are **not** just restricted to KD from smaller teachers, and cover even the typical practical settings where a larger teacher is used during KD (see, e.g., arXiv:2503.19786 and blog by Meta on Llama 4). Furthermore, our bounds also subsume standard pre-training without KD as a special case and provide novel risk bounds for this setting as well (see relevant discussion in L55-57 in the introduction and L199-205 in Section 3.2).
>
> Guided by our analysis, we propose to utilize selective knowledge transfer from smaller teachers in an implicit manner via SALT, which can be further made more explicit by incorporating data selection as in SALT$\_{\rm DS}$.
>
> Unlike Qin et al. 2022, our exploration is grounded in rigorous theoretical foundation. Qin et al. 2022 do not explore the utility of data selection via small LM, and as stated in L457-459 (Section 6), our exploration demonstrates the utility of adaptive distillation from smaller LM on a large scale, covering its advantage towards improving both general model capabilities (as measured by a comprehensive few-shot evaluation) as well as target mode quality (as measured by post-SFT performance on specific downstream tasks). Finally, we also justify that the key intuition behind SALT holds in practice by dissecting the model performance on easy vs. hard examples as the training progresses in Section 5.4.
>
> As for the distinction between our work and Ankner et al. 2024, besides utilizing a different aggregation mechanism, our selection mechanism crucially prevents noisy (unlearnable) tokens from influencing the data selection procedure (see L274-275 and L463-464).
>
> Overall, we believe that our work provides the first comprehensive treatment to the topic of leveraging small LMs to improve the pre-training of larger LMs, providing a novel theoretical framework that guides the utilization of small LMs for both richer supervision (soft-labels) as well as data selection to demonstrably improve both quality and efficiency at large scale.
>
>
>
> > …and the theoretical contribution appears to be an application of existing analyses to this specific setting.
>
> We respectfully disagree that our theoretical contributions are merely an application of existing analysis to a specific setting. The main challenge in analyzing KD for language modeling stems from the dependencies among different tokens in a sequence, which is not present in prior works (see L471-473 and Appendix A.4) that focus on analyzing KD in non-sequential learning settings. This required us to derive our results from first principles by leveraging **fundamentally different techniques**; e.g., our proof of Theorem 3.4 leverages martingale-based analysis and relies on a natural stability condition specific to LMs (Assumption 3.3), which we empirically validate in Fig. 2 and Appendix C.
>
> As we have stated in L56-57 of introduction and L199-204, our results also provide novel excess risk bounds for standard pre-training (*without* KD) after substituting $\omega=0$. Even in this setting, our different proof techniques allow our risk bounds to offer desirable traits compared to the prior works such as Lotfi et al. (2024b) and Zekri et al. (2024). For example, unlike Lotfi et al. (2024b), our bound with $\omega = 0$ can guarantee test-time generalization for the contexts unseen during training (L202-203). Similarly, different from Zekri et al. (2024) that connects LMs to Markov chains, our bound holds under a more natural and less restrictive condition in Assumption 3.3.

---

> ### Author Response · Authors · 2025-11-23
> **Response to Reviewer SNmq (Part 2)**
>
> > The paper lacks direct empirical comparisons against stronger, contemporary baselines for both data selection and KD scheduling. It is also unclear if the standard baseline was sufficiently tuned.
>
> We believe that our Baseline is a fair and natural baseline for our empirical evaluation. Note that both BASELINE (standard pre-training) and SALT share the same underlying model architecture, data, and training framework. Thus, our experiments provide a clear signal about the utility that adaptive knowledge transfer from a smaller teacher LM brings compared to the standard pre-training methods that is widely used to train LLMs.
>
> Similarly, the purpose of SALT$\_{\rm DS}$ is to evaluate if data selection can further enhance SALT which only utilizes the teacher provided supervision, and our experiments also provide an affirmative answer to this in a controlled setting. In light of these positive results, as we state in L483-485, a more comprehensive study of data selection via small LM is a promising direction for future exploration.
>
> In our experiments, all the non SALT-specific hyperparameters such as distillation weight $\omega$ and teacher temperature $\rho$, including optimizer, peak learning rate, and learning rate schedule are shared between BASELINE and SALT (Appendix E); the same holds for the SFT (L410-412). As for $\omega$ and $\rho$, they were selected with a hyperparameters search at small scale experiments (see L309-310 and Appendix G) and were kept fixed for all the experiments. Our ablation study in Appendix K shows the robustness of model performance with respect to $n\_{\rm KD}$ for a large range of values of $n\_{\rm KD}$. Thus, the consistent gains we observe with SALT at different sizes for teachers and students as well as different architectures for teacher and students (e.g., deep & narrow vs. shallow & wide and multi-head attention vs. multi-query attention; see Table 10 in Appendix E) highlight the robustness of SALT with respect to hyperparameter choices and inspire a confident in the effectiveness of SALT.
>
> > The method performed poorly with a 0.5B teacher. What is the explanation for this performance degradation, and what does it imply for the method's practical limits?
>
> The ablation in Table 3 aims to test the limit of the SALT framework by varying the student-teacher size ratio. As this ratio increases, i.e., the teacher becomes much weaker/smaller, it becomes more challenging to leverage the teacher to improve the larger student. Our results in Table 3 shows that, for 8.6B sized student LM, when the teacher is 17x smaller, SALT does not help. It’s very likely that the bias introduced by such a significantly smaller teacher on certain data domains significantly outweighs the benefit of variance reduction and degrades the model performance (cf. Section 3.3).
>
> That said, as noted in L400-401, even with 17x smaller teacher, SALT improves the final performance of the student on 4 out of 7 evaluation domains (cf. Table 18). This hints at a couple of alternative practical approaches to leverage significantly smaller teacher LMs:
>
>  - One should leverage KD for only certain data sources, e.g., code data; or
>  - One could leverage multiple significantly smaller teachers where each teacher is trained to be expert on a particular domain, e.g., a code expert teacher, a reasoning expert, a creative expert. Note that, it is feasible for a small LM to master a particular domain even if it can not achieve good performance on all the domains (see, e.g., arXiv:2306.11644). Now, during training, one can leverage different experts on different training sequences depending on the domain those sequences belong to.
>
> We believe that both of these directions constitute interesting directions for future work. Please also see L482-483 in the conclusion section.
>
> > If training step is longer, does the improvement disappear?
>
> We train the models on 545B tokens which is reasonably large for the model scales we considered. While we can’t say what will happen if we train for longer, we can compare the models at 70% and 100% steps and see if the gains are diminishing. For the 8.6B Baseline, the average few-shot scores are 51.21 and 51.73 at 70% and 100% steps whereas the SALT 8.6B model achieves 52.25 and 52.96 at the same fractions of training. The gaps of 1.04 and 1.23 at 70% and 100% steps indicate no diminishing returns as we train for longer at least until this horizon of 545B tokens.
>
> Here, we would also like to draw the reviewer’s attention to a relevant exploration in Appendix K (L2381-2387 and Table 21). SALT can potentially take advantage of a longer trained smaller teacher LM to increase the performance gap with respect to Baseline even more, further underscoring the continued importance of KD.

---

> ### Author Response · Authors · 2025-11-23
> **Response to Reviewer SNmq (Part 3)**
>
> > If training data is more recent clean one like fineweb-edu, does the improvement disappear?
>
> KD for language modeling has proven to be consistently beneficial in different settings (see, e.g.,  arXiv:2503.19786 and blog by Meta on Llama 4 for very recent application of KD in large scale settings). Thus, inspired by our theoretical foundation and detailed experiments, we would surmise that the adaptive/selective knowledge transfer from smaller teacher LMs (as done in SALT) would continue to be advantageous with other datasets (as all datasets consist of domains that are easy vs. hard).
>
> That said, given the prohibitive cost of exhaustive pre-training experiments, we believe that assessing the precise gains from leveraging smaller LMs for LLM pre-training in a specific setting remains a task for the practitioner interested in that specific setting.
>
>
> > How about the cost of hyper parameter search? And, generalization toward reuse of the selected hyperparameters to other settings?
>
>
> As mentioned in Section 5.1 (L309-313), we did search for SALT-specific hyperparameters at small scale which would ensure relatively small cost. Furthermore, as noted in L1887-1889 (Appendix G), we use the same set of hyperparameters for all model sizes. This not only shows the robustness/generalization of the selected parameters to various settings but also highlights that the cost of hyperparameter search can be amortized over multiple training runs, potentially for training models of different sizes.

---

> ### Comment · Reviewer_SNmq · 2025-11-24
>
> Thank you for your detailed response. I am currently reconsidering the value of the paper, particularly regarding the theoretical contributions you emphasized. This may take some time to fully assess.
>
> However, before finalizing my decision, I would like to clarify the intent behind my questions regarding novelty and baselines, as my previous comments might have been ambiguous.
>
> >The core components (small-to-large KD, early-stage KD, data selection via small models) are not new. The claimed novelty rests on a specific combination of these ideas, ...
> >Can you clarify the specific novel contributions over prior work (like Qin et al., 2022, Ankner et al. 2024) beyond scale and the specific data selection heuristic?
>
> Behind these parts, my thoughts are that the paper's empirical validation relies on straw man or weak baselines rather than the most competitive prior methods. The experiments show that the proposed method is good but does not show better than the existing ones or other options. If so, it is a little difficult to estimate the impact of the proposal, right?
> This is important for distillation and data selection. Each of them has existing studies (as the paper referred). If it avoids comparison, the work may not be "the first comprehensive treatment to the topic." Discussion without experiments (in 6. RELATED WORK) is not convincing for showing superiority.
>
> In my understanding,
> - Qin (2022): continuous shift from KD to pure learning
> - This paper (2025): discrete shift from KD to pure learning (+ theory + modern scale of training)
>
> If this comparison is correct, the novelty of "proposing the method" seems very limited and cannot be evaluated without comparisons. (And, possibly, even the introduction should have said something about Qin because this paper actually sounds like a "revisiting." I like revisiting-style paper, though.)
> And, if the paper claims the novelty about data selection, Anker or other methods may be a good target of comparison.
>
> By the way, given your response, the main contribution of this study seems to be the theory more than I expected. I'm rethinking the value later.

---

> ### Author Response · Authors · 2025-12-03
> **Response to Reviewer SNmq's latest comments (Part 1)**
>
> We thank the reviewer for responding to our rebuttal. We are glad that our rebuttal underscored the significance of our theoretical contributions and prompted the reviewer to reconsider the value of our submission.
>
> Below we respond to the key points raised by the reviewer in their final comments.
>
> > …my thoughts are that the paper's empirical validation relies on straw man or weak baselines rather than the most competitive prior methods.
>
> As highlighted in our response to Reviewer UdEv, throughout our study we mainly compare our method with **standard pre-training**, which is also referred to as BASELINE in the submission. We believe that this makes for a fair comparison as BASELINE corresponds to how LLMs are typically trained in practice, especially in settings where larger teacher models are not available to distill from. Also, BASELINE and SALT share both the same architecture and the same data, allowing us to cleanly test the utility of our proposed solution.
>
> > In my understanding, 1) Qin (2022): continuous shift from KD to pure learning 2) This paper (2025): discrete shift from KD to pure learning (+ theory + modern scale of training)…the novelty of "proposing the method" seems very limited and cannot be evaluated without comparisons.
>
> We would like to emphasize that our theoretical contributions aimed at providing a statistical framework to study KD for language modeling is not an afterthought, but **it’s the key step towards developing an understanding of when and how KD helps language modeling** (please see our initial response regarding the novelty of our theoretical contributions). It’s this understanding (see Section 3.3) that outlines how even a smaller LM can be a beneficial teacher as long as it is utilized selectively on the relatively easier domains of the data distribution. Our case study in Section 5.4 further provides a strong support for this guiding principle grounded in our rigorous analysis – the KD from a smaller teacher LM quickly helps large student LM acquire proficiency on easier domains and transition to standard training (`pure learning`) is crucial for letting the LLM achieve good performance on the rest of the domains where small teacher exhibits poor quality.
>
> We believe how one performs this transition from knowledge transfer/distillation phase to standard training phase is less important. E.g., our ablations in Appendix K (L2397-2408) already show that, for (long) pre-training, Step transition performed on-par other approaches with a smoother (more continuous) transition.
>
> Additionally, as highlighted in our submission (L457-459) and initial response, compared to Qin et al. 2022:
> our empirical exploration establishes the utility of (adaptive) knowledge transfer from smaller LM to LLM during pre-training at scale with larger LMs (upto 8.6B parameters) and longer data horizon (with ~545B tokens), while taking into account both general model capabilities (as measured by a comprehensive few-shot evaluation) as well as targeted model quality (as measured by post-SFT performance on specific downstream tasks).
> we also explore the synergistic benefit of utilizing small LMs for both providing richer supervision via soft-labels as well enabling data selection during the knowledge transfer phase.
>
> Furthermore, our comprehensive empirical study provides detailed exploration around hyperparameter robustness (see L309-314 and Appendix G), limits of student-teacher size ratios (L390-401), ablations of key design choices such as transition point $n\_{\rm KD}$ (Appendix K). All of these experiments provide clear evidence that the proposed SALT approach can improve both quality and training efficiency of modern large scale LMs.
>
>
> > …even the introduction should have said something about Qin because this paper actually sounds like a "revisiting." I like revisiting-style paper, though.
>
> Thank you for this important comment. Based on your suggestion, we plan to add the following to the revised introduction:
>
> *While prior work (Qin et al. 2022) have showcased the utility of such an approach, a rigorous understanding of how a weaker teacher LM can benefit a stronger student LM during pre-training is missing from the literature. Furthermore, it is not evident whether such (reverse) knowledge distillation can provide value at scale.*

---

> ### Author Response · Authors · 2025-12-03
> **Response to Reviewer SNmq's latest comments (Part 2)**
>
> > And, if the paper claims the novelty about data selection, Anker or other methods may be a good target of comparison.
>
> Besides the key technical differences between our work and Anker et al. 2024 that we highlighted in our initial response, i.e., different aggregation mechanism and exclusion of noisy (unlearnable) tokens from influencing the data selection procedure (see L274-275 and L463-464), our work on data selection is aimed at a different goal. Inspired by the key takeaway of our theoretical analysis, we focus on **establishing the additional value of data selection towards enhancing selective knowledge transfer** from a smaller teacher LM to larger student LM by **explicitly** prioritizing distillation on the (challenging sequences from the) relatively easier domains of the data distribution. As a result, we only leveraged data selection in Stage 1 where knowledge transfer from the small teacher is being employed. Our results in Table 2 clearly demonstrate the **added value of data selection via small LM** on top of the soft-lables provided by the small teacher LM.
>
> In contrast, Anker et al. 2024 (and a larger number of other existing works) on data selection focus on data selection as a standalone technique to improve the LM pre-training. Notably, different from our approach, Anker et al. 2024 employed data selection *throughout the LLM pre-training* and  *via ground truth token-based next-token prediction task*.
>
> In general, we recognize the large literature on data selection mechanisms (L462-470) and believe that a comprehensive exploration on how different selection mechanisms can aid in selective knowledge transfer via SALT is an interesting avenue for future research (L483-L485).

---

### Official Review · Reviewer_KDHw · 2025-10-31

**Soundness:** 3
**Presentation:** 3
**Contribution:** 2
**Rating:** 4
**Confidence:** 4

**Summary:**

This paper addresses a core problem in the large language model (LLM) field: high pre-training costs. The authors propose a novel and efficient training paradigm, leveraging small language models (SLMs) to aid the training of LLMs, which they name SALT (Small model aided large model training).

The core contributions of this method are twofold:
1.  **Theoretical Framework:** The paper first establishes a statistical framework to theoretically analyze the application of knowledge distillation (KD) in language models. Specifically, it explores the feasibility of "reverse distillation" (using a *weaker* SLM as a teacher to train a *stronger* LLM student). The theory reveals this as a **bias-variance trade-off**: soft labels from the SLM can reduce training variance, but due to its weaker capability, it also introduces bias (especially on "hard" samples).
2.  **SALT Algorithm:** Guided by this theory, the authors designed the SALT algorithm. It is a **two-stage training method**:
    * **Stage 1 (KD):** In the early phase of training ($n_{KD}$ steps), it uses the SLM as a teacher for knowledge distillation, capitalizing on its low bias and variance-reduction benefits in "easy" data regions.
    * **Stage 2 (Standard):** In the subsequent phase, it switches back to standard (ground-truth-based) next-token prediction training, allowing the LLM to learn the "hard" samples that the SLM could not master.
3.  **$SALT_{DS}$ (Data Selection):** The paper further proposes an extension, $SALT_{DS}$, where the SLM is also used for **data selection**. It uses a scoring function (Eq. 10) to filter for "**challenging yet learnable**" training sequences, specifically for the KD in the first stage.

**Experimental Results:**
The authors validate their method by training 2.8B and 8.6B parameter LLMs on the Pile dataset (using 1.5B and 2.8B SLMs as teachers).
* **Efficiency:** SALT-trained LLMs can match (or exceed) the performance of standard-trained (BASELINE) LLMs using **less than 70% of the training steps**.
* **Performance:** The final SALT models outperform the BASELINE on a wide range of few-shot benchmarks and supervised fine-tuning (SFT) tasks.
* **Time Savings:** This translates to an estimated **~25% (2.8B) to ~28% (8.6B) wall-clock time saving**.

**Strengths:**

1.  **Addresses a Critical Problem:** The work focuses on reducing LLM pre-training costs, which is a highly important and valuable research direction.
2.  **Reported Efficiency Gains:** The paper reports significant wall-clock time savings (~25-28%). If robust, this result is of high practical value.
3.  **Theoretical Framework:** The paper provides a theoretical framework that attempts to explain the "weak-teacher-strong-student" distillation mechanism via a bias-variance trade-off.

**Weaknesses:**

1.  **Unclear and Minimal Contribution of $SALT_{DS}$:** This is a **major weakness**. The paper introduces $SALT_{DS}$ (with data selection) as a key extension, yet the experimental results show it offers **no significant or consistent benefit** over the simpler SALT baseline (e.g., 8.6B model avg @100% steps: SALT 52.96 vs $SALT_{DS}$ 52.81). This makes the contribution of the data selection part *nearly void*. The authors need to justify why this more complex method (requiring an extra SLM scoring pass) is proposed if it provides no tangible benefit.
2.  **Severe Lack of Ablation for Data Selection:** The core of $SALT_{DS}$ is the scoring function in Eq. (10), which relies on a critical hyperparameter $k$ (set to 10) to define "learnability". The paper provides **no ablation or sensitivity analysis** on this $k$ value. What happens if $k=1$ or $k=50$? Without this analysis, the effectiveness of this data selection mechanism is **unproven**, and its design appears arbitrary.
3.  **Unprincipled Choice of $n_{KD}$:** The choice of the transition point $n_{KD}$ (36K steps) seems **arbitrary**. While Appendix K shows robustness for $n_{KD}$ between 20k and 60k, the paper fails to discuss how one might *principally* determine this optimal transition point. Lacking this discussion makes the method feel more like a specific "recipe" than a general approach.

**Questions:**

1.  **($SALT$ vs $SALT_{DS}$) Necessity of $SALT_{DS}$:** As noted in the weaknesses, the performance of $SALT$ and $SALT_{DS}$ is very close. Can the authors elaborate on whether the practical benefit of $SALT_{DS}$ justifies its additional complexity (i.e., a full forward pass over the pre-training data by the SLM)? Under what conditions would you expect $SALT_{DS}$ to *significantly* outperform $SALT$?
2.  **($SALT_{DS}$) Data Selection Hyperparameter $k$:** The choice of $k$ in Eq. (10) seems critical. You used $k=10$. Did you experiment with other $k$ values? For example, how would performance be affected by a very small $k$ (e.g., $k=1$, focusing only on tokens the SLM gets right) or a very large $k$ (e.g., $k=100$)? This is important for understanding the definition of "learnability."
3.  **(SALT) Choice of Transition Point $n_{KD}$:** The ablation in Appendix K (Table 22) shows $n_{KD}=60K$ has slightly better average performance (47.99) than $n_{KD}=36K$ (47.94). You mention choosing 36K for efficiency. My question is: is there a *principled* way to determine the optimal $n_{KD}$? For example, does it correspond to a point where the SLM teacher's training loss begins to "saturate," or some metric indicating that "easy" samples have been sufficiently learned?

---

> ### Author Response · Authors · 2025-11-23
> **Response to Reviewer KDHw (Part 1)**
>
> We thank the reviewer for a thorough review of our submission and for providing valuable feedback. We are happy to learn that the reviewer found our efficiency gains of high practical value and acknowledged the theoretical framework for ‘weak-teacher-strong-student’ distillation introduced by our work. Below, we provide our point-by-point response to the reviewer’s remaining questions.
>
>
> > Unclear and Minimal Contribution of SALT$\_{\rm DS}$: This is a major weakness. The paper introduces  SALT$\_{\rm DS}$ (with data selection) as a key extension, yet the experimental results show it offers no significant or consistent benefit over the simpler SALT baseline (e.g., 8.6B model avg @100% steps: SALT 52.96 vs  SALT$\_{\rm DS}$ 52.81). This makes the contribution of the data selection part nearly void. The authors need to justify why this more complex method (requiring an extra SLM scoring pass) is proposed if it provides no tangible benefit.
>
> Thank you for this comment, but **we respectfully disagree that SALT$\_{\rm DS}$ provides no tangible benefit**. Concretely, compared to baseline SALT, SALT$\_{\rm DS}$ at 100% training step budget achieves best performance in:
>
>  - **5 out of 7 domains** (few-shot) and **4 out of 6 benchmarks** (post-SFT) in the 1.5B → 2.8B setting (Table 2, Table 4)
>  - **4 out of 7 domains** (few-shot) and **3 out of 6 benchmarks** (post-SFT) in the 2.8B → 8.6B setting (Table 13, Table 14)
>
> Please note that the objective of our empirical study is to establish that the data selection via a small LM has the potential to improve the training of a larger LM. Our results for SALT$\_{\rm DS}$, as discussed above, do demonstrate this point (also see our response to Reviewer UdEv [https://openreview.net/forum?id=UrGsJphPnJ&noteId=voiNUXumjl] where ${\rm DS only}$ further highlights the utility of data selection). We acknowledge that there is scope for maximizing the gain from the data selection procedure, e.g., by optimizing the value of $k$ (also, see the response for the next question) or by utilizing a different data selection criterion itself. However, given the large **training cost** of LLM pre-training, we leave such exploration for future work (see L483-485).
>
> As for the additional **inference cost** associated with SALT$\_{\rm DS}$ or any other similar data selection method due to SLM scoring pass, unlike model training, one can perform such scoring for training sequences asynchronously in parallel, on **relatively inexpensive hardware** (e.g., TPU v5 lite for inference vs. TPU v5e pod for training), **without cross-chip communication**. Furthermore, we would like to highlight that since the data selection is independent of the LLM being pre-trained, these selected data can be leveraged to train multiple large student LMs, potentially of different sizes. This essentially amortizes the data selection cost across multiple pre-training runs.

---

> ### Author Response · Authors · 2025-11-23
> **Response to Reviewer KDHw (Part 2)**
>
> > Severe Lack of Ablation for Data Selection: The core of SALT$\_{\rm DS}$ is the scoring function in Eq. (10), which relies on a critical hyperparameter (set to 10) to define "learnability". The paper provides no ablation or sensitivity analysis on this  value. What happens if  k = 1 or k = 50 ? Without this analysis, the effectiveness of this data selection mechanism is unproven, and its design appears arbitrary.
>
> Thank you for raising this important point. Please note the parameter $k$ dictates which tokens in a training sequence $\mathbf{x}$ contribute towards the final selection score in Eq. (10). In particular, whenever a ground truth next-token does not belong to the set of top-$k$ predicted next tokens by (an early checkpoint of) SLM, we deem the token `unlearnable’ and remove it from the calculation of the final selection score.
>
> A too small value for $k$ would aggressively remove a large number of tokens from the computation of the selection score. This might hurt the training as the selection process would not take into account the tokens that larger LM could learn due to its larger model capacity, especially when it has access to the teacher provided predictive distribution as additional supervision. This issue would be further exacerbated when utilizing an early checkpoint of an SLM for selection, due to the relatively lower top-k accuracy of the early checkpoint. We would like to note that we had indeed tested a smaller value of $k=2$ which we had found detrimental for the reasons explained above. In particular, for training a 2.8B LM with the aid of a 1.5B LM, SALT$\_{\rm DS}$ with **$k=2$ achieved the average few-shot performance of 47.13** compared to the average few-shot performance of 48.26 with $k=10$.
>
> On the other hand, utilizing a much larger value of $k$, e.g., $k=50$, would result in very mild filtering and most of the tokens (including unlearnable or too hard tokens) would contribute to the selection score. This would interact poorly with the objective of SALT where we want to perform knowledge transfer from the small teacher only on the relatively easier portion of the data distribution during the early stage of LLM pre-training.
>
> We plan to include the above discussion along with the results for $k=2$ in the final version of our submission.
>
> Given the large compute cost of pre-training LLMs, we have not conducted an extensive ablation regarding the value of $k$ and focused more on demonstrating the potential additional utility of small LM-based data selection for pre-training LLM via SALT. As mentioned in the response to your previous comment, we believe that a more comprehensive exploration of various data selection strategies would be a valuable direction for a future exploration.
>
> > Unprincipled Choice of $n\_{\rm KD}$: The choice of the transition point n$\_{\rm KD}$ (36K steps) seems arbitrary. While Appendix K shows robustness for  between 20k and 60k, the paper fails to discuss how one might principally determine this optimal transition point. Lacking this discussion makes the method feel more like a specific "recipe" than a general approach…is there a principled way to determine the optimal n$\_{\rm KD}$? For example, does it correspond to a point where the SLM teacher's training loss begins to "saturate," or some metric indicating that "easy" samples have been sufficiently learned?
>
> Thank you for bringing up this point and for acknowledging our ablation study in Appendix K that shows the robustness of model performance with respect to $n\_{\rm KD}$ for a large range of values of $n\_{\rm KD}$. As mentioned in Appendix K (L2393-2396), ideally one would like to pick the smallest value of $n\_{\rm KD}$ without significantly compromising the model performance. This ensures that one achieves a favorable trade-off between model quality and training efficiency.
>
> As for the general approach to select the value of $n\_{\rm KD}$, there are several natural strategies:
> - As the reviewer suggested, one could potentially rely on the training dynamics of the SLM teacher to dictate the transition point.
> - Another potential approach (and which we have adopted in our exploration) is to track the LLM performance on a small set of representative eval benchmarks as training progresses and as the rate of performance improvement on these benchmarks starts saturating, one could switch to Stage 2.
> - Alternatively, when the BASELINE run is already available **or** its performance at different training fractions can be predicted via a scaling law, one can decide to transition to Stage 2 as soon as the gap between the performance of SALT and (predicted) performance of BASELINE starts to narrow down.
>
> We will include this discussion in the final version of our submission.

---

### Official Review · Reviewer_UdEv · 2025-10-31

**Soundness:** 3
**Presentation:** 3
**Contribution:** 3
**Rating:** 6
**Confidence:** 3

**Summary:**

The authors propose SALT (Small‑model Aided Large‑model Training), a two‑stage pre‑training recipe: early knowledge distillation (KD) from a smaller teacher followed by standard next‑token training; an extended variant (SALTDS) uses the SLM to select “challenging yet learnable” sequences for the KD phase (Algorithm 1, p. 5). A statistical framework (Theorems 3.2 & 3.4) provides excess risk bounds for language modeling under KD, highlighting a bias–variance trade‑off where SLM‑provided soft labels can reduce variance if teacher–data divergence remains small (pp. 3–4). Empirically, 2.8B/8.6B students pre‑trained on The Pile with UL2 show that SALT/SALTDS reach or exceed baseline few‑shot performance using 70% of training steps and achieve ~25–29% wall‑clock savings, with further downstream gains after supervised fine‑tuning (Tables 1–4, pp. 7–8; Appendix J, p. 44). A teacher‑size ablation (Table 3, p. 8) shows benefits diminish when the teacher is very small (0.5B).

**Strengths:**

- Clear, simple method with practical payoff. SALT’s two‑stage schedule is easy to implement and consistently improves overall few‑shot averages at the same step budget, and matches/beat baselines at 70% steps, delivering ~25–29% time savings in the authors’ TPU setup (Table 2 Table 13).

- Theory tailored to language modeling under KD. The paper gives sequence‑level generalization bounds for KD in LMs (Theorem 3.4), relating gains to reduced variance vs teacher–data divergence, and motivates selective/early KD (Remark 3.5 & §3.3).

- Thoughtful diagnostics & positioning. The histograms of  (Fig. 2; Appendix C) empirically support the stability assumption; the bucketed analyses (e.g., Table 5 on XLSum‑EN,; Appendix M) illustrate that early KD mostly helps “easy” slices, which fits the theory and design.

- Robustness checks. Ablations on transition strategies and KD duration and on teacher quality (Table 21) give a fuller picture; the method still helps across two student sizes (2.8B & 8.6B).

**Weaknesses:**

1. Fairness of the efficiency claim at 70% steps. The core headline—“SALT surpasses BASELINE at 146k (~70%) steps” (Table 2 Table 13)—is not matched with a BASELINE@70% few‑shot evaluation. The only “early” baseline shown near the main text is BASELINE@36k (Table 1 & Fig. 3), which is too early to compare to 146k. Without BASELINE@146k, it’s hard to attribute step‑efficiency to KD rather than general training dynamics. Please include BASELINE evaluated at identical step counts reported for SALT/SALTDS.

2. Theory–practice gap in key assumptions. The token‑level bound (Theorem 3.4) assumes a finite function class
Θ and bounded per‑token log‑loss (Assumption 3.1), and quantifies bias via TV distance to the data conditional distribution—a quantity that’s intractable to estimate in practice. The empirical proxy uses completions from a strong LM “oracle”, which approximates model‑to‑model divergence, not teacher‑to‑data divergence; this weakens the validation of DIV Clarify how the bounds should be interpreted operationally given these gaps.

3. Baselines could be stronger. Reverse KD (RKD) is a useful strawman but not a competitive baseline. Consider curriculum KD top‑k token KD (Appendix A.2, p. 19) with tuned  𝑘 or self‑distillation controls. For SALTDS, a data‑selection baseline that mimics the same selected subset without KD would isolate the data‑selection contribution more fairly.

4. Statistical reporting. Few‑shot results are single‑run with no seeds/variance/confidence intervals. Given small deltas (e. g., +0.62 average points for 2.8B at 100% steps; Table 2), error bars are important.

5. Cost accounting transparency. Wall‑clock savings (25–29%) depend on specific hardware and rematerialization settings. Reporting FLOPs or a normalized throughput‑adjusted cost would make claims more portable.

6. Scope of evaluation. The Pile is English‑heavy; a brief multilingual check or domain‑shift test (e.g., code/math) beyond MBPP and MATH citations would strengthen generality claims. (You do show strong LAMBADA gains—Table 2—but broader coverage would help.)

**Questions:**

1. BASELINE@146k: Can you report few‑shot averages and domain breakdowns for BASELINE at 146k steps (2.8B and 8.6B) to match the SALT reporting? This is critical for the step‑efficiency claim.

2. SALTDS selection: In Eq. (10) (p. 6), you use median of filtered per‑token losses with top‑k masking. Did you try mean/trimmed‑mean or per‑document entropy to balance “challenging yet learnable”? Any diversity constraint to avoid duplicative sequences?

## Suggestions

- Add BASELINE@70% few‑shot (and maybe @80%, @90%) to align curves across methods. Also show SALT@70% vs BASELINE@70% on key individual tasks (beyond overall averages)

- Report variability. Provide ≥3 seeds for the few‑shot suite with mean±std or 95% CIs; likewise for SFT results

- Stronger baselines. Include self‑distillation and top‑k token KD; try KD weight annealing beyond the tested linear variants (Appendix K, Table 23)

- Wider evaluation. Include multilingual (e.g., TyDi beyond English) or code/math reasoning benchmarks, and a brief data‑shift analysis to test whether SALT’s gains persist out of distribution.

---

> ### Author Response · Authors · 2025-11-23
> **Response to Reviewer UdEv (Part 1)**
>
> We thank the reviewer for taking the time to review our submission and providing their detailed feedback. Below we provide a point-wise response to your questions/concerns.
>
> > Fairness of the efficiency claim at 70% steps. The core headline—“SALT surpasses BASELINE at 146k (~70%) steps” (Table 2 Table 13)—is not matched with a BASELINE@70% few‑shot evaluation. Please include BASELINE evaluated at identical step counts reported for SALT/SALTDS
>
> Thank you for raising this important point. Based on the reviewer’s suggestion, we present the expansions of Table 2 and Table 13 below after including the Baseline performance at 70% training step budget. Please note the advantage of SALT and SALT$\_{\rm DS}$ over Baseline at 70% training step budget.
>
>
> ### Domain-wise few-shot performance of 2.8B pre-trained LMs (Expansion of Table 2 in our submission)
>
> | **Domain (# Tasks)**      |   |   | **Baseline (@70%)** |   |   | **SALT (@70%)** |   | **SALT$\_{\rm DS}$ (@70%)** |   |   | **Baseline (@100%)** |   |   | **SALT (@100%)** |   |   | **SALT$\_{\rm DS}$ (@100%)** |
> |---------------------------|---|---|---------------------|---|---|-----------------|---|-----------------------------|---|---|----------------------|---|---|------------------|---|---|------------------------------|
> | World Knowledge (4)       |   |   |   20.95  |   |   |      21.59      |   | 20.64   |   |   |        *22.19*       |   |   |     **22.70**    |   |   |    21.72 |
> | Reading Comprehension (4) |   |   |   51.65   |   |   |      53.55      |   |  54.35  |   |   |         53.00        |   |   |      *54.55*     |   |   |  **54.93**  |
> | Commonsense Reasoning (7) |   |   | 61.06    |   |   |      61.27      |   |           *62.00*           |   |   |         61.99        |   |   |       61.67      |   |   |   **62.10**          |
> | LAMBADA (1) |   |   |        39.60        |   |   |     *50.70*     |   |  48.00  |   |   |         36.20        |   |   |       48.30      |   |   |           **53.00**          |
> | SuperGlue (8)  |   |   |        63.51        |   |   |    **66.30**    |   |           *65.99*           |   |   |         65.53        |   |   |       65.28      |   |   |             65.58            |
> | NLG (3)   |   |   |         4.40        |   |   |       4.63      |   |            *4.80*           |   |   |         4.60         |   |   |  4.73 |   |   |           **4.83**           |
> | MBPP (1) |   |   |        14.00        |   |   |      15.60      |   |  16.60 |   |   | 16.20  |   |   |      *17.00*     |   |   |     **17.80**   |
> | Avg (28)|   |   |        46.17        |   |   |      47.86      |   |  47.89  |   |   |   47.32 |   |   |       47.94      |   |   |   48.26            |
>
>
>
> ### Domain-wise few-shot performance of 8.6B pre-trained LMs (Expansion of Table 13 in our submission)
>
> | **Domain (# Tasks)**      |   |   | **Baseline (@70%)** |   |   | **SALT (@70%)** |   | **SALT$\_{\rm DS}$ (@70%)** |   |   | **Baseline (@100%)** |   |   | **SALT (@100%)** |   |   | **SALT$\_{\rm DS}$ (@100%)** |
> |---------------------------|---|---|---------------------|---|---|-----------------|---|-----------------------------|---|---|----------------------|---|---|------------------|---|---|------------------------------|
> | World Knowledge (4)       |   |   |   26.04 |   |   |      27.66      |   |            28.04            |   |   |         26.91        |   |   |     **28.97**    |   |   |            *28.47*           |
> | Reading Comprehension (4) |   |   |   56.53|   |   |      56.83      |   |            56.10            |   |   |         56.40        |   |   |      *57.42*     |   |   |           **57.48**          |
> | Commonsense Reasoning (7) |   |   |        65.49        |   |   |      66.89      |   |            66.61            |   |   |         66.01        |   |   |      *67.09*     |   |   |           **67.24**          |
> | LAMBADA (1) |   |   |        59.30        |   |   |**65.50** |   | 54.30 |   |   |         58.70        |   |   |      *64.80*     |   |   |             55.00            |
> | SuperGlue (8)|   |   |        68.94        |   |   |      69.19      |   |  *71.06*           |   |   |         69.69        |   |   |       70.38      |   |   |           **71.26**          |
> | NLG (3)  |   |   |         5.17        |   |   |     **5.97**    |   |             5.23            |   |   |         5.40         |   |   |     **5.97**     |   |   |             5.30             |
> | MBPP (1) |   |   |        18.80        |   |   |      19.80      |   |           *22.80*           |   |   |         20.80        |   |   |       22.00      |   |   |           **23.20**          |
> | Avg (28) |   |   |        51.21        |   |   |      52.24      |   |            52.29            |   |   |         51.73        |   |   |       52.96      |   |   |             52.81            |

---

> ### Author Response · Authors · 2025-11-23
> **Response to Reviewer UdEv (Part 2)**
>
> > Theory–practice gap in key assumptions. The token‑level bound (Theorem 3.4) assumes a finite function class $\Theta$ and bounded per‑token log‑loss (Assumption 3.1), and quantifies bias via TV distance to the data conditional distribution—a quantity that’s intractable to estimate in practice.
>
> Both of our theoretical results Theorem 3.2 (sequence level) and Theorem 3.4 (token level) employ a bounded loss assumption as stated in Assumption 3.1. In L147-149, we discuss the prevalence of the similar assumption in the literature and how it holds due to simple implementation tricks such as loss clipping. In general, we don’t believe this assumption to be very limiting as one could relax the assumption and work with essential supremum being bounded without changing the main takeaway of the analysis.
>
> The requirement of finite function class $\Theta$ in Theorem 3.4 is a limitation of our analysis. We agree that extending our results to non-finite classes (similar to Theorem 3.2) is an interesting and challenging problem for future work as it requires carefully handling the dependence within different tokens in training sequences.
>
> We also agree with the reviewer’s comment about TV distance being an intractable quantity. However, we believe this term still sheds conceptual light on how the excess risk decays. Indeed, note that the TV distance term in both Theorem 3.2 and 3.4 only depends on the ground truth distribution and teacher’s distribution. Thus, the bounds can still be utilized to understand the excess risk as one scales the number of training sequences/tokens and works with different student function classes which affect the quantities such as $U\_N, V\_N$, $C\_t$, and $V\_t$.
>
> > The empirical proxy uses completions from a strong LM “oracle”, which approximates model‑to‑model divergence, not teacher‑to‑data divergence; this weakens the validation of DIV Clarify how the bounds should be interpreted operationally given these gaps.
>
> Could the reviewer please clarify what part of the paper they are referring to while discussing `...weakens the validation of DIV.`? We suspect that the reviewer is referring to L171-182 & Appendix C, which leverages “oracle” generated completions to validate the **feasibility** of Assumption 3.3. Please note that estimation of $\widehat{\xi}\_t(\mathbf{x}; \mathbf{\theta})$ requires *multiple completions for a fixed context* which are typically not present in the training data. Assuming that a large LM has approximated the data distribution well, we rely on “oracle” generated completion. This is a reasonable approach to test the feasibility of Assumption 3.3 given the limited access to ground truth distribution $\mathscr{D}$ through training data. That said, our analysis and bound is not affected by the preciseness of this empirical validation.
>
> More generally, the relevance of our analysis to *leveraging SLMs for training LLMs* is that **it serves as a guiding principle** to motivate our two-stage training method. We acknowledge that obtaining more precise bounds that can provide quantitative measures beyond qualitative guidance still remains open and quite challenging. We hope that our work will serve as a stepping stone towards obtaining such bounds.

---

> ### Author Response · Authors · 2025-11-23
> **Response to Reviewer UdEv (Part 3)**
>
> > Baselines could be stronger. Reverse KD (RKD) is a useful strawman but not a competitive baseline. Consider curriculum KD top‑$k$ token KD (Appendix A.2, p. 19) with tuned $k$ or self‑distillation controls
>
> Throughout our study we mainly compare ourselves with standard pre-training, which is also referred to as BASELINE in the submission. We believe that this makes for a fair comparison as BASELINE corresponds to how LLMs are typically trained, especially in settings where larger teacher models are not available to distill from. Also, BASELINE and SALT share both the same architecture and the same data, allowing us to cleanly test the utility of our proposed solution. Reverse KD (RKD) is included for completeness and to underscore the need for adaptive knowledge transfer from small teacher LMs.
>
> Regarding the controls suggested by the reviewer, curriculum KD top-$k$ token KD with tuned $k$ sounds like a promising idea. Typically top-$k$ token KD is employed when one wants to annotate training data with teacher prediction in an offline manner to avoid running inference through the teacher during student training. Since LM vocabs are very large, storing the full per-token predictive distribution for the entire training dataset has prohibitive storage cost and storing only top-$k$ per-token predictions allows one to significantly reduce the storage cost. It is not immediately clear whether with fixed $k$ (even if tuned), top-$k$ token KD would outperform SALT which has access to the full distribution. **Could we please ask the reviewer if they are familiar with any prior work that has already shown promise of top-$k$ token KD for distilling from a weaker/small teacher model?** Otherwise, we believe that getting top-$k$ token KD to work in the setting where a larger LM is trained via a smaller LM would in itself be a novel algorithmic contribution, which can be a promising future research direction.
>
> As for the self-distillation control, we believe that **self-distillation would not be a fair baseline** for SALT, which is motivated from a setting where larger or same size teachers are not available to begin with. In more detail, self-distillation would require first training a large LM before employing the trained model as the teacher for another same sized large LM. Furthermore, unlike SALT where the same smaller teacher LM can be used to train multiple larger LMs of varying sizes (e.g., we train 2.8B and 8.6B LMs with the same 1.5B teacher in Table 2 and 16, respectively), which allows for amortization of the training cost of small LM (see Remark 5.2), self-distillation requires training a separate teacher for each LLM of different size. This significantly increases the training overhead.

---

> ### Author Response · Authors · 2025-11-23
> **Response to Reviewer UdEv (Part 4)**
>
> >  For SALT$\_{\rm DS}$, a data‑selection baseline that mimics the same selected subset without KD would isolate the data‑selection contribution more fairly. Need to run this experiment for SALT$\_{\rm DS}$ comparison.
>
> Thank you for this comment. The following table is an expansion of Table 2 in our submission by including the suggested experiment by the reviewer, namely ${\rm DS~only}$. As evident from the table, both SALT and SALT$\_{\rm DS}$ significantly outperform ${\rm DS only}$ at **70% training step budget**, highlighting the utility of the KD signal to realize **both** quality and efficiency wins. On the other hand, with longer training (at 100% training step), ${\rm DS only}$ performs comparable to SALT$\_{\rm DS}$, which establishes the value of the small model-based data selection scheme.
>
> | **Domain (# Tasks)**      |   |   | **Baseline (@100%)** |   |   | **SALT (@70%)** |   | **SALT (@100%)** |   |   | **SALT$\_{\rm DS}$ (@70%)** |   |   | **SALT$\_{\rm DS}$ (@100%)** |   |   | **DS only (@70%)** |   |   | **DS only (@100%)** |
> |---------------------------|---|---|----------------------|---|---|-----------------|---|------------------|---|---|-----------------------------|---|---|------------------------------|---|---|--------------------|---|---|---------------------|
> | World Knowledge (4) |   |   | *22.19*  |   |   | 21.59 |   | **22.70**  |   |   |20.64 |   |   | 21.72 |   |   | 21.06  |   |   | 21.78   |
> | Reading Comprehension (4) |   |   |  53.00 |   |   |  53.55   |   |  *54.55* |   |   |   54.35  |   |   |    **54.93**  |   |   |  52.18 |   |   |  53.25  |
> | Commonsense Reasoning (7) |   |   |  61.99   |   |   |  61.67  |   |   61.67  |   |   |   62.00   |   |   |    62.10  |   |   | *62.26* |   |   | **62.56** |
> | LAMBADA (1)  |   |   |  36.20   |   |   |  *50.70*    |   |48.30 |   |   |   48.00  |   |   |   **53.00** |   |   |  50.60 |   |   |   49.40  |
> | SuperGlue (8)   |   |   |  65.53   |   |   |     *66.30*     |   | 65.28 |   |   |   65.99  |   |   |    65.58   |   |   |  64.54 |   |   |  **66.53**  |
> | NLG (3)   |   |   |  4.60   |   |   | 4.63 |   |    *4.73*  |   |   |  4.80  |   |   |  **4.83**   |   |   |  4.27   |   |   | 4.63  |
> | MBPP (1)  |   |   | 16.20  |   |   | 15.60  |   | 17.00 |   |   |  16.60  |   |   |   **17.80**   |   |   |  15.80   |   |   |  **17.80**   |
> | Avg (28)   |   |   |   47.32  |   |   |  47.86   |   | 47.94   |   |   |  47.89  |   |   |   48.26   |   |   |   47.29 |   |   |   48.26  |
>
>
> > Statistical reporting. Few‑shot results are single‑run with no seeds/variance/confidence intervals. Given small deltas (e. g., +0.62 average points for 2.8B at 100% steps; Table 2), error bars are important…Report variability. Provide ≥3 seeds for the few‑shot suite with mean±std or 95% CIs; likewise for SFT results
>
> Please note that  **it is standard in the LM pre-training literature to only report a single pre-training run** (e.g., arXiv:2404.14219, arXiv:2402.19427). This is owing to the significant expense of LM pre-training.
>
> Regarding the seemingly small deltas in few-shot performance, as stated in Remark 5.1 (L347-352), one needs to contextualize those gains with respect to model scaling. For example, in terms of average few-shot performance, from Table 2, @100% steps, SALT has a gain of 47.94 - 47.32 = 0.62 over Baseline, which is  **~13% of the gain** from nearly doubling the model size – from 42.56 for 1.5B to 47.32 for 2.8B. Similarly, from Table 13, SALT realizes a gain of 52.96 - 51.73 = 1.23, which is  **~28% of the gain** of 51.73 - 47.32 = 4.41 realized by increasing model size ~3x from 2.8B to 8.6B. Thus, by contextualizing the improvements relative to the gains from a significant increase in model size (which typically brings large gains), we can ascertain the utility of SALT.
>
> As for reporting variability for SFT results, we followed the reviewer's suggestion. Below, we present mean$\pm$std (computed over 5 runs) post-SFT performance for 2.8B sized LM on two representative datasets – GSM8K and XSum –  from Table 4 of the submission (see Section 5.3 for further details). The presented results strengthen our claims that SALT (with or without data selection) enables sizable improvements for the downstream domains.
>
> ### SFT results for 2.8B LMs with 5 runs
>
> |   |   |   | **GSM8K (Accuracy)** |   |   | **XSum** *(Rouge-1)* |   | **XSum** *(Rouge-2)* |   |   | **XSum** *(Rouge-Lsum)* |
> |----------------------|---|---|----------------------|---|---|----------------------|---|----------------------|---|---|-------------------------|
> | **Baseline**  |   |   | 32.04 $\pm$ 0.58 |   |   | 43.39 $\pm$ 0.04   |   | 21.04 $\pm$ 0.05   |   |   |  35.84 $\pm$ 0.06 |
> | **SALT**   |   |   |   35.00 $\pm$ 0.29|   |   |   43.47 $\pm$ 0.06   |   |21.20 $\pm$ 0.06   |   |   | 36.00 $\pm$ 0.10 |
> | **SALT$\_{\rm DS}$** |   |   | 35.48 $\pm$ 0.27 |   |   |43.79 $\pm$ 0.09 |   | 21.46 $\pm$ 0.06 |   |   | 36.24 $\pm$ 0.11|

---

> ### Author Response · Authors · 2025-11-23
> **Response to Reviewer UdEv (Part 5)**
>
> > Cost accounting transparency. Wall‑clock savings (25–29%) depend on specific hardware and rematerialization settings. Reporting FLOPs or a normalized throughput‑adjusted cost would make claims more portable.
>
> Thank you for the suggestion. Please find below the table that lists FLOPS and TPU-hours required to train 2.8B and 8.6B LMs via BASELINE and SALT.
>
> | Model | #Params | #Teacher Params | Steps/s (S1) | Steps/s (S2) | Steps (S1) | Steps (S2) | GPU Hours | Frac. GPU (vs Base) | FLOPs (6ND) | Frac. FLOPs (vs Baseline) |
> | :--------------------------------------------------------- | :--- | :--- | :--- | :--- | :--- | :--- | :--- | :--- | :--- | :--- |
> | 1.5B | 1,555,335,296 | - | 1.613 | - | 208,000 | - | 36,680 | - | 5.09E+21 | - |
> | 2.8B | 2,820,938,880 | - | 0.887 | - | 208,000 | - | 66,702 | - | 9.23E+21 | - |
> | 8.6B | 8,600,811,648 | - | 0.630 | - | 208,000 | - | 93,912 | - | 2.81E+22 | - |
> | SALT 1.5B to 2.8B (70% steps) | 2,820,938,880 | 1,555,335,296 | 0.757 | 0.887 | 36,000 | 110,000 | 48,802 | 0.732 | 6.77E+21 | 0.734 |
> | SALT 2.8B to 8.6B (70% steps) | 8,600,811,648 | 2,820,938,880 | 0.562 | 0.630 | 36,000 | 110,000 | 67,886 | 0.723 | 2.03E+22 | 0.721 |
> | SALT 1.5B to 8.6B (70% steps) | 8,600,811,648 | 1,555,335,296 | 0.590 | 0.630 | 14,000 | 132,000 | 66,347 | 0.706 | 1.99E+22 | 0.706 |
>
>
>
> > Scope of evaluation. The Pile is English‑heavy; a brief multilingual check or domain‑shift test (e.g., code/math) beyond MBPP and MATH citations would strengthen generality claims. (You do show strong LAMBADA gains—Table 2—but broader coverage would help.)
>
> Thank you for acknowledging the performance gains we already demonstrated on LAMBADA and MBPP. Based on the reviewer’s suggestion we present the few-shot performance (F1 score) of BASELINE, SALT, SALT$\_{\rm DS}$ on the TyDi QA eval benchmark. Please note that SALT and SALT$\_{\rm DS}$ outperform BASELINE on both entire TyDi QA benchmark as well as the **Non-English** portion of the benchmark.
>
>
> ### Few-shot performance of 2.8B pre-trained LMs on the TyDi-QA benchmark. Note that SALT and SALT$\_{\rm DS}$ pre-train 2.8B LM with the help of a 1.5B LM.
>
> |                          |   |   | **Baseline** |   |   | **SALT (@70%)** |   | **SALT (@100%)** |   |   | **SALT$\_{\rm DS}$ (@70%)** |   |   | **SALT$\_{\rm DS}$ (@100%)** |
> |--------------------------|---|---|--------------|---|---|-----------------|---|------------------|---|---|-----------------------------|---|---|------------------------------|
> | TyDiQA (Avg)             |   |   |     42.6     |   |   |       46.3      |   |       48.6       |   |   |             45.3            |   |   |             47.7             |
> | TyDiQA Non-English (Avg) |   |   |     40.6     |   |   |       44.7      |   |       47.1       |   |   |             43.5            |   |   |             46.0             |
>
>
> > SALT$\_{\rm DS}$ selection: In Eq. (10) (p. 6), you use median of filtered per‑token losses with top‑k masking. Did you try mean/trimmed‑mean or per‑document entropy to balance “challenging yet learnable”? Any diversity constraint to avoid duplicative sequences?
>
> These are promising methods of data selection. Our objective was to show that there is value in data selection and we picked a rule that makes intuitive sense as median is a more stable aggregation rule compared to mean. We believe that the specific rule itself is of less importance for the purpose of our empirical study to justify the value of data selection via small LM. There are potentially better ways of doing this than our proposal, probably including the ones that the reviewer suggested. We did not add any specific diversity constraints. As we state in L483-485, a more comprehensive study of data selection via small LM is a promising direction for future exploration.

---

### Official Review · Reviewer_XJTj · 2025-11-01

**Soundness:** 3
**Presentation:** 3
**Contribution:** 3
**Rating:** 4
**Confidence:** 3

**Summary:**

The paper studies the use of pretrained smaller language model (SLM) to train larger ones via knowledge distillation (KD).
The paper first provide theoretical studies of the risk bound including the effects of KD.
Then, it proposes a two-stage approach (SALT) of pretraining reusing SLM: First, perform KD to train the model (optionally select learnable data to further accelerate training). Then, pretrain the model as usual.
This utilize the intuition that SLM can help learning easier tasks, and at the latter stage the capacity of the larger model can be leveraged to further train itself.

The paper provides experimental validation comparing mainly with from-scratch pretraining and KD (without second stage). Results for post-training are also given.

**Strengths:**

- The paper provides both theoretical and empirical contributions to KD with smaller models.
- Experiments are performed with care, with ablation studies given to justify the hyperparameters used, etc.

**Weaknesses:**

1. Data selection method is not so practical as it involves using early model checkpoint, which is not typically available if one wishes to use off the shelf public models. Moreover, it involves more hyper parameters for tuning.
2. Lack of baseline: I think the proposed method should be compared to model growth methods as baselines. These methods are quite straightforward compared to SALT (simply stacking or expanding the widths works well; [2405.15319]) and are also reusing small models to accelerate the training of large ones.
3. While the theory provides some clue on why KD helps, it does not provide concrete guidance on when or how to transition from KD to pretraining without it. Using a two stage training process works well intuitively without the theory as well (weak teacher can provide supervision to a strong but blank-state student only in the beginning), making the theoretical contribution seemingly redundant.

**Questions:**

1. The accuracies seem to be converging with larger train steps. I wonder if at even larger train steps from-scratch pretraining becomes better and KD becomes unnecessary?
2. The improvement seems to be small compared to from-scratch pretraining. I wonder, when both baseline and KD are compared under equal compute (KD requires extra compute due to the inference of SLM), from-scratch pretraining could perform better?

---

> ### Author Response · Authors · 2025-11-23
> **Response to Reviewer XJTj  (Part 1)**
>
> We thank the reviewer for their valuable feedback. We are glad that the reviewer acknowledged both theoretical and empirical contributions of our submission while also noting the carefulness of our experimental study. Below, we provide a point-wise response to the reviewer’s questions/concerns.
>
> > Data selection method is not so practical as it involves using early model checkpoint, which is not typically available if one wishes to use off the shelf public models.
>
> Thank you for raising this point. Please note that **it is common practice to store intermediate checkpoints while pre-training language models**. Thus, for **in-house** model development by institutions/labs, one can leverage SALT$\_{\rm DS}$ with an early checkpoint of a small LM.
>
> If such an early checkpoint is not available, e.g., if one aims to leverage a (small) off-the-shelf public model to train a larger LM, the practitioners can rely on the available (final) checkpoint to perform data selection *or* they can simply resort to SALT without data selection which also provides significant improvements over the standard pre-training approach.
>
> > Moreover, it [data selection] involves more hyper parameters for tuning.
>
> Any selection method would typically be accompanied by selection-specific hyperparameters. In our exploration, we used the same values for $k$ and $n\_0$ – the additional hyperparameters specific to our selection method – to demonstrate the additional utility of data selection for training both 2.8B (L308-309 in Section 5.1) and 8.6B (L1949-1950 in Appendix I) models, which highlights the stability of the selection method with respect to these hyperparameters. That said, similar to other design choices (e.g., optimizer, learning rate schedule, distillation loss weight etc), practitioners can decide to optimize the hyperparameter selection depending on the amount of compute resources at their disposal.
>
> > …I think the proposed method should be compared to model growth methods as baselines. These methods are quite straightforward compared to SALT (simply stacking or expanding the widths works well; [2405.15319]) and are also reusing small models to accelerate the training of large ones.
>
> Thank you for your comment. We did discuss **stacking** (progressive training) and **warm-starting from smaller models** (via checkpoint expansion) as additional approaches to leverage smaller models to train larger LMs (see L 448-452  in our related work section which also cites [arXiv:2405.15319] among other relevant works). Notably, some of these techniques require architectural overlaps between small and large LM. On the other hand, SALT is general enough to be applicable even when SLM and LLM are from completely different model families, potentially with even different vocab/tokenizers (e.g., see [arXiv:2402.12030]).
>
> That said, as stated in the related work section (L453-455) and in the conclusion (L485), we believe that these approaches are **complementary to SALT**. E.g., one can progressively grow an LLM during pre-training while utilizing an SLM as a teacher during the early stages of the progressive growth. Given the *expensive nature of pre-training experiments*, we defer such a synergistic exploration for future investigation.

---

> ### Author Response · Authors · 2025-11-23
> **Response to Reviewer XJTj (Part 2)**
>
> > While the theory provides some clue on why KD helps, it does not provide concrete guidance on when or how to transition from KD to pretraining without it. Using a two stage training process works well intuitively without the theory as well (weak teacher can provide supervision to a strong but blank-state student only in the beginning), making the theoretical contribution seemingly redundant.
>
> Our theoretical results aim to provide excess risk analysis for **KD in language modeling setting**, which to the best of our knowledge are novel (see the discussion in L472-473 in Section 6 and L1035-1044 in Appendix A.4). We would like to highlight that our bounds are **not** restricted to KD from smaller teachers, and cover even the typical practical settings where a larger teacher is used during KD (see, e.g., arXiv:2503.19786 and blog by Meta on Llama 4). Furthermore, our bounds also subsume standard pre-training without KD as a special case and provide novel risk bounds for this setting as well (see relevant discussion in L55-57 in the introduction and L199-205 in Section 3.2).
>
> As for the relevance of our analysis to leveraging SLMs to training LLMs, **our analysis serves as a guiding principle** to motivate our two-stage training method, namely SALT with or without data selection (see, e.g., L79 and L236-244). We acknowledge that theoretically characterizing the *precise* performance of the proposed two-stage method in its entirety and deriving quantitative/concrete guidance from it is very hard. We are hopeful that we would be able to make progress in this direction in the future.
>
> In general, the theory for LLMs and distillation is evolving, and not at a stage where it can explain all the phenomena associated with distillation, especially when model-dependent dynamics (such as curriculum) are involved. In this context, we do not claim that we answer all relevant real-world LLM learning phenomena with our theory, we merely make an advance in that direction.
>
>
> > The accuracies seem to be converging with larger train steps. I wonder if at even larger train steps from-scratch pretraining becomes better and KD becomes unnecessary?
>
> Next-token prediction accuracy & log perplexity are only proxies for general purpose downstream abilities. Figure 3 and Table 1 are consistent with modern scaling laws [arXiv:2203.15556], which posit that as one increases total compute or data for a given model, the **absolute improvements** for (log-)perplexity decrease. (Perplexity plots are in Fig 8, App. L. Also, see explanation in L1564-1568.)
>
> We instead focus on **few-shot performance** as well as **post-SFT performance** to assess the benefits of SALT over Baseline, as these performances are more aligned with real world tasks. Here, as stated in Remark 5.1, **we observe significant gains when compared to naive model scaling** (see, e.g., Table 2 & 4). Focusing on the few-shot performance, from Table 2, @100% steps, SALT has a gain of 47.94 - 47.32 = 0.62 over Baseline, which is  **~13% of the gain** from nearly doubling the model size – from 42.56 for 1.5B to 47.32 for 2.8B. Similarly, from Table 13, SALT realizes a gain of 52.96 - 51.73 = 1.23, which is  **~28% of the gain** of 51.73 - 47.32 = 4.41 realized by increasing model size ~3x from 2.8B to 8.6B. Thus, by contextualizing the improvements relative to the gains from a significant increase in model size (which typically brings large gains), we can ascertain the utility of SALT. Please see Table 4 and Table 15 for the post-SFT improvements via SALT methods.
>
> Regarding whether the gains from SALT would remain after longer training, let’s consider the performance trend for the Baseline and SALT methods. As detailed in our response to Reviewer UdEV, for the 8.6B model, Baseline achieves the average few-shot performance of  51.21 and 51.73 at 70% training steps and at 100% training steps, respectively. On the other hand, SALT achieves the average few-shot performance of 52.25 and 52.96 at 70% training steps and at 100% training steps, respectively. Thus throughout the training SALT is preserving its edge over the Baseline.
>
>
> Here, we would like to draw the reviewer’s attention to a relevant exploration in Appendix K (L2381-2387 and Table 21). SALT can potentially take advantage of a higher quality (longer trained) smaller teacher LM to increase the performance gap with respect to Baseline even more, further underscoring the continued importance of KD.

---

> ### Author Response · Authors · 2025-11-23
> **Response to Reviewer XJTj (Part 3)**
>
> > The improvement seems to be small compared to from-scratch pretraining. I wonder, when both baseline and KD are compared under equal compute (KD requires extra compute due to the inference of SLM), from-scratch pretraining could perform better?
>
> We would like to note that we already **exhibit quality improvement** over the Baseline (i.e., standard pre-training OR from-scratch pre-training) while **also realizing wall clock saving** that **already takes into account the extra compute** due to the inference of SLM.
>
> For example, Table 2 shows that SALT achieves average few-shot performance of 47.86 at 70% training step budget compared to the average few-shot performance of 47.32 by Baseline at 100% training step budget. This translates to **~25% wall clock saving** (L373). Similar to Remark 5.1, one can contextualize the average performance gain 47.86 - 47.32 = 0.54 against the average performance of 47.32 - 42.56 = 4.76 realized by nearly doubling the model size from 1.5B to 2.8B – SALT provided gain is ~11% of the gain coming from doubling the model size.
>
> Similarly, Table 13 shows that SALT at 70% training step budget archives average few-shot performance of 52.24 compared to Baseline’s average performance of 51.73 at 100% training budget. This translates to **~28% wall clock saving** (L374) while still being quality positive. Notably, the average performance gain 52.24 - 51.73 = 0.51 is ~12% of the average performance gain one obtains by nearly tripling the model from 2.8B (average performance of 47.32) to 8.6B (average performance of 51.73).

---

### Meta-Review · Area_Chair_mGTj · 2025-12-25

**Summary:**

1. Incremental and Minimal Contribution: The experimental results show the proposed method offers no significant or consistent benefit over the simpler SALT baseline. The data selection method is hard to use in practice, and it involves more hyperparameters for tuning. (Reviewer XJTj, KDHw, SNmq)
2. Lack some necessary analysis: (Reviewer KDHw, SNmq)
    - Severe Lack of Ablation for Data Selection: The effectiveness of this data selection mechanism is unproven, and its design appears arbitrary.
    - The paper fails to discuss how one might principally determine this optimal transition point.
    - The method performed poorly with a 0.5B teacher. What is the explanation for this performance degradation, and what does it imply for the method's practical limits?
3. Lacks some baselines, e.g., model growth methods, data selection, KD scheduling, curriculum KD top‑k token KD, and self‑distillation control. (Reviewer XJTj, SNmq, UdEv)
4. Theoretical contribution is seemingly unclear and redundant. (Reviewer XJTj, UdEv)
5. Lacks FLOPs or a normalized throughput‑adjusted cost. (Reviewer UdEv)
6. Limited scope of evaluation: The Pile is English‑heavy; a brief multilingual check or domain‑shift test (e.g., code/math) beyond MBPP and MATH citations would strengthen generality claims. (Reviewer UdEv)

**Reviewer Concerns:**

Although the authors provided point-by-point responses to the reviewer's concerns, the paper still requires further revision. The theoretical analysis remains weak due to the gap between theory and practice, and the experimental section is still missing some important analysis and baselines to fully validate the method's actual contributions.

**Reviewer Scores:**

Reviewer XJTj: retains 4

Reviewer UdEv: retains 6

Reviewer KDHw: retains 4

Reviewer SNmq: (may) 4 --> 6

---

### Decision · Program_Chairs · 2026-01-26

Reject